# LOCAL SGD AND FEDERATED AVERAGING THROUGH THE LENS OF TIME COMPLEXITY

## ABSTRACT

We revisit the classical Local SGD and Federated Averaging (FedAvg) methods for distributed optimization and federated learning. While prior work has primarily focused on *iteration complexity*, we analyze these methods through the lens of *time complexity*, taking into account both computation and communication costs. Our analysis reveals that, despite its favorable *iteration complexity*, the *time complexity* of canonical Local SGD is provably worse than that of Minibatch SGD and Hero SGD (locally executed SGD). We introduce a corrected variant, Dual Local SGD, and further improve it by increasing the local step sizes, leading to a new method called Decaying Local SGD. Our analysis shows that these modifications, together with Hero SGD, are optimal in the nonconvex setting (up to logarithmic factors), closing the time complexity gap. Finally, we use these insights to improve the theory of a number of other asynchronous and local methods.

## 1 INTRODUCTION

We re-examine the classical Local SGD and Federated Averaging (FedAvg) approaches that solve the distributed optimization problem (McMahan et al., 2017a; Stich, 2019):

$$\min_{x \in \mathbb{R}^d} \left\{ f(x) := \frac{1}{n} \sum_{i=1}^{n} \mathbb{E}_{\xi_i} \left[ f_i(x; \xi_i) \right] \right\}, \tag{1}$$

where $f_i : \mathbb{R}^d \times \mathbb{S}_{\xi_i} \to \mathbb{R}^d$ and $\xi_i$ are random variables with distributions $\mathcal{D}_i$. Here, $n$ is the number of workers collaboratively solving the problem, where each worker $i$ can only sample stochastic gradients $f_i(x; \xi_i)$ of its local loss function $f_i(x)$. We begin our work by considering the *homogeneous* setting, where all clients share the same distribution $\mathcal{D}_i = \mathcal{D}$ and satisfy $f_i = f$ for all $i \in [n] := \{1, \ldots, n\}$. We discuss the *heterogeneous* setting in Section 2.3. Such problems arise in the training of modern machine learning models, large language models, and in federated learning (Touvron et al., 2023; Konečný et al., 2016).

Unlike most previous works that focus on *iteration complexity*, i.e., the number of communication rounds needed so as to reach an $\varepsilon$–stationary point, we analyze methods from the perspective of *time complexity*(Tyurin and Richtárik, 2023; Tyurin and Richtárik, 2024; Tyurin, 2024). In particular, we consider the following assumption.

**Assumption 1.1** (Computation and Communication Time)**.**
- *Computing a single stochastic gradient takes* exactly *$h$ seconds.*
- *Communicating vectors (from $\mathbb{R}^d$) among the workers, e.g., via a server or an* AllReduce *operation, requires* exactly *$\tau$ seconds.*

Our main goal is to investigate Local SGD and the other aforementioned methods under this realistic assumption, to compare them, and to offer a new perspective. Looking ahead, this leads to new and unexpected insights about Local SGD and other local methods.

**Algorithm 1:** Local SGD (canonical version)

**Require:** initial point $x^0$, local stepsize $\eta_\ell$, # communication rounds $R$, number of local steps $K$

1: **for** $t = 0, 1, \ldots, R-1$ **do**
2:   **for** worker $i \in \{1, \ldots, n\}$ **in parallel do**
3:     $z_{i,0}^t = x^t$
4:     **for** $j = 0, \ldots, K-1$ **do**
5:       $z_{i,j+1}^t = z_{i,j}^t - \eta_\ell \nabla f(z_{i,j}^t; \xi_{i,j}^t)$, $\xi_{i,j}^t \sim \mathcal{D}$
6:     **end for**
7:   **end for**
8:   $x^{t+1} = \frac{1}{n} \sum_{i=1}^n z_{i,K}^t$

   $\equiv x^t - \frac{\eta_\ell}{n} \sum_{i=1}^n \sum_{j=0}^{K-1} \nabla f(z_{i,j}^t; \xi_{i,j}^t)$

9: **end for**

**Algorithm 2:** Minibatch SGD

**Require:** initial point $x^0$, global stepsize $\eta_g$, # communication rounds $R$, batch size $K$

1: **for** $t = 0, 1, \ldots, R-1$ **do**
2:   **for** worker $i \in \{1, \ldots, n\}$ **in parallel do**
3:     **for** $j = 0, \ldots, K-1$ **do**
4:       Calculate[3] $\nabla f(x^t; \xi_{i,j}^t)$, $\xi_{i,j}^t \sim \mathcal{D}$
5:     **end for**
6:   **end for**
7:   $x^{t+1} = x^t - \eta_g \sum_{i=1}^n \sum_{j=0}^{K-1} \nabla f(x^t; \xi_{i,j}^t)$
8: **end for**

**Algorithm 3:** Hero SGD (on a single worker)

**Require:** # iterations $R$, initial point $x^0$, stepsize $\eta$

1: **for** $t = 0, 1, \ldots, R-1$ **do**
2:   $x^{t+1} = x^t - \eta \nabla f(x^t; \xi^t), \xi^t \sim \mathcal{D}$
3: **end for**

## 1.1 PREVIOUS WORK

At the beginning, we investigate the Local SGD, Minibatch SGD, and Hero SGD[1] methods described in Algorithms 1, 2, and 3[2]. These are among the most explored and well-studied distributed methods. In Local SGD, each worker performs $K$ local SGD steps, followed by periodic synchronization steps via a server or the AllReduce operation. In contrast, each worker in Minibatch SGD computes $K$ stochastic gradients at the same point. The idea behind both methods is to reduce overall communication by choosing a large $K \gg 1$, thus making the costly communication time $\tau$ less significant. The Hero SGD method is the standard SGD algorithm (Lan, 2020), executed locally on a single worker without communication. We consider the following standard assumptions:

**Assumption 1.2.** *$f$ is differentiable and $L$–smooth, i.e., $\|\nabla f(x) - \nabla f(y)\| \leq L \|x - y\|$ for all $x, y \in \mathbb{R}^d$. We define $\Delta := f(x^0) - \inf_{x \in \mathbb{R}^d} f(x)$, where $x^0$ is a starting point of numerical methods.*

**Assumption 1.3.** *The stochastic gradients satisfy $\mathbb{E}_\xi[\nabla f(x; \xi)] = \nabla f(x)$ (unbiasedness) and $\mathbb{E}_\xi[\|\nabla f(x; \xi) - \nabla f(x)\|^2] \leq \sigma^2$ (bounded variance) for all $x \in \mathbb{R}^d$, where $\sigma^2 \geq 0$.*

The goal in the nonconvex world is to find an $\varepsilon$–stationary point: a (possibly) random point $\bar{x} \in \mathbb{R}^d$ such that $\mathbb{E}[\|\nabla f(\bar{x})\|^2] \leq \varepsilon$. In the convex setting, we consider the assumption below and want to find an $\varepsilon$–solution, a point $\bar{x}$ such that $\mathbb{E}[f(\bar{x})] - f(x^*) \leq \varepsilon$.

**Assumption 1.4.** *$f$ is convex and attains its minimum at least at a point $x^* \in \mathbb{R}^d$. We define $B := \|x^0 - x^*\|$ in the convex setting, where $x^*$ is the closest minimum to $x^0$.*

The state-of-the-art *iteration complexity* analysis of Local SGD (Algorithm 1) for *convex* problems was obtained by Khaled et al. (2020); Woodworth et al. (2020), which is optimal *for this method* due to the result by Glasgow et al. (2022). In the *nonconvex* setting, under Assumptions 1.2 and 1.3, Koloskova et al. (2020) provide the current state-of-the-art *iteration complexity* to find an $\varepsilon$–stationary point. Considering $\rho$–weak convexity, Luo et al. (2025) improved the rate. The iteration complexity of Minibatch SGD (Algorithm 2) can be easily inferred from the classical analysis of SGD, since Minibatch SGD is essentially SGD with a batch size of $nK$. The iteration complexities of the discussed results are presented in Table 1.

---

[1] A locally executed SGD on a single worker without communication.

[2] While it may seem fair to compare Algorithms 1 to 3 by running Hero SGD for $KR$ iterations, we follow the convention of running Hero SGD for only $R$ iterations. Although this choice will lead to weaker convergence bound than in the fair setting, it suffices to demonstrate the suboptimality of Local SGD in terms of time complexity.

[3] In the heterogeneous setting, we calculate $\nabla f_i(\cdot; \cdot)$.

Table 1: Known iteration complexities of Minibatch SGD and Local SGD in the homogeneous convex and nonconvex settings. We assume Assumptions 1.2 and 1.3 in the nonconvex setting, and Assumptions 1.2, 1.3, and 1.4 in the convex setting. Abbr.: $L$ = smoothness constant; $x^0$ = starting point; $\Delta = f(x^0) - \inf_{x \in \mathbb{R}^d} f(x)$; $\sigma^2$ = variance of stochastic gradients; $n$ = # of workers; $K$ = # of local steps between synchronizations; $R$ = # of communication rounds (or iterations for Hero SGD); $B = \left\| x^0 - x^* \right\|$. **One of the main contributions of this paper is to explain that this comparison is misleading, and that a better one is given in Table 2.**

| **Convex Setting** | | **Nonconvex Setting** | |
| --- | --- | --- | --- |
| **Algorithm** | **Iteration Complexity** | **Algorithm** | **Iteration Complexity** |
| Hero SGD (no communications) | $\frac{LB^2}{R} + \frac{\sigma B}{\sqrt{R}}$ | Hero SGD (no communications) | $\frac{L\Delta}{R} + \sqrt{\frac{L\sigma^2\Delta}{R}}$ |
| Minibatch SGD | $\frac{LB^2}{R} + \frac{\sigma B}{\sqrt{nKR}}$ | Minibatch SGD | $\frac{L\Delta}{R} + \sqrt{\frac{L\sigma^2\Delta}{nKR}}$ |
| Local SGD (Alg. 1) (Khaled et al., 2020) | $\frac{LB^2}{KR} + \frac{\sigma B}{\sqrt{nKR}} + \frac{L^{\frac{1}{3}}\sigma^{\frac{2}{3}}B^{\frac{4}{3}}}{K^{\frac{1}{3}}R^{\frac{2}{3}}}$ | Local SGD (Alg. 1) (Koloskova et al., 2020) | $\frac{L\Delta}{R} + \sqrt{\frac{L\sigma^2\Delta}{nKR}} + \frac{(L\sigma\Delta)^{\frac{2}{3}}}{K^{\frac{1}{3}}R^{\frac{2}{3}}}$ |
| Lower bound for Local SGD (Glasgow et al., 2022) | $\frac{LB^2}{KR} + \frac{\sigma B}{\sqrt{nKR}} + \min\left\{ \frac{L^{\frac{1}{3}}\sigma^{\frac{2}{3}}B^{\frac{4}{3}}}{K^{\frac{1}{3}}R^{\frac{2}{3}}}, \frac{\sigma B}{\sqrt{KR}} \right\}$ | Local SGD (Alg. 1) (Luo et al., 2025) (with $\rho$–weak convexity) | $\left(\frac{L}{K} + \rho\right)\frac{\Delta}{R} + \sqrt{\frac{L\sigma^2\Delta}{nKR}} + \frac{(L\sigma\Delta)^{\frac{2}{3}}}{K^{\frac{1}{3}}R^{\frac{2}{3}}}$ |

While the iteration complexities seem to suggest that Local SGD is better than Hero SGD/Minibatch SGD when $K$ is large, in Sec. 1.2, 2, we prove that this is not the case, and **Local SGD is never better, but might be worse! See Table 2.**

There are many other works that consider local steps. Mishchenko et al. (2022) focus on convex optimization; Patel et al. (2022) consider a different setting from Assumption 1.3 and require the mean-squared smoothness property to use variance reduction techniques (Fang et al., 2018; Cutkosky and Orabona, 2019); Karimireddy et al. (2021); Malinovsky et al. (2023) focus on the finite-sum setting; and Jhunjhunwala et al. (2023); Li et al. (2024); Anyszka et al. (2024) analyze the problem using the proximal operator; Crawshaw et al. (2025) consider logistic regression exclusively; and Tyurin and Sivtsov (2025) develop a general framework for local and asynchronous optimization.

Local SGD (Koloskova et al., 2020; Luo et al., 2025) and SCAFFOLD (Karimireddy et al., 2020) are considered the theoretical state-of-the-art methods in our general setting because they possess a desirable theoretical property: their iteration complexities scale with the number of local iterations $K$ (see Table 1). In the convex setting, comparing the rates $\frac{LB^2}{R} + \frac{\sigma B}{\sqrt{nKR}}$ and $\frac{LB^2}{KR} + \frac{\sigma B}{\sqrt{nKR}} + \min\left\{ \frac{L^{\frac{1}{3}}\sigma^{\frac{2}{3}}B^{\frac{4}{3}}}{K^{\frac{1}{3}}R^{\frac{2}{3}}}, \frac{\sigma B}{\sqrt{KR}} \right\}$ for Minibatch SGD and Local SGD, respectively, it is clear to the community that the latter rate is significantly better when $K$ is large, and there is no doubt that Local SGD performs much better in this regime, and that local steps with periodic communication provably help (Woodworth et al., 2020; Luo et al., 2025). However, once we start comparing algorithms under Assumption 1.1, we observe that this is not the case, and Local SGD (Algorithm 1) is provably *worse* than Minibatch SGD and Hero SGD.

## 1.2 CONTRIBUTIONS

♠ **A Fresh Perspective on Local SGD.** We start our work by proving Theorems 2.1 and 2.2. While the iteration complexities in Table 1 indicate the superiority of Local SGD, the time complexities tell us the complete opposite. In particular, the lower bound (2) for the time complexity of Local SGD can be *significantly worse* than the time complexity (3) of Minibatch SGD/Hero SGD.

One can show that (2) $\geq$ (3) in all regimes (consider two cases: $h\sigma^2/\varepsilon \geq \tau L$ and $h\sigma^2/\varepsilon < \tau L$, and substitute them into the first term of (2)). However, for instance, in the realistic regime where $h\sigma^2 B^2/\varepsilon^2 \geq \sqrt{\tau h L\sigma^2 B^4/\varepsilon^3} \geq h\sigma^2 B^2/n\varepsilon^2$ and $h\sigma^2 B^2/\varepsilon^2 \geq hLB^2/\varepsilon$ (i.e., $\varepsilon$ is small and $n$ is large), we have (2) $\simeq \sqrt{\tau h L\sigma^2 B^4/\varepsilon^3} + hLB^2/\varepsilon$, and the term $\sqrt{\tau h L\sigma^2 B^4/\varepsilon^3}$ can become arbitrarily larger than the first term $\tau LB^2/\varepsilon + h\left(LB^2/\varepsilon + \sigma^2 B^2/n\varepsilon^2\right)$ in the min of (3) when $\varepsilon$ is small and $n$ is large. This is formalized in Lemma 2.1.

A similar comparison is done for *nonconvex* functions in Section 2.2, where there is a gap between Local SGD and Minibatch SGD/Hero SGD even under $\rho$–weak convexity, and in the *heterogeneous*

*setting* even under $\rho$–weak convexity, as well as the first and second-order similarity assumptions. See Section 2.3.

♣ **Improving the Canonical Local SGD Method.** We began investigating the gap more closely to identify possible reasons and solutions to bridge it. It turns out that the problem arises from an incorrect aggregation scheme in Line 8 of Algorithm 1. Surprisingly, the correct update is $x^{t+1} = x^t - \frac{\eta_\ell}{\sqrt{n}} \sum_{i=1}^{n} \sum_{j=0}^{K-1} \nabla f(z_{i,j}^t; \xi_{i,j}^t)$, where we scale by $\sqrt{n}$ instead of $n$. We derive this update through the analysis of Local SGD with two step sizes, Dual Local SGD. While our work is not the first to consider Dual Local SGD in the literature, to the best of our knowledge, this is the first work to show that the canonical version of Local SGD (Algorithm 1) is suboptimal, and that a modification via Dual Local SGD, when combined with Hero SGD, leads to the optimal *time complexity* (9) (up to logarithmic factors).

♦ **A New Local SGD Method with Larger Local Step Sizes.** We noticed that Dual Local SGD can be improved and went further by increasing the local step sizes. We provide a new Local SGD method, called Decaying Local SGD. Instead of the local step size rule $\eta_\ell = \sqrt{n}\eta_g$ in Dual Local SGD, where $\eta_g$ is a global step size, we propose to use the step size rule $\eta_j = \sqrt{b/(j+1)(\log K+1)} \times \eta_g$, where $j$ is an index of the local step size iteration and $b$ is a parameter. This step size is never worse than $\eta_\ell = \sqrt{n}\eta_g$ (up to the logarithmic factor) and, in fact, can be arbitrarily larger.

♥ **Extension to Other Asynchronous and Local Approaches.** Using the insights from our theory of Local SGD methods, we extend them to other asynchronous and local methods, and improve the theory of Tyurin and Sivtsov (2025): in their framework, they use the step size rule $\eta_\ell = \eta_g$ in local updates, while we show that it is possible to take $\eta_j = \sqrt{b/(j+1)(\log K+1)} \times \eta_g$ instead, where $j$ is the "tree distance" in their context. See details in Sections F and G.

## 2 TIME COMPLEXITY ANALYSIS OF EXISTING METHODS

### 2.1 CONVEX SETUP

We start with the homogeneous and convex setting, where a lower bound for the iteration complexity of Local SGD has already been established (Glasgow et al., 2022) (Table 1). Using the lower bound on the iteration complexity, we can prove the following lower bound on the time complexity:

**Theorem 2.1** (Lower bound for Local SGD)**.** *Under Assumptions 1.1, 1.2, 1.3, and 1.4, the time complexity of* Local SGD *(Algorithm 1) to find an $\varepsilon$–solution* ***is not better than***

$$\Omega\left(\min\left\{\sqrt{\tau h\left(\frac{L\sigma^2 B^4}{\varepsilon^3}\right)} + h\left(\frac{LB^2}{\varepsilon} + \frac{\sigma^2 B^2}{n\varepsilon^2}\right), h\left(\frac{LB^2}{\varepsilon} + \frac{\sigma^2 B^2}{\varepsilon^2}\right)\right\}\right),\tag{2}$$

*for any choice of the input parameters , up to constant factors.*

For Minibatch SGD and Hero SGD, we can prove the theorem below, where (3) represents the best time complexity achieved by either method.

**Theorem 2.2** (Upper bound for Minibatch SGD/Hero SGD)**.** *Under Assumptions 1.1, 1.2, 1.3, and 1.4, the time complexity of* Minibatch SGD *and* Hero SGD *(Algorithms 2 and 3) to find an $\varepsilon$–solution* ***is no worse than***

$$O\left(\min\left\{\tau\frac{LB^2}{\varepsilon} + h\left(\frac{LB^2}{\varepsilon} + \frac{\sigma^2 B^2}{n\varepsilon^2}\right), h\left(\frac{LB^2}{\varepsilon} + \frac{\sigma^2 B^2}{\varepsilon^2}\right)\right\}\right),\tag{3}$$

*with a proper choice of the input parameters, up to constant factors.*

Both time complexities (2) and (3) depend on computation time $h$ and communication time $\tau$: the larger the values of $\tau$ and $h$, the more time it takes to converge to an $\varepsilon$–solution. However, the time complexity of Minibatch SGD/Hero SGD is never worse than that of Local SGD. This is formally estbalished below:

**Lemma 2.1** (Time complexity of Minibatch SGD/Hero SGD is never worse than that of Local SGD)**.** *Under Assumptions 1.1, 1.2, 1.3, and 1.4, let*

$$T_L := \Omega\left(\min\left\{\sqrt{\tau h\left(\frac{L\sigma^2 B^4}{\varepsilon^3}\right)} + h\left(\frac{LB^2}{\varepsilon} + \frac{\sigma^2 B^2}{n\varepsilon^2}\right), h\left(\frac{LB^2}{\varepsilon} + \frac{\sigma^2 B^2}{\varepsilon^2}\right)\right\}\right),$$

Table 2: Time complexities to get an $\varepsilon$–solution or an $\varepsilon$–stationary point in the homogeneous regime under Assumption 1.1. We use the same notations as in Table 1, plus $\tau =$ communication time; $h =$ computation time.

| Algorithm | Convex Setting | Nonconvex Setting |
|---|---|---|
| | **Time Complexity** | **Time Complexity** |
| Local SGD (Thm. 2.1 and Cor. 2.1) | $\min\left\{ \sqrt{\tau h\left(\frac{L\sigma^2 B^4}{\varepsilon^3}\right)} + h\left(\frac{LB^2}{\varepsilon} + \frac{\sigma^2 B^2}{n\varepsilon^2}\right),\; h\left(\frac{LB^2}{\varepsilon} + \frac{\sigma^2 B^2}{\varepsilon^2}\right) \right\}$ | $^5 \geq \sqrt{\tau h\left(\frac{L^2\sigma^2\Delta^2}{\varepsilon^3}\right)} + h\left(\frac{L\Delta}{\varepsilon} + \frac{L\sigma^2\Delta}{n\varepsilon^2}\right)$ |
| Hero SGD (Lan, 2020) | $h\left(\frac{LB^2}{\varepsilon} + \frac{\sigma^2 B^2}{\varepsilon^2}\right)$ | $h\left(\frac{L\Delta}{\varepsilon} + \frac{L\sigma^2\Delta}{\varepsilon^2}\right)$ |
| Minibatch SGD, Dual Local SGD, Decaying Local SGD, or Decaying Local SGD (async version) (Thm. 2.3, 3.1, 4.1, G.3, C.1, C.2) | $\tau\frac{LB^2}{\varepsilon} + h\left(\frac{LB^2}{\varepsilon} + \frac{\sigma^2 B^2}{n\varepsilon^2}\right)$ | $\tau\frac{L\Delta}{\varepsilon} + h\left(\frac{L\Delta}{\varepsilon} + \frac{L\sigma^2\Delta}{n\varepsilon^2}\right)$ |
| Accelerated Minibatch SGD // Accelerated Hero SGD (Lan, 2020; Tyurin and Richtárik, 2024) | $\min\left\{ \tau\frac{\sqrt{L}B}{\sqrt{\varepsilon}} + h\left(\frac{\sqrt{L}B}{\sqrt{\varepsilon}} + \frac{\sigma^2 B^2}{n\varepsilon^2}\right),\; h\left(\frac{\sqrt{L}B}{\sqrt{\varepsilon}} + \frac{\sigma^2 B^2}{\varepsilon^2}\right) \right\}$ | — |
| Lower Bound (matches the best of two previous lines in the nonconvex setting) | — | $\tilde{\Omega}\left( \min\left\{ \tau\frac{L\Delta}{\varepsilon}, h\frac{L\sigma^2\Delta}{\varepsilon^2} \right\} + h\left(\frac{L\Delta}{\varepsilon} + \frac{L\sigma^2\Delta}{n\varepsilon^2}\right) \right)$ (Tyurin and Richtárik, 2024) |

*denotes the lower bound on the time complexity of* Local SGD *and, for* $K = \max\left\{ \left\lceil \frac{\sigma^2}{L\varepsilon n} \right\rceil, 1 \right\}^4$ *let*

$$T_M := O\left( \min\left\{ \tau\frac{LB^2}{\varepsilon} + h\left(\frac{LB^2}{\varepsilon} + \frac{\sigma^2 B^2}{n\varepsilon^2}\right), h\left(\frac{LB^2}{\varepsilon} + \frac{\sigma^2 B^2}{\varepsilon^2}\right) \right\} \right),$$

*be an upper bound on the time complexity of* Minibatch SGD/Hero SGD. *Then, it holds*

$$T_M \lesssim T_L,$$

*i.e., the runtime of* Minibatch SGD/Hero SGD *is, up to constant factors, never worse than that of* Local SGD

Moreover, the time complexity of Local SGD can be arbitrarily larger. For instance, in the regime where $\frac{h\sigma^2}{n\varepsilon} \leq \tau L \leq \frac{h\sigma^2}{\varepsilon}$, we have

$$T_L = \Omega\left( \sqrt{\tau h\left(\frac{L\sigma^2 B^4}{\varepsilon^3}\right)} + h\frac{LB^2}{\varepsilon} \right) \quad \text{and} \quad T_M = O\left( \tau\frac{LB^2}{\varepsilon} + h\frac{LB^2}{\varepsilon} \right),$$

hence $\frac{T_L}{T_M} = \Omega\left( \frac{\sqrt{\tau h}}{\tau+h}\sqrt{\frac{\sigma^2}{L\varepsilon}} \right)$. Now, let $\tau, h > 0$ such that $\tau L = \frac{h\sigma^2}{n\varepsilon}$ then $\frac{T_L}{T_M} = \Omega\left( \frac{\sigma^2/\sqrt{n}L\varepsilon}{\sigma^2/nL\varepsilon+1} \right)$ which can be arbitrarily large when $n \gg 1$, $\varepsilon \ll 1$ and $\sigma^2/nL\varepsilon = \Theta(1)$.

## 2.2 NONCONVEX SETUP

Compared to the convex setting, where Glasgow et al. (2022) established a lower bound for the iteration complexity of Local SGD, to the best of our knowledge, there is no lower bound available in the nonconvex setting. Therefore, to analyze the time complexity in the nonconvex case, we rely on the state-of-the-art convergence rates provided by Koloskova et al. (2020); Luo et al. (2025).

**Corollary 2.1** (Upper bound for Local SGD). *Under Assumptions 1.1, 1.2, and 1.3, (and $\rho$–weak convexity[6]), the time complexity of* Local SGD *(Algorithm 1) to find an $\varepsilon$–stationary point is **not better than***

$$\Omega\left( \sqrt{\tau h\left(\frac{L^2\sigma^2\Delta^2}{\varepsilon^3}\right)} + h\left(\frac{L\Delta}{\varepsilon} + \frac{L\sigma^2\Delta}{n\varepsilon^2}\right) \right), \tag{4}$$

*up to constant factors, using the analysis by Koloskova et al. (2020); Luo et al. (2025).*

---

[4] See the proof of Theorem 2.2.

[6] Considering it does not help to improve the time complexity of Minibatch SGD/Hero SGD.

[6] Not better than the following complexity.

**Theorem 2.3** (Upper bound for Minibatch SGD/Hero SGD). *Under Assumptions 1.1, 1.2, and 1.3, the time complexity of* Minibatch SGD *and* Hero SGD *(Algorithms 2 and 3) to find an $\varepsilon$–stationary point **is no worse than***

$$O\left(\min\left\{\tau\frac{L\Delta}{\varepsilon} + h\left(\frac{L\Delta}{\varepsilon} + \frac{L\sigma^2\Delta}{n\varepsilon^2}\right), h\left(\frac{L\Delta}{\varepsilon} + \frac{L\sigma^2\Delta}{\varepsilon^2}\right)\right\}\right), \tag{5}$$

*up to constant factors, where we take $\eta \simeq \min\{1/L, \varepsilon n/L\sigma^2\}$ in* Hero SGD, *and $K = \max\left\{\lceil \sigma^2/\varepsilon n\rceil, 1\right\}$ and $\eta_g \simeq \min\left\{\varepsilon/L\sigma^2, 1/nL\right\}$. Moreover, it is optimal up to logarithmic factors due to a result of* Tyurin and Richtárik (2024).

The gap between Theorems 2.1 and 2.3 is similar to the gap between Theorems 2.1 and 2.2 with $L\Delta/\varepsilon$ and $L\sigma^2\Delta/\varepsilon$ instead of $LB^2/\varepsilon^2$ and $\sigma^2 B^2/\varepsilon^2$.

**In total, our results illustrate that Local SGD (Algorithm 1) is not only non-better, but also provably strictly worse than Minibatch SGD and Hero SGD (Algorithms 2 and 3)!**

### 2.3 HETEROGENEOUS SETTING

In the heterogeneous setting, we require the assumption below:

**Assumption 2.1.** *The stochastic gradients satisfy $\mathbb{E}_\xi[\nabla f_i(x;\xi)] = \nabla f_i(x)$ (unbiasedness) and $\mathbb{E}_\xi[\|\nabla f_i(x;\xi) - \nabla f_i(x)\|^2] \leq \sigma^2$ for all $x \in \mathbb{R}^d$, where $\sigma^2 \geq 0$ (bounded variance). For all $i \in [n]$, worker $i$ can only calculate $\nabla f_i(\cdot;\cdot)$.*

Similarly to Theorems 2.1 and 2.3, which work in the homogeneous regime, we can prove the following theorems in the heterogeneous setting.

**Corollary 2.2** (Upper bound for Local SGD). *Assume that all functions $f_i$ satisfy Assumptions 1.1. Under Assumptions 1.2 and 2.1, (and $\rho$–weak convexity, the first and the second-order similarity), the time complexity of* Local SGD *and* SCAFFOLD *to find an $\varepsilon$–stationary point is **not better than***

$$\Omega\left(\sqrt{\tau h\left(\frac{L^2\sigma^2\Delta^2}{\varepsilon^3}\right)} + h\left(\frac{L\Delta}{\varepsilon} + \frac{L\sigma^2\Delta}{n\varepsilon^2}\right)\right), \tag{6}$$

*up to constant factors, using the analysis by* Koloskova et al. (2020); Luo et al. (2025) *(best known in terms of scaling with the number of local steps $K$).*

**Theorem 2.4** (Upper bound for Minibatch SGD). *Under Assumptions 1.1, 1.2, and 2.1, the time complexity of* Minibatch SGD *(Algorithms 2) to find an $\varepsilon$–stationary point **is no worse than***

$$O\left(\tau\frac{L\Delta}{\varepsilon} + h\left(\frac{L\Delta}{\varepsilon} + \frac{L\sigma^2\Delta}{n\varepsilon^2}\right)\right), \tag{7}$$

*up to constant factors, where we take $K = \max\left\{\lceil \sigma^2/\varepsilon n\rceil, 1\right\}$ and $\eta_g \simeq \min\left\{\varepsilon/L\sigma^2, 1/nL\right\}$. Moreover, it is optimal up to constant factors due to a result of* Tyurin and Richtárik (2024).

Even under additional assumptions, the current state-of-the-art methods, Local SGD and SCAFFOLD, are worse than the optimal Minibatch SGD method in the regime when $\varepsilon$ is small and $n$ is large[7]. Moreover, under Assumptions 1.1, 1.2, and 2.1, they cannot be better due to the optimality of Minibatch SGD. A natural question is whether we can design a more practical method with local steps in the nonconvex stochastic heterogeneous setting that at least matches (7) under Assumptions 1.1, 1.2, and 2.1, or under slightly stronger assumptions. There is such a method, namely SCAFFOLD by Karimireddy et al. (2020), which yields the iteration complexity $L_{\max}\Delta/\varepsilon + L_{\max}\sigma^2\Delta/nK\varepsilon^2$ and the time complexity $\tau L_{\max}\Delta/\varepsilon + h\left(L_{\max}\Delta/\varepsilon + L_{\max}\sigma^2\Delta/n\varepsilon^2\right)$, where $L_{\max}$ is the largest smoothness constant among the functions $f_i$. Thus, SCAFFOLD by Karimireddy et al. (2020) almost matches Minibatch SGD, but the time complexity of SCAFFOLD can still be $L_{\max}/L$ times larger. Besides, collaboration can provide benefits beyond variance reduction, as Patel et al. (2024) show that in mildly heterogeneous settings, where local updates provably help clients explore the loss landscape.

---

[7]For instance, up to $\varepsilon$, (6) $\simeq 1/\varepsilon^{3/2}$ and (7) $\simeq 1/\varepsilon$ when $n$ is large enough

---

**Algorithm 4:** Dual Local SGD (Local SGD with two step-sizes: global and local)

---

**Require:** initial point $x^0$, global step size $\eta_g$, local step size $\eta_\ell$, communication rounds $R$, number of local steps $K$

1: **for** $t = 0, 1, \ldots, R - 1$ **do**
2:    $z_{i,0}^t = x^t$
3:    **for** worker $i \in \{1, \ldots, n\}$ **in parallel do**
4:       **for** $j = 0, \ldots, K - 1$ **do**
5:          $z_{i,j+1}^t = z_{i,j}^t - \eta_\ell \nabla f(z_{i,j}^t; \xi_{i,j}^t), \xi_{i,j}^t \sim \mathcal{D}$
6:       **end for**
7:    **end for**
8:    $x^{t+1} = x^t - \eta_g \sum\limits_{i=1}^{n} \sum\limits_{j=0}^{K-1} \nabla f(z_{i,j}^t; \xi_{i,j}^t)$
9: **end for**

---

## 2.4 ACCELERATED CONVEX OPTIMIZATION

The same question arises with accelerated methods in convex optimization in both homogeneous and heterogeneous setups. Under Assumptions 1.1, 1.4, 1.2, and 1.3, the state-of-the-art time complexity is $\Theta\left(\min\{\tau\sqrt{L}B/\sqrt{\varepsilon} + h(\sqrt{L}B/\sqrt{\varepsilon} + \sigma^2 B^2/n\varepsilon^2), h(\sqrt{L}B/\sqrt{\varepsilon} + \sigma^2 B^2/\varepsilon^2)\}\right)$ which is minimax-optimal (Woodworth et al., 2021), and is achieved by the best of Accelerated Minibatch SGD and Accelerated Hero SGD (Lan, 2020; Tyurin and Richtárik, 2024). In the heterogeneous setting, under Assumptions 1.1, 1.2, 2.1, and 1.4, the state-of-the-art time complexity $\Theta\left(\min\{\tau\sqrt{L}B/\sqrt{\varepsilon} + h(\sqrt{L}B/\sqrt{\varepsilon} + \sigma^2 B^2/n\varepsilon^2)\}\right)$ is achieved by Accelerated Minibatch SGD (Lan, 2020; Tyurin and Richtárik, 2024). There have been several attempts to accelerate algorithms with local steps, taking into account both computational and communication complexities, including (Mishchenko et al., 2022; Malinovsky et al., 2022). While the communication complexities are accelerated in these approaches, the variance $\sigma^2$ does not decrease with the number of local steps, and the resulting computational complexity remains non-accelerated. To the best of our knowledge, no existing method achieves the time complexity of Accelerated Minibatch SGD while using local steps.

## 3 IMPROVED VERSIONS OF Local SGD

We now return back to nonconvex homogeneous optimization. Notice that (5) is optimal up to logarithmic factors; thus, there is no hope of designing an alternative Local SGD method that achieves a better time complexity in the nonconvex setting. Nevertheless, since the canonical version of Local SGD is suboptimal and does not match the optimal time complexity (5), we began investigating the possibility of modifying Local SGD to make it optimal as well, in order to obtain a complete picture.

Our starting point is Minibatch SGD (Algorithm 2), as it is optimal up to logarithmic factors in the nonconvex setting when $h\sigma^2/\varepsilon \geq \tau L$. A straightforward observation is that Minibatch SGD can be viewed as a Local SGD method in which the local step sizes are set to zero. Therefore, instead of Minibatch SGD (Algorithm 2), we consider Dual Local SGD (Algorithm 4), Local SGD with two step sizes, which reduces to Minibatch SGD when $\eta_\ell = 0$ and $\eta_g = \min\{\varepsilon/L\sigma^2, 1/nL\}$. Meanwhile, Dual Local SGD (Algorithm 4) reduces to the *suboptimal* Local SGD (Algorithm 1) when $\eta_g = \eta_\ell/n$.

We now analyze Algorithm 4 and present our new time complexity guarantees. We will show that it is possible to match the time complexity of Minibatch SGD by using the global step size $\eta_g = \min\{\varepsilon/L\sigma^2, 1/nL\}$, the same as in Minibatch SGD, but with a non-zero local step size $\eta_\ell$.

**Theorem 3.1** (Upper bound for Dual Local SGD). *Under Assumptions 1.2 and 1.3, Dual Local SGD (Algorithm 4) with $\eta_g = \min\left\{\frac{\varepsilon}{8L\sigma^2}, \frac{1}{4nL}\right\}$ and $\eta_\ell \leq \sqrt{n}\eta_g$ finds an $\varepsilon$-stationary point after at most $R = \lceil 32L\Delta/\varepsilon \rceil$ communication rounds with $K = \max\left\{\lceil \sigma^2/\varepsilon n \rceil, 1\right\}$. Additionally, under Assumption 1.1, it requires at most*

$$\tau \frac{64L\Delta}{\varepsilon} + 64h\left(\frac{L\Delta}{\varepsilon} + \frac{L\sigma^2\Delta}{n\varepsilon^2}\right) \tag{8}$$

*seconds to find an $\varepsilon$−stationary point. Moreover, when combined[8] with* Hero SGD *(Algorithms 3), the time complexity is no worse than*

$$O\left(\min\left\{\tau\frac{L\Delta}{\varepsilon} + h\left(\frac{L\Delta}{\varepsilon} + \frac{L\sigma^2\Delta}{n\varepsilon^2}\right), h\left(\frac{L\Delta}{\varepsilon} + \frac{L\sigma^2\Delta}{\varepsilon^2}\right)\right\}\right), \quad (9)$$

*up to constant factors, where we take $\eta \simeq \min\{1/L, \varepsilon n/L\sigma^2\}$ in* Hero SGD, *and (9) is also optimal up to logarithmic factors due to the result by Tyurin and Richtárik (2024).*

### 3.1 DISCUSSION AND COMPARISON WITH PREVIOUS WORK

An interesting observation regarding the choice of parameters is that we can take $\eta_\ell = \sqrt{n} \times \eta_g$, which is equivalent to $\eta_g = \eta_\ell/\sqrt{n}$. Substituting this choice into Dual Local SGD (Algorithm 4) yields precisely the canonical Local SGD method (Algorithm 1), but with an important modification: we should run the update $x^{t+1} = x^t - \eta_\ell/\sqrt{n} \sum_{i=1}^n \sum_{j=0}^{K-1} \nabla f(z_{i,j}^t; \xi_{i,j}^t)$ instead of Line 8 of Algorithm 1. Remarkably, the right scaling is $\sqrt{n}$ instead of $n$. We should admit that this is not the first time a modification of the Local SGD method has achieved the time complexity $\tau L\Delta/\varepsilon + h\left(L\Delta/\varepsilon + L\sigma^2\Delta/n\varepsilon^2\right)$ and obtained the optimal time complexity in the regime where $h\sigma^2/\varepsilon \geq \tau L$ (up to logarithmic factors). A recent work by Tyurin and Sivtsov (2025), which analyzes different asynchronous and parallel methods, also proposes an alternative version of Local SGD with one step size and matching complexity. However, while their global update is the same, their local updates are smaller by a factor of $\sqrt{n}$. Thus, our new theory captures a more practical version of Local SGD. In Section 4, we prove that the local step size can be increased even further.

This is not the first work to explore Local SGD with two step sizes. Many previous papers have studied this direction, including (Charles and Konečný, 2020; Karimireddy et al., 2020; Yang et al., 2021; 2022; Malinovsky et al., 2023; Jhunjhunwala et al., 2023; Huang et al., 2023). Yang et al. (2021) and Karimireddy et al. (2020) analyze methods similar to Algorithm 4. However, Yang et al. (2021) choose a different pair of step sizes that do not necessarily lead to the complexity (8). Moreover, Karimireddy et al. (2020) did not prove that their algorithm achieves the optimal complexity. Although many of these prior works implicitly employ the $1/\sqrt{n}$ scaling (notably by selecting stepsizes of order $\eta = \Theta(\sqrt{n})$ in conjunction with a $1/n$ averaging factor (Khaled et al., 2020; Woodworth et al., 2020)), these methods typically *couple* the local and global stepsizes. As a consequence, the respective contributions of the local and global updates to the overall convergence behaviour remain obscured. By explicitly decoupling $\eta_\ell$ and $\eta_g$, our analysis isolates their individual effects and this perspective also makes transparent that the canonical implementation of Local SGD effectively adopts a suboptimal $1/n$ scaling. Besides, this decoupling allows us to recover Minibatch SGD in a principled manner as the limiting case $\eta_\ell = 0$, which is not the case of most prior works (e.g., SCAFFOLD in Karimireddy et al. (2020)). Lastly, the work by Glasgow et al. (2022), which established the tight lower bound for Algorithm 1, left open the question of whether Dual Local SGD can provably improve upon Algorithm 1. Our work provides an affirmative answer to this question.

To the best of our knowledge, ours is the first work to show that the canonical version of Local SGD (Algorithm 1) is suboptimal, and that a modification via Dual Local SGD, when combined with Hero SGD, leads to optimal performance (9) (up to logarithmic factors). Moreover, in Section C.2.2, we provide an analysis of (non-accelerated) Dual Local SGD for completeness.

## 4 TOWARDS EVEN LARGER AND ADAPTIVE LOCAL STEP SIZES

While Dual Local SGD and Hero SGD indeed achieve the optimal time complexity (up to logarithmic factors), comparing Theorem 2.3 and Theorem 3.1 shows that the same complexity can also be achieved with Minibatch SGD, a method that does not perform any local steps. Nevertheless, it has often been observed that Local SGD outperforms Minibatch SGD because, intuitively, it explores the optimization landscape more effectively through its local steps (McMahan et al., 2017b). Notice that Minibatch SGD runs local steps with $\eta_\ell = 0$, and Dual Local SGD with $\eta_\ell = \sqrt{n}\eta_g$ in Theorem 3.1. Can we increase $\eta_\ell$ further? In order to answer the question, we consider Decaying Local SGD (Algorithm 5) and provide the following theorem.

---

[8]As we are considering time complexities, by "combined", we mean that both methods (Local SGD and Hero SGD) are run in parallel from the same starting point, and the result of whichever finishes first is returned.

---

**Algorithm 5:** Decaying Local SGD (Local SGD with a global step and decaying local steps)

---

**Require:** initial point $x^0$, step size $\eta_g$, parameter $b$, rounds $R$, number of local steps $K$

1: **for** $t = 0, 1, \ldots, R - 1$ **do**
2:     $z_{i,0}^t = x^t$
3:     **for** worker $i \in \{1, \ldots, n\}$ **in parallel do**
4:         **for** $j = 0, \ldots, K - 1$ **do**
5:             $z_{i,j+1}^t = z_{i,j}^t - \eta_j \nabla f(z_{i,j}^t; \xi_{i,j}^t),\ \xi_{i,j}^t \sim \mathcal{D}$, where $\eta_j = \sqrt{\frac{b}{(j+1)(\log K+1)}} \times \eta_g$
6:         **end for**
7:     **end for**
8:     $x^{t+1} = x^t - \eta_g \sum_{i=1}^{n} \sum_{j=0}^{K-1} \nabla f(z_{i,j}^t; \xi_{i,j}^t)$
9: **end for**

---

**Theorem 4.1** (Upper bound for Decaying Local SGD). *Under Assumptions 1.2 and 1.3, Decaying Local SGD (Algorithm 5) with $\eta_g = \min\{\frac{\varepsilon}{8L\sigma^2}, \frac{1}{4nL}\}$ and $b = \max\{\frac{\sigma^2}{\varepsilon}, n\}$ finds an $\varepsilon$–stationary point after at most $R = \lceil 32L\Delta/\varepsilon \rceil$ communication rounds with $K = \max\{\lceil \sigma^2/\varepsilon n \rceil, 1\}$, and under Assumption 1.1, it requires at most $\tau \frac{64L\Delta}{\varepsilon} + 64h\left(\frac{L\Delta}{\varepsilon} + \frac{L\sigma^2\Delta}{n\varepsilon^2}\right)$ sec. to find an $\varepsilon$–stationary point.*

Theorem 4.1 guarantees the same time complexity as Theorem 3.1. However, up to a logarithmic factor, our choice of local step sizes is larger. Indeed, instead of $\eta_\ell = \sqrt{n}\eta_g$, we take $\eta_j = \sqrt{\frac{b}{(j+1)(\log K+1)}}\eta_g \geq \sqrt{\frac{n}{(\log K+1)}}\eta_g = \tilde{\Theta}(\sqrt{n}\eta_g)$, where $j$ is the index of the local iteration and $j < K \simeq \max\{\sigma^2/\varepsilon n, 1\} = b/n$. Up to a factor of $\log K$, the new step size rule is never worse than $\sqrt{n}\eta_g$. However, especially in the first local iterations, our new step size rule can be significantly larger by a factor of $\tilde{\Theta}(\sqrt{b/(j+1)n})$. If $j \approx 1$ and $\varepsilon$ is small, then the increase is $\tilde{\Theta}(\sqrt{\sigma^2/n\varepsilon})$. The factor $\log K$ is a minor price for the adaptivity. We obtain a similar result in the convex setting; see Section C.2.2. Beyond the decreasing scheme in line 5 of Algorithm 5, our analysis applies to any local stepsize sequence $\{\eta_\ell\}_{0 \leq \ell \leq K-1}$ satisfying $\sum_{\ell=0}^{K-1} \eta_\ell^2 \lesssim \eta_g^2 b$, achieving the same iteration and time complexity as $\eta_\ell = \sqrt{n}\,\eta_g$. This includes constant, decreasing, or oscillating schedules, providing more flexible local updates than traditional Local SGD. While our approach does not capture all practical behaviors of these schedules, it does identify a unified set of constraints under which a wide range of stepsize schemes attain the same iteration and time complexity guarantees, up to constant factors

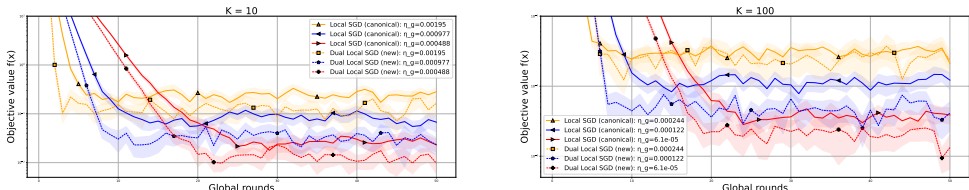

Figure 1: Experiments on the toy adversarial problem from (Glasgow et al., 2022).

## 5 NUMERICAL EXPERIMENTS

Before we present our numerical experiments, we want to stress that the main goal of this paper is to explain that the previous *theoretical* comparison in Table 1 might be misleading, and a better one is presented in Table 2, where we prove the *theoretical* suboptimality of the canonical Local SGD method. At the same time, the canonical Local SGD method (FedAvg) remains one of the most widely evaluated and tested algorithms in distributed and federated learning, and there is no doubt that it is a strong method for *practical* optimization tasks. Our experiments confirm this when comparing it to Dual and Decaying Local SGD; nevertheless, we also find that Dual and Decaying Local SGD can achieve superior performance.

**Toy example**. Our first experiment focuses on the special function $f : \mathbb{R} \to \mathbb{R}$ defined as $f(x) = x^2/2$ if $x \geq 0$, and $f(x) = x^2/4$ if $x < 0$, with $\nabla f(x; \xi) = \nabla f(x) + \xi$, where $\xi \sim \mathcal{N}(0, \sigma)$. This function is an adversarial problem for Algorithm 1 (Glasgow et al., 2022). Taking the starting point $x^0 = -30$, noise level $\sigma = 10$, and number of workers $n = 100$, we compare Algorithm 1 with our results in Theorems 3.1 and C.1. To obtain a fair comparison, we tune $\eta_g \in \{2^{-i} \mid i \in \{1, \ldots, 16\}\}$ in Algorithm 4 and set $\eta_\ell = n \times \eta_g$ to recover Algorithm 1, and $\eta_\ell = \sqrt{n} \times \eta_g$ to obtain the results from Theorems 3.1 and C.1. To ensure robustness, we run each experiment 30 times and plot 90% confidence intervals. We also verify the methods with different numbers of local steps: $K = 10$ and $K = 100$. In Figure 1, we plot the convergence rates of the algorithms for different values of $\eta_g$. The smaller the $\eta_g$, the lower the plot converges, which is theoretically expected since $\eta_g$ controls the size of the neighborhood in which the algorithm oscillates. However, for a fixed $\eta_g$, Dual Local SGD with our local step size choice converges to the corresponding neighborhood faster than Local SGD with its local step size rule.

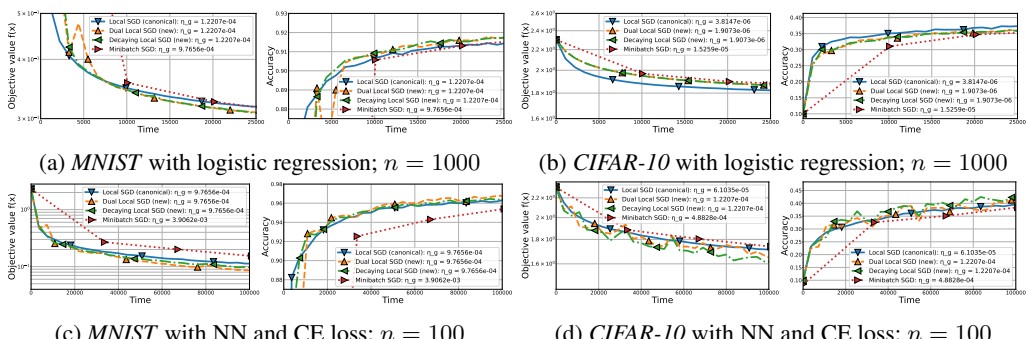

(a) *MNIST* with logistic regression; $n = 1000$     (b) *CIFAR-10* with logistic regression; $n = 1000$

(c) *MNIST* with NN and CE loss; $n = 100$     (d) *CIFAR-10* with NN and CE loss; $n = 100$

Figure 2: Experiments on practical machine learning problems.

**Practical machine learning problems**. We compare the methods on image recognition tasks using *MNIST* (LeCun et al., 2010) and *CIFAR-10* (Krizhevsky et al., 2009). Following the same setup as in the previous experiment, we take $n = 1000$ workers, fix $K = 10$, tune $\eta_g \in \{2^i \mid i \in \{-20, \ldots, 1\}\}$ to ensure a fair comparison, and plot the best corresponding curve. We consider the homogeneous setup, where each worker has access to the same dataset locally, and when it computes a stochastic gradient, it samples one data point uniformly from the dataset. In Figures 2a and 2b, we consider the standard logistic regression problem and observe that the canonical Local SGD method, Dual Local SGD, and Decaying Local SGD achieve comparable performance. In all plots, we report function value or accuracy as a function of *wall-clock time*. To reflect a communication-limited regime, we set the computation cost to $h = 1$ second and the communication delay to $\tau = 100$ seconds. As expected, Minibatch SGD is consistently slower in this setting: since it synchronizes at every iteration, the algorithm incurs the full communication delay $\tau$ at each step, which dominates its runtime and leads to significantly slower progress compared with methods that perform multiple local updates between communications. Besides, on *MNIST*, Dual and Decaying Local SGD achieve higher performance, whereas on *CIFAR-10* Algorithm 1 performs slightly better. We observe that Dual and Decaying Local SGD perform better on both datasets for the problem with a two-layer neural network (NN), Linear(input_dim, 32) $\to$ ReLU $\to$ Linear(32, num_classes), and the cross-entropy (CE) loss (see Figures 2c and 2d). Dual and Decaying Local SGD enjoy stronger theoretical guarantees, making them more robust to adversarial functions, as we can see in the adversarial example and in practical machine learning problems. Nevertheless, on practical "average" problems, the performance of all algorithms is very similar, and, consistent with numerous previous experiments, the canonical Local SGD performs well.

## 6    CONCLUSION

In this work, we show that the canonical Local SGD method is suboptimal and propose new methods that close the gap to the lower bound in the nonconvex setting. We extend our insights to other local and asynchronous methods. While our work shows that the new versions of Local SGD are optimal (up to logarithmic factors), it does not establish that they are strongly better (which can not be done due to the lower bounds). Our findings reopen the question of whether local steps can improve the *time complexity* of Minibatch SGD/Hero SGD.

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

CONTENTS

## A  NOTATION

| Asymptotic | Meaning |
|---|---|
| $g = \mathrm{O}(f)$ | There exists $C > 0$ such that $g \leq Cf$ for input parameters |
| $g = \Omega(f)$ | There exists $c > 0$ such that $g \geq cf$ for all input parameters |
| $g = \Theta(f)$ | When both $g = \mathrm{O}(f)$ and $g = \Omega(f)$ |
| $\tilde{\Theta}, \tilde{\Omega}, \tilde{\Theta}$ | The same as $\Theta, \Omega, \Theta$, but up to (hidden) logarithmic factors |
| $g \simeq f$ | When $g = f$ up to a positive universal constant |
| **Sets and intervals** | **Meaning** |
| $\mathbb{N}_0, \mathbb{N}$ | The set of non-negative (left) and positive (right) integers |
| $[a..b]\ (a, b \in \mathbb{N}_0)$ | The set $[a, b] = \{a, a+1, \ldots, b-1, b\}$ |
| $[n]\ (n \in \mathbb{N})$ | The set $\{1, 2, \ldots, n\}$ |
|  | **Meaning** |
| $n$ | The number of distinct workers |
| $R$ | The number of communication rounds |
| $K$ | The number of local steps performed by each worker |
| **Symbol** | **Meaning** |
| $\|\cdot\|, \langle\cdot, \cdot\rangle$ | The standard Euclidean norm and dot product |
| $\mathbb{P}(\cdot), \mathbb{P}(\cdot|\cdot)$ | Probability and conditional probability symbols |
| $\mathbb{E}[\cdot], \mathbb{E}[\cdot \mid \cdot]$ | Expectation and conditional expectation symbols |

## B  TIME COMPLEXITIES OF Local SGD AND Minibatch SGD

**Theorem 2.1** (Lower bound for Local SGD). *Under Assumptions 1.1, 1.2, 1.3, and 1.4, the time complexity of* Local SGD *(Algorithm 1) to find an $\varepsilon$–solution **is not better than***

$$\Omega\left(\min\left\{\sqrt{\tau h\left(\frac{L\sigma^2 B^4}{\varepsilon^3}\right)} + h\left(\frac{LB^2}{\varepsilon} + \frac{\sigma^2 B^2}{n\varepsilon^2}\right), h\left(\frac{LB^2}{\varepsilon} + \frac{\sigma^2 B^2}{\varepsilon^2}\right)\right\}\right), \tag{2}$$

*for any choice of the input parameters , up to constant factors.*

*Proof.* The lower bound iteration complexity of Local SGD to find an $\varepsilon$–solution (Glasgow et al., 2022) is

$$\min\left\{\frac{LB^2}{K\varepsilon} + \frac{\sigma^2 B^2}{nK\varepsilon^2} + \frac{L^{\frac{1}{2}}\sigma B^2}{K^{\frac{1}{2}}\varepsilon^{\frac{3}{2}}}, \frac{LB^2}{K\varepsilon} + \frac{\sigma^2 B^2}{K\varepsilon^2}\right\}$$

. Under Assumption 1.1, the time complexity to find an $\varepsilon$–solution is

$$T_{\mathrm{L}} := \tau \min\left\{\frac{LB^2}{K\varepsilon} + \frac{\sigma^2 B^2}{nK\varepsilon^2} + \frac{L^{\frac{1}{2}}\sigma B^2}{K^{\frac{1}{2}}\varepsilon^{\frac{3}{2}}}, \frac{LB^2}{K\varepsilon} + \frac{\sigma^2 B^2}{K\varepsilon^2}\right\}$$

$$+ hK \min\left\{\frac{LB^2}{K\varepsilon} + \frac{\sigma^2 B^2}{nK\varepsilon^2} + \frac{L^{\frac{1}{2}}\sigma B^2}{K^{\frac{1}{2}}\varepsilon^{\frac{3}{2}}}, \frac{LB^2}{K\varepsilon} + \frac{\sigma^2 B^2}{K\varepsilon^2}\right\},$$

up to constant factors, where the first bracket is the communication complexity (one communication takes $\tau$ seconds), and the second bracket is the computational complexity ($K$ computations of

stochastic gradients take $h \times K$ seconds in one iteration of each worker). Thus,

$$T_{\mathrm{L}} = \min \left\{ \tau \left( \frac{LB^2}{K\varepsilon} + \frac{\sigma^2 B^2}{nK\varepsilon^2} + \frac{L^{\frac{1}{2}}\sigma B^2}{K^{\frac{1}{2}}\varepsilon^{\frac{3}{2}}} \right) + h \left( \frac{LB^2}{\varepsilon} + \frac{\sigma^2 B^2}{n\varepsilon^2} + \frac{K^{\frac{1}{2}} L^{\frac{1}{2}} \sigma B^2}{\varepsilon^{\frac{3}{2}}} \right), \right.$$
$$\left. \tau \left( \frac{LB^2}{K\varepsilon} + \frac{\sigma^2 B^2}{K\varepsilon^2} \right) + h \left( \frac{LB^2}{\varepsilon} + \frac{\sigma^2 B^2}{\varepsilon^2} \right) \right\} \tag{10}$$

For all $K \geq 0$,

$$T_{\mathrm{L}} \geq \min \left\{ \tau \left( \frac{L^{\frac{1}{2}}\sigma B^2}{K^{\frac{1}{2}}\varepsilon^{\frac{3}{2}}} \right) + h \left( \frac{LB^2}{\varepsilon} + \frac{\sigma^2 B^2}{n\varepsilon^2} + \frac{K^{\frac{1}{2}} L^{\frac{1}{2}} \sigma B^2}{\varepsilon^{\frac{3}{2}}} \right), h \left( \frac{LB^2}{\varepsilon} + \frac{\sigma^2 B^2}{\varepsilon^2} \right) \right\}$$
$$\geq \min \left\{ 2\sqrt{\tau h \left( \frac{L\sigma^2 B^4}{\varepsilon^3} \right)} + h \left( \frac{LB^2}{\varepsilon} + \frac{\sigma^2 B^2}{n\varepsilon^2} \right), h \left( \frac{LB^2}{\varepsilon} + \frac{\sigma^2 B^2}{\varepsilon^2} \right) \right\},$$

where we ignore non-negative terms and use the AM-GM inequality. $\qquad \square$

**Theorem 2.2** (Upper bound for Minibatch SGD/Hero SGD). *Under Assumptions 1.1, 1.2, 1.3, and 1.4, the time complexity of* Minibatch SGD *and* Hero SGD *(Algorithms 2 and 3) to find an $\varepsilon$–solution **is no worse than***

$$O \left( \min \left\{ \tau \frac{LB^2}{\varepsilon} + h \left( \frac{LB^2}{\varepsilon} + \frac{\sigma^2 B^2}{n\varepsilon^2} \right), h \left( \frac{LB^2}{\varepsilon} + \frac{\sigma^2 B^2}{\varepsilon^2} \right) \right\} \right), \tag{3}$$

*with a proper choice of the input parameters, up to constant factors.*

*Proof.* The second term in $\min$ comes from the classical analysis of SGD (Lan, 2020) and the fact that it takes $h$ seconds to calculate one stochastic gradient. The first term comes from Minibatch SGD, which needs at most

$$\tau \left( \frac{LB^2}{\varepsilon} + \frac{\sigma^2 B^2}{Kn\varepsilon^2} \right) + hK \left( \frac{LB^2}{\varepsilon} + \frac{\sigma^2 B^2}{Kn\varepsilon^2} \right)$$
$$= \tau \left( \frac{LB^2}{\varepsilon} + \frac{\sigma^2 B^2}{Kn\varepsilon^2} \right) + h \left( \frac{KLB^2}{\varepsilon} + \frac{\sigma^2 B^2}{n\varepsilon^2} \right)$$

seconds. Taking $K = \max \left\{ \left\lceil \frac{\sigma^2}{L\varepsilon n} \right\rceil, 1 \right\}$, we obtain the first term in $\min$. $\qquad \square$

**Lemma 2.1** (Time complexity of Minibatch SGD/Hero SGD is never worse than that of Local SGD). *Under Assumptions 1.1, 1.2, 1.3, and 1.4, let*

$$T_L := \Omega \left( \min \left\{ \sqrt{\tau h \left( \frac{L\sigma^2 B^4}{\varepsilon^3} \right)} + h \left( \frac{LB^2}{\varepsilon} + \frac{\sigma^2 B^2}{n\varepsilon^2} \right), h \left( \frac{LB^2}{\varepsilon} + \frac{\sigma^2 B^2}{\varepsilon^2} \right) \right\} \right),$$

*denotes the lower bound on the time complexity of* Local SGD *and, for $K = \max \left\{ \left\lceil \frac{\sigma^2}{L\varepsilon n} \right\rceil, 1 \right\}$[9] let*

$$T_M := O \left( \min \left\{ \tau \frac{LB^2}{\varepsilon} + h \left( \frac{LB^2}{\varepsilon} + \frac{\sigma^2 B^2}{n\varepsilon^2} \right), h \left( \frac{LB^2}{\varepsilon} + \frac{\sigma^2 B^2}{\varepsilon^2} \right) \right\} \right),$$

*be an upper bound on the time complexity of* Minibatch SGD/Hero SGD. *Then, it holds*

$$T_M \lesssim T_L,$$

*i.e., the runtime of* Minibatch SGD/Hero SGD *is, up to constant factors, never worse than that of* Local SGD

*Proof.* We distinguish between the regimes $h\sigma^2/\varepsilon \geq \tau L$ and $h\sigma^2/\varepsilon < \tau L$:

---

[9]See the proof of Theorem 2.2.

- if $h\sigma^2/\varepsilon \geq \tau L$ then, we have

$$\sqrt{\tau h \left( \frac{L\sigma^2 B^4}{\varepsilon^3} \right)} \geq \sqrt{\tau^2 \left( \frac{L^2 B^4}{\varepsilon^2} \right)} = \tau \frac{LB^2}{\varepsilon},$$

which, when summed together with $h \left( \frac{LB^2}{\varepsilon} + \frac{\sigma^2 B^2}{n\varepsilon^2} \right)$ gives the first term in the $\min\{\ldots\}$ of $T_M$ (up ot constant factors) hence, it holds that $T_M \lesssim T_L$ in this regime.

- Otherwise, if $h\sigma^2/\varepsilon < \tau L$, then

$$\sqrt{\tau h \left( \frac{L\sigma^2 B^4}{\varepsilon^3} \right)} \geq \sqrt{h^2 \left( \frac{L^2 \sigma^4 B^4}{\varepsilon^4} \right)} = h \frac{\sigma^2 B^2}{\varepsilon^2},$$

so $T_L = \Omega \left( h \left( \frac{LB^2}{\varepsilon} + \frac{\sigma^2 B^2}{\varepsilon^2} \right) \right) \gtrsim T_M$, as claimed.

Finally, this proves that in all possible regimes, we have $T_M \lesssim T_L$. □

**Corollary 2.1** (Upper bound for Local SGD). *Under Assumptions 1.1, 1.2, and 1.3, (and $\rho$–weak convexity[10]), the time complexity of Local SGD (Algorithm 1) to find an $\varepsilon$–stationary point is **not better than***

$$\Omega \left( \sqrt{\tau h \left( \frac{L^2 \sigma^2 \Delta^2}{\varepsilon^3} \right)} + h \left( \frac{L\Delta}{\varepsilon} + \frac{L\sigma^2 \Delta}{n\varepsilon^2} \right) \right), \tag{4}$$

*up to constant factors, using the analysis by Koloskova et al. (2020); Luo et al. (2025).*

*Proof.* The proof is similar to the proof of Theorem 2.1. Using the result by Koloskova et al. (2020); Luo et al. (2025) (see Table 1), we get the time complexity to find an $\varepsilon$–stationary point at least equal to

$$\tau \left( \frac{L\Delta}{\varepsilon K} + \frac{L\sigma^2 \Delta}{nK\varepsilon^2} + \frac{L\sigma\Delta}{K^{\frac{1}{2}}\varepsilon^{\frac{3}{2}}} \right) + hK \left( \frac{L\Delta}{\varepsilon K} + \frac{L\sigma^2 \Delta}{nK\varepsilon^2} + \frac{L\sigma\Delta}{K^{\frac{1}{2}}\varepsilon^{\frac{3}{2}}} \right)$$

$$\geq \tau \left( \frac{L\sigma\Delta}{K^{\frac{1}{2}}\varepsilon^{\frac{3}{2}}} \right) + h \left( \frac{L\Delta}{\varepsilon} + \frac{L\sigma^2 \Delta}{n\varepsilon^2} + \frac{K^{\frac{1}{2}}L\sigma\Delta}{\varepsilon^{\frac{3}{2}}} \right)$$

(where we ignored the term with $\rho$), which can be lower bounded by

$$2\sqrt{\tau h \frac{L^2 \sigma^2 \Delta^2}{\varepsilon^3}} + h \left( \frac{L\Delta}{\varepsilon} + \frac{L\sigma^2 \Delta}{n\varepsilon^2} \right)$$

for all $K > 0$. □

**Theorem 2.3** (Upper bound for Minibatch SGD/Hero SGD). *Under Assumptions 1.1, 1.2, and 1.3, the time complexity of Minibatch SGD and Hero SGD (Algorithms 2 and 3) to find an $\varepsilon$–stationary point is **no worse than***

$$O \left( \min \left\{ \tau \frac{L\Delta}{\varepsilon} + h \left( \frac{L\Delta}{\varepsilon} + \frac{L\sigma^2 \Delta}{n\varepsilon^2} \right), h \left( \frac{L\Delta}{\varepsilon} + \frac{L\sigma^2 \Delta}{\varepsilon^2} \right) \right\} \right), \tag{5}$$

*up to constant factors, where we take $\eta \simeq \min\{1/L, \varepsilon n/L\sigma^2\}$ in Hero SGD, and $K = \max \left\{ \lceil \sigma^2/\varepsilon n \rceil, 1 \right\}$ and $\eta_g \simeq \min \left\{ \varepsilon/L\sigma^2, 1/nL \right\}$. Moreover, it is optimal up to logarithmic factors due to a result of Tyurin and Richtárik (2024).*

*Proof.* Similarly to Theorem 2.2, the second term in the $\min$ expression comes from the iteration complexity of the classical SGD method (Lan, 2020). The first term represents the time complexity of Minibatch SGD with $K = \max \left\{ \lceil \frac{\sigma^2}{\varepsilon n} \rceil, 1 \right\}$.

---

[10]Considering it does not help to improve the time complexity of Minibatch SGD/Hero SGD.

[10]Not better than the following complexity.

Tyurin and Richtárik (2024) considered an arbitrary computational and communication setup in the nonconvex setting. We can reuse their Theorem 1 with $\tau_{i \to j} \equiv \tau$ and $h_i \equiv h$ to get the lower bound

$$\Omega \left( \frac{1}{1 + \log(n+1)} \times \min \left\{ \tau \frac{L\Delta}{\varepsilon} + h \left( \frac{L\Delta}{\varepsilon} + \frac{L\sigma^2\Delta}{n\varepsilon^2} \right), h \left( \frac{L\Delta}{\varepsilon} + \frac{L\sigma^2\Delta}{\varepsilon^2} \right) \right\} \right),$$

which matches (5) up to the logarithmic factor. □

**Corollary 2.2** (Upper bound for Local SGD). *Assume that all functions $f_i$ satisfy Assumptions 1.1. Under Assumptions 1.2 and 2.1, (and $\rho$–weak convexity, the first and the second-order similarity), the time complexity of* Local SGD *and* SCAFFOLD *to find an $\varepsilon$–stationary point is **not better than***

$$\Omega \left( \sqrt{\tau h \left( \frac{L^2\sigma^2\Delta^2}{\varepsilon^3} \right)} + h \left( \frac{L\Delta}{\varepsilon} + \frac{L\sigma^2\Delta}{n\varepsilon^2} \right) \right), \tag{6}$$

*up to constant factors, using the analysis by Koloskova et al. (2020); Luo et al. (2025) (best known in terms of scaling with the number of local steps $K$).*

*Proof.* Ignoring some non-negative terms related to weak convexity and the first and second heterogeneity assumptions, in the iteration complexities of Local SGD and SCAFFOLD by Koloskova et al. (2020); Luo et al. (2025), the iteration complexity of these methods are greater than or equal to

$$\frac{L\Delta}{\varepsilon K} + \frac{L\sigma^2\Delta}{nK\varepsilon^2} + \frac{L\sigma\Delta}{K^{\frac{1}{2}}\varepsilon^{\frac{3}{2}}}$$

to find an $\varepsilon$–stationary point. Thus, the time complexity is not better than

$$\tau \left( \frac{L\Delta}{\varepsilon K} + \frac{L\sigma^2\Delta}{nK\varepsilon^2} + \frac{L\sigma\Delta}{K^{\frac{1}{2}}\varepsilon^{\frac{3}{2}}} \right) + hK \left( \frac{L\Delta}{\varepsilon K} + \frac{L\sigma^2\Delta}{nK\varepsilon^2} + \frac{L\sigma\Delta}{K^{\frac{1}{2}}\varepsilon^{\frac{3}{2}}} \right).$$

From this point we reuse the proof of Theorem 2.1. □

**Theorem 2.4** (Upper bound for Minibatch SGD). *Under Assumptions 1.1, 1.2, and 2.1, the time complexity of* Minibatch SGD *(Algorithms 2) to find an $\varepsilon$–stationary point **is no worse than***

$$O \left( \tau \frac{L\Delta}{\varepsilon} + h \left( \frac{L\Delta}{\varepsilon} + \frac{L\sigma^2\Delta}{n\varepsilon^2} \right) \right), \tag{7}$$

*up to constant factors, where we take $K = \max\left\{ \lceil \sigma^2/\varepsilon n \rceil, 1 \right\}$ and $\eta_g \simeq \min\left\{ \varepsilon/L\sigma^2, 1/nL \right\}$. Moreover, it is optimal up to constant factors due to a result of Tyurin and Richtárik (2024).*

*Proof.* This proof almost repeats the proof of Theorem 2.2 in the homogeneous regime. The only difference is that Hero SGD cannot converge in general due to the heterogeneity. The term comes from the analysis of Minibatch SGD, which needs at most

$$\tau \left( \frac{L\Delta}{\varepsilon} + \frac{\sigma^2 L\Delta}{Kn\varepsilon^2} \right) + hK \left( \frac{L\Delta}{\varepsilon} + \frac{\sigma^2 L\Delta}{Kn\varepsilon^2} \right) \quad = \tau \left( \frac{L\Delta}{\varepsilon} + \frac{\sigma^2 L\Delta}{Kn\varepsilon^2} \right) + h \left( \frac{KL\Delta}{\varepsilon} + \frac{\sigma^2 L\Delta}{n\varepsilon^2} \right)$$

seconds because $\frac{L\Delta}{\varepsilon} + \frac{\sigma^2 L\Delta}{Kn\varepsilon^2}$ is the iteration complexity of SGD with batch size $nK$ (even with heterogeneous gradients), $\tau$ is the time required to synchronize, and $hK$ is the time needed to compute $K$ stochastic gradients on each worker. Taking $K = \max\left\{ \lceil \frac{\sigma^2}{\varepsilon n} \rceil, 1 \right\}$, we obtain the result. The optimality follows from Theorem 7 in (Tyurin and Richtárik, 2024) with $\tau_{i \to j} = \tau$ and $h_i = h$. □

## C    CONVERGENCE ANALYSIS OF Dual Local SGD AND Decaying Local SGD

In this section, we provide complete proofs for both Dual Local SGD (Algorithm 4) and Decaying Local SGD methods (Algorithm 5). These proofs align more closely with the philosophy of previous works analysing Local SGD with two step sizes (Charles and Konečný, 2020; Woodworth et al., 2020; Karimireddy et al., 2020; Khaled et al., 2020; Huang et al., 2023; Jhunjhunwala et al., 2023; Malinovsky et al., 2023). We do this for the purpose of highlighting the main differences between the *standard* approach to study Local SGD and our novel approach which leverages the recent framework of Tyurin and Sivtsov (2025) and allows one to derive convergence results and time complexities for

a large family of asynchronous and local methods as discussed in Sections F and G while recovering the same results with much more simple proofs.

For the sake of providing a general convergence analysis which encompasses both Dual Local SGD and Decaying Local SGD as a special case, we denote by $\eta_{i,j}^t$ the local step size used by worker $i \in [n]$ for the $j^{\text{th}}$ local step during the $t^{\text{th}}$ communication round. This step size appears in line 5 and 6 of Algorithm 4 and Algorithm 5 respectively.

Before expanding on the convergence analysis, let us formalize the crucial following observation. The observation follows by the design of both Dual Local SGD and Decaying Local SGD.

**Corollary C.1.** *For all $t \geq 0$ $i \in [n]$ and $j \in \{0, \ldots, K-1\}$, conditionally on $z_{i,j}^t$, the random point $\xi_{i,j}^t$ is statistically independent from all the past iterates and randomness*[11].

Additionally, it is important to mention that for every communication round $t \geq 0$, conditionally on $x^t$, the iterates and random points $\{(z_{i,j}^t)\}_{j \in \{0,\ldots,K-1\}}$ and $\{(z_{i',j}^t)\}_{j \in \{0,\ldots,K-1\}}$ for distinct $i, i' \in [n]$ are statistically independent since the clients work independently from each other.

## C.1 NONCONVEX SETUP

### C.1.1 PRELIMINARY LEMMAS

We state below the descent lemma. This lemma helps to bound the decrease in function value after one communication round. We will unroll it in a subsequent lemma later and then establish the time complexity of both Dual Local SGD and Decaying Local SGD.

**Lemma C.1** (A Descent Lemma; Proof in Section D.1.1). *Under Assumptions 1.2 and 1.3 the sequences of iterates $\{x^t\}_{t \geq 0}$ and $\{z_{i,j}^t\}$ in Algorithms 4 and 5 satisfy for any integer $t \geq 0$*

$$
\mathbb{E}\left[f\left(x^{t+1}\right)\right] \leq \mathbb{E}\left[f\left(x^t\right)\right] - \frac{\eta_g n K}{2}\mathbb{E}\left[\left\|\nabla f\left(x^t\right)\right\|^2\right]
$$

$$
- \eta_g\left(\frac{1}{2} - \eta_g n L K\right)\sum_{i=1}^{n}\sum_{j=0}^{K-1}\mathbb{E}\left[\left\|\nabla f\left(z_{i,j}^t\right)\right\|^2\right]
$$

$$
+ \frac{\eta_g L^2}{2}\sum_{i=1}^{n}\sum_{j=0}^{K-1}\mathbb{E}\left[\left\|z_{i,j}^t - x^t\right\|^2\right]
$$

$$
+ \eta_g^2 \sigma^2 n L K,
$$

**Lemma C.2** (Residual Estimation; Proof in Section D.1.2). *Under Assumption 1.3 the sequences of iterates $\{x^t\}_{t \geq 0}$ and $\{z_{i,j}^t\}_{t \geq 0}$ in Algorithms 4 and 5 satisfy for any integers $t \geq 0$, $i \in [n]$ and $j \in \{0, \ldots, K-1\}$*

$$
\mathbb{E}\left[\left\|z_{i,j}^t - x^t\right\|^2\right] \leq 2\left(\sum_{\ell=0}^{j-1}\left(\eta_{i,\ell}^t\right)^2\right)\sum_{\ell=0}^{j-1}\mathbb{E}\left[\left\|\nabla f\left(z_{i,\ell}^t\right)\right\|^2\right] + 2\sigma^2\sum_{\ell=0}^{j-1}\left(\eta_{i,\ell}^t\right)^2.
$$

We are now ready to unroll the descent lemma.

**Lemma C.3** (Unrolling the Descent Lemma; Proof in Section D.1.3). *Under Assumptions 1.2 and 1.3 the sequences of iterates $\{x^t\}_{t \geq 0}$ and $\{z_{i,j}^t\}_{t \geq 0}$ in Algorithms 4 and 5 satisfy for any integer $R \geq 1$*

$$
\frac{1}{R}\sum_{t=0}^{R-1}\mathbb{E}\left[\left\|\nabla f\left(x^t\right)\right\|^2\right] \leq \frac{2\Delta}{\eta_g n K R} + 2\eta_g \sigma^2 L + \frac{2\sigma^2 L^2}{nR}\sum_{t=0}^{R-1}\sum_{i=1}^{n}\sum_{\ell=0}^{K-1}\left(\eta_{i,\ell}^t\right)^2
$$

$$
- \frac{2}{nKR}\sum_{t=0}^{R-1}\sum_{i=1}^{n}\left(\frac{1}{2} - \eta_g n L K - \left(\sum_{\ell=0}^{K-1}\left(\eta_{i,\ell}^t\right)^2\right)L^2 K\right)\sum_{j=0}^{K-1}\mathbb{E}\left[\left\|\nabla f\left(z_{i,j}^t\right)\right\|^2\right].
$$

---

[11]To avoid unnecessarily complicated notation to characterize these *past iterates and randomness* we simply mention that it represents all previous iterates and random point computed up to the time where $z_{i,j}^t$ is computed.

### C.1.2 MAIN RESULTS

**Theorem 3.1.** *Under Assumptions 1.2 and 1.3,* Dual Local SGD *(Algorithm 4) with* $\eta_g = \min\left\{\frac{\varepsilon}{8L\sigma^2}, \frac{1}{4nL}\right\}$ *and* $\eta_\ell \leq \sqrt{n}\eta_g$ *finds an $\varepsilon$–stationary point after at most $R = \lceil 32L\Delta/\varepsilon \rceil$ communication rounds with $K = \max\left\{\lceil \sigma^2/\varepsilon n \rceil, 1\right\}$. Additionally, under Assumption 1.1, it requires at most*

$$\tau\frac{64L\Delta}{\varepsilon} + 64h\left(\frac{L\Delta}{\varepsilon} + \frac{L\sigma^2\Delta}{n\varepsilon^2}\right) \tag{11}$$

*seconds to find an $\varepsilon$–stationary point.*

*Proof.* In Algorithm 4 the local stepsize is constant so it does not depend on the communication round $t$, neither on the worker index $i \in [n]$ nor on the number of local steps $\ell \in \{0, \ldots, K-1\}$ thus the inequality from Lemma C.3 can be simplified to

$$\frac{1}{R}\sum_{t=0}^{R-1}\mathbb{E}\left[\left\|\nabla f\left(x^t\right)\right\|^2\right] \leq \frac{2\Delta}{\eta_g nKR} + 2\eta_g\sigma^2 L + 2\eta_\ell^2\sigma^2 KL^2$$

$$-\frac{2}{nKR}\sum_{t=0}^{R-1}\sum_{i=1}^{n}\left(\frac{1}{2} - \eta_g nLK - \eta_\ell^2 L^2 K^2\right)\sum_{j=0}^{K-1}\mathbb{E}\left[\left\|\nabla f\left(z_{i,j}^t\right)\right\|^2\right],$$

for any integer $R \geq 1$. Then, let us check the non-negativity of $\frac{1}{2} - \eta_g nLK - \eta_\ell^2 L^2 K^2$ using the conditions $\eta_g = \min\left\{\frac{\varepsilon}{8L\sigma^2}, \frac{1}{4nL}\right\}$, $\eta_\ell \leq \sqrt{n}\eta_g$ and $K = \max\left\{\lceil \sigma^2/\varepsilon n \rceil, 1\right\}$. We distinguish two cases: if $\sigma^2/n\varepsilon \leq 1$ then $K = 1$ and

$$\eta_g nLK = \eta_g nL \leq \frac{1}{4}, \tag{12}$$

since $\eta_g \leq \frac{1}{4nL}$. Otherwise, if $\sigma^2/n\varepsilon > 1$ then $K = \lceil \sigma^2/\varepsilon n \rceil$ and using $\eta_g \leq \frac{\varepsilon}{8L\sigma^2}$

$$\eta_g nLK = \eta_g nL\left\lceil\frac{\sigma^2}{n\varepsilon}\right\rceil \leq \frac{n\varepsilon}{8\sigma^2}\left(1 + \frac{\sigma^2}{n\varepsilon}\right) \leq \frac{n\varepsilon}{8\sigma^2} + \frac{1}{8} \leq \frac{1}{4}. \tag{13}$$

Moreover, since $\eta_\ell \leq \sqrt{n}\eta_g$ we have

$$\eta_\ell^2 K^2 L^2 \leq \eta_g^2 nL^2 K^2 \leq (\eta_g nLK)^2 \overset{(12)+(13)}{\leq} \frac{1}{16}, \tag{14}$$

because $n \geq 1$. Combining the upper bounds (12), (13) and (14) we obtain

$$\frac{1}{2} - \eta_g nLK - \eta_\ell^2 L^2 K^2 \geq 0,$$

hence,

$$\frac{1}{R}\sum_{t=0}^{R-1}\mathbb{E}\left[\left\|\nabla f\left(x^t\right)\right\|^2\right] \leq \frac{2\Delta}{\eta_g nKR} + 2\eta_g\sigma^2 L + 2\eta_\ell^2\sigma^2 KL^2$$

$$\leq \frac{2\Delta}{\eta_g nKR} + 2\eta_g\sigma^2 L\left(1 + \eta_g nLK\right)$$

$$\overset{(12)+(13)}{\leq} \frac{2\Delta}{\eta_g nKR} + 2\eta_g\sigma^2 L\left(1 + \frac{1}{4}\right)$$

$$= \frac{2\Delta}{\eta_g nKR} + \frac{5}{2}\eta_g\sigma^2 L$$

$$\overset{(a)}{\leq} \frac{2\Delta}{\eta_g nKR} + \frac{\varepsilon}{2},$$

where in (a) we use $\eta_g \leq \frac{\varepsilon}{8L\sigma^2}$. Then, it remains to choose $R \geq 1$ such that $\frac{2\Delta}{\eta_g nKR} \leq \frac{\varepsilon}{2}$, that is,

$$R \geq \frac{4\Delta L}{\eta_g nLK\varepsilon} = \frac{4\Delta L}{\varepsilon}\max\left\{\frac{8L\sigma^2}{\varepsilon nLK}, \frac{4nL}{nLK}\right\} = \frac{4\Delta L}{\varepsilon}\max\left\{\frac{8\sigma^2}{\varepsilon nK}, \frac{4}{K}\right\}.$$

Now, observe that $K \geq 1$ thus $\frac{4}{K} \leq 4$. Moreover, if $\sigma^2/n\varepsilon \leq 1$ then

$$\frac{8\sigma^2}{\varepsilon nK} \leq \frac{8}{K} \leq 8, \tag{15}$$

while, if $\sigma^2/n\varepsilon > 1$ we have $K = \lceil \sigma^2/n\varepsilon \rceil \geq \sigma^2/n\varepsilon$ so

$$\frac{8\sigma^2}{n\varepsilon K} \leq 8. \tag{16}$$

Combining the upper bounds (15) and (16) we have $\max\left\{ \frac{8\sigma^2}{\varepsilon nK}, \frac{4}{K} \right\} \leq 8$ hence, it is enough to take

$$R \geq \max\left\{ 1, \frac{32L\Delta}{\varepsilon} \right\},$$

so as to guarantee Dual Local SGD to find an $\varepsilon$–stationary point.

To derive the time complexity (11), we know that there are $R$ communication rounds and in any of these rounds each worker performs $K$ local steps thus under Assumption 1.1 it requires at most

$$\tau R + hKR \leq \tau R + hR \left( 1 + \frac{\sigma^2}{n\varepsilon} \right) \leq \tau \frac{64L\Delta}{\varepsilon} + 64h \left( \frac{L\Delta}{\varepsilon} + \frac{L\sigma^2\Delta}{n\varepsilon^2} \right)$$

seconds for Dual Local SGD to find an $\varepsilon$–stationary point, as long as $\varepsilon \lesssim L\Delta$[12]. $\qquad\square$

**Theorem 4.1.** *Under Assumptions 1.2 and 1.3,* Decaying Local SGD *(Algorithm 5) with $\eta_g = \min\left\{ \frac{\varepsilon}{8L\sigma^2}, \frac{1}{4nL} \right\}$ and $b = \max\left\{ \frac{\sigma^2}{\varepsilon}, n \right\}$ finds an $\varepsilon$–stationary point after at most $R = \lceil 32L\Delta/\varepsilon \rceil$ communication rounds with $K = \max\left\{ \lceil \sigma^2/\varepsilon n \rceil, 1 \right\}$. Additionally, under Assumption 1.1, it requires at most*

$$\tau \frac{64L\Delta}{\varepsilon} + 64h \left( \frac{L\Delta}{\varepsilon} + \frac{L\sigma^2\Delta}{n\varepsilon^2} \right)$$

*seconds to find an $\varepsilon$–stationary point.*

*Proof.* In Algorithm 5 the local stepsize depends on $j \in \{0, \ldots, K-1\}$ and it does not depend on the communication round $t$ nor on the worker index $i \in [n]$ hence in Lemma C.3 we have the simplifications

$$\frac{1}{nR} \sum_{t=0}^{R-1} \sum_{i=1}^{n} \sum_{\ell=0}^{K-1} \left( \eta_{i,\ell}^t \right)^2 = \sum_{\ell=0}^{K-1} \eta_\ell^2,$$

and

$$\sum_{\ell=0}^{K-1} \eta_\ell^2 = \eta_g^2 \sum_{\ell=0}^{K-1} \frac{b}{(\ell+1)(\log K + 1)} = \frac{\eta_g^2 b}{\log K + 1} \sum_{j=1}^{K} \frac{1}{j} \overset{\text{(a)}}{\leq} \frac{\eta_g^2 b (\log K + 1)}{\log K + 1} = \eta_g^2 b, \tag{17}$$

where (a) follows from the well-known inequality $1 + \frac{1}{2} + \cdots + \frac{1}{n} \leq 1 + \log(n)$ which holds for all integer $n \geq 1$. Thus, using Lemma C.3 and (17) we obtain

$$\frac{1}{R} \sum_{t=0}^{R-1} \mathbb{E}\left[ \left\| \nabla f\left(x^t\right) \right\|^2 \right] \leq \frac{2\Delta}{\eta_g nKR} + 2\eta_g \sigma^2 L + 2\eta_g^2 \sigma^2 L^2 b$$

$$- \frac{2}{nKR} \sum_{t=0}^{R-1} \sum_{i=1}^{n} \left( \frac{1}{2} - \eta_g nLK - \eta_g^2 L^2 Kb \right) \sum_{j=0}^{K-1} \mathbb{E}\left[ \left\| \nabla f\left(z_{i,j}^t\right) \right\|^2 \right],$$

---

[12]From Lemma E.10 we know that $\left\| \nabla f\left(x^0\right) \right\|^2 \leq 2L\left( f\left(x^0\right) - f^{\inf} \right) = 2L\Delta$ hence if $\varepsilon \geq 2L\Delta$ it means $x^0$ is already an $\varepsilon$–stationary point and we can simply return $x^0$. On the other hand, if $\varepsilon < 2L\Delta$ then $\frac{32L\Delta}{\varepsilon} \geq 16 > 1$ so it is enough to take $R = \lceil 32L\Delta/\varepsilon \rceil$. Additionally, we have

$$R = \left\lceil \frac{32L\Delta}{\varepsilon} \right\rceil \leq \frac{32L\Delta}{\varepsilon} + 1 \leq \frac{64L\Delta}{\varepsilon}.$$

for any integer $R \geq 1$. Then, let us check the non-negativity of $\frac{1}{2} - \eta_g nLK - \eta_g^2 L^2 Kb$ using the conditions $\eta_g = \min\left\{\frac{\varepsilon}{8L\sigma^2}, \frac{1}{4nL}\right\}$, $K = \max\left\{\left\lceil \sigma^2/\varepsilon n \right\rceil, 1\right\}$ and $b = \max\left\{\sigma^2/\varepsilon, n\right\}$. We distinguish two cases: if $\sigma^2/n\varepsilon \leq 1$ then $K = 1$ and $b = n$ so

$$\eta_g nLK = \eta_g nL \leq \frac{1}{4}, \tag{18}$$

and

$$\eta_g^2 L^2 Kb = \eta_g^2 nL^2 \leq \frac{1}{16n} \leq \frac{1}{4},$$

since $n \geq 1$ and $\eta_g \leq \frac{1}{4nL}$. Otherwise, if $\sigma^2/n\varepsilon > 1$ then $K = \left\lceil \sigma^2/\varepsilon n \right\rceil$ and $b = \frac{\sigma^2}{\varepsilon}$ and using $\eta_g \leq \frac{\varepsilon}{8L\sigma^2}$ we have

$$\eta_g nLK = \eta_g nL \left\lceil \frac{\sigma^2}{n\varepsilon} \right\rceil \leq \frac{n\varepsilon}{8\sigma^2}\left(1 + \frac{\sigma^2}{n\varepsilon}\right) \leq \frac{n\varepsilon}{8\sigma^2} + \frac{1}{8} \leq \frac{1}{4}, \tag{19}$$

and

$$\eta_g^2 L^2 Kb = \eta_g^2 nL^2 \left\lceil \frac{\sigma^2}{n\varepsilon} \right\rceil \frac{\sigma^2}{n\varepsilon} \leq \frac{1}{n}\left(\eta_g nL \left\lceil \frac{\sigma^2}{n\varepsilon} \right\rceil\right)^2 \overset{(19)}{\leq} \frac{1}{16n} \leq \frac{1}{4}, \tag{20}$$

because $n \geq 1$. Combining the upper bounds (18), (19) and (20) we obtain

$$\frac{1}{2} - \eta_g nLK - \eta_g^2 L^2 Kb \geq 0.$$

Moreover, we have $2\eta_g \sigma^2 L \leq \frac{\varepsilon}{4}$ and by definition of $b$

$$2\eta_g^2 \sigma^2 L^2 b = 2\eta_g \sigma^2 L \max\left\{\frac{\eta_g \sigma^2 L}{\varepsilon}, \eta_g nL\right\}$$

$$\leq \frac{\varepsilon}{4} \max\left\{\frac{\eta_g \sigma^2 L}{\varepsilon}, \eta_g nL\right\}$$

$$\leq \frac{\varepsilon}{4} \max\left\{\frac{1}{8}, \frac{1}{4}\right\}$$

$$\leq \frac{\varepsilon}{4},$$

thus $2\eta_g \sigma^2 L + 2\eta_g^2 \sigma^2 L^2 b \leq \frac{\varepsilon}{2}$. Now, notice that the global step size $\eta_g$ and the number of local steps $K$ are the same in Theorem 3.1 and in Theorem 4.1 thus as done in the proof of Theorem 3.1, it is enough to take

$$R \geq \max\left\{1, \frac{32L\Delta}{\varepsilon}\right\},$$

in order to ensure Decaying Local SGD finds an $\varepsilon$–stationary point. Moreover, under Assumption 1.1 Decaying Local SGD requires at most

$$\tau \frac{64L\Delta}{\varepsilon} + 64h\left(\frac{L\Delta}{\varepsilon} + \frac{L\sigma^2\Delta}{n\varepsilon^2}\right)$$

second to find such an $\varepsilon$–stationary point. This achieves the proof of the result. $\qquad\square$

## C.2 CONVEX SETUP

### C.2.1 PRELIMINARY LEMMAS

**Lemma C.4** (A First Descent Lemma; Proof in Section D.2.1). *Under Assumptions 1.2 to 1.4 the sequences of iterates $\{x^t\}_{t\geq 0}$ and $\{z_{i,j}^t\}$ in Algorithms 4 and 5 satisfy for any integer $t \geq 0$*

$$\mathbb{E}\left[\left\|x^{t+1} - x^*\right\|^2\right] \leq \mathbb{E}\left[\left\|x^t - x^*\right\|^2\right] + 2\eta_g L \sum_{i=1}^{n}\sum_{j=0}^{K-1} \mathbb{E}\left[\left\|z_{i,j}^t - x^*\right\|^2\right]$$

$$- 2\eta_g\left(\frac{3}{4} - \eta_g nLK\right)\sum_{i=1}^{n}\sum_{j=0}^{K-1} \mathbb{E}\left[\left\langle\nabla f\left(z_{i,j}^t\right), z_{i,j}^t - x^*\right\rangle\right] + 2\eta_g^2 \sigma^2 nK.$$

**Lemma C.5** (A Descent Lemma on $\{f(z_{i,j}^t)\}$; Proof in Section D.2.2). *Under Assumptions 1.2 to 1.4 the sequences of iterates $\{x^t\}_{t\geq 0}$ and $\{z_{i,j}^t\}$ in Algorithms 4 and 5 satisfy for any integer $t \geq 0$*

$$\frac{\eta_g}{2} \sum_{i=1}^{n} \sum_{j=0}^{K-1} \left(\mathbb{E}\left[f\left(z_{i,j}^t\right)\right] - f^{\inf}\right)$$

$$\leq \mathbb{E}\left[\left\|x^t - x^*\right\|^2\right] - \mathbb{E}\left[\left\|x^{t+1} - x^*\right\|^2\right] + 2\eta_g L \sum_{i=1}^{n} \sum_{j=0}^{K-1} \mathbb{E}\left[\left\|z_{i,j}^t - x^*\right\|^2\right]$$

$$- 2\eta_g \left(\frac{1}{2} - \eta_g n L K\right) \sum_{i=1}^{n} \sum_{j=0}^{K-1} \mathbb{E}\left[\left\langle \nabla f\left(z_{i,j}^t\right), z_{i,j}^t - x^*\right\rangle\right] + 2\eta_g^2 \sigma^2 n K.$$

**Lemma C.6** (A Descent Lemma on $\{f(x^t)\}$; Proof in Section D.2.3). *Under Assumptions 1.2 to 1.4 the sequences of iterates $\{x^t\}_{t\geq 0}$ and $\{z_{i,j}^t\}$ in Algorithms 4 and 5 satisfy for any integer $t \geq 0$*

$$\frac{\eta_g}{4} \left(\mathbb{E}\left[f\left(x^t\right)\right] - f^{\inf}\right)$$

$$\leq \frac{1}{nK}\left(\mathbb{E}\left[\left\|x^t - x^*\right\|^2\right] - \mathbb{E}\left[\left\|x^{t+1} - x^*\right\|^2\right]\right) + \frac{5\eta_g L}{2nK} \sum_{i=1}^{n} \sum_{j=0}^{K-1} \mathbb{E}\left[\left\|z_{i,j}^t - x^*\right\|^2\right]$$

$$- 2\eta_g \left(\frac{1}{2} - \eta_g n L K\right) \frac{1}{nK} \sum_{i=1}^{n} \sum_{j=0}^{K-1} \mathbb{E}\left[\left\langle \nabla f\left(z_{i,j}^t\right), z_{i,j}^t - x^*\right\rangle\right] + 2\eta_g^2 \sigma^2.$$

**Lemma C.7** (Residual Estimation; Proof in Section D.2.4). *Under Assumptions 1.3 and 1.4 the sequences of iterates $\{x^t\}_{t\geq 0}$ and $\{z_{i,j}^t\}_{t\geq 0}$ in Algorithms 4 and 5 satisfy for any integers $t \geq 0$, $i \in [n]$ and $j \in \{0, \ldots, K-1\}$*

$$\mathbb{E}\left[\left\|z_{i,j}^t - x^t\right\|^2\right] \leq 2L \left(\sum_{\ell=0}^{j-1} \left(\eta_{i,\ell}^t\right)^2\right) \sum_{\ell=0}^{j-1} \mathbb{E}\left[\left\langle \nabla f\left(z_{i,\ell}^t\right), z_{i,\ell}^t - x^*\right\rangle\right] + 2\sigma^2 \sum_{\ell=0}^{j-1} \left(\eta_{i,\ell}^t\right)^2.$$

We are now ready to unroll the descent lemma, we state formally the bound we obtain in the following lemma.

**Lemma C.8** (Unrolling the Descent Lemma C.5; Proof in Section D.2.5). *Under Assumptions 1.2 to 1.4 the sequences of iterates $\{x^t\}_{t\geq 0}$ and $\{z_{i,j}^t\}_{t\geq 0}$ in Algorithms 4 and 5 satisfy for any integer $R \geq 1$*

$$\frac{1}{R} \sum_{t=0}^{R-1} \left(\mathbb{E}\left[f\left(x^t\right)\right] - f^{\inf}\right)$$

$$\leq \frac{4B^2}{\eta_g nKR} + 8\eta_g \sigma^2 + \frac{20\sigma^2 L}{nR} \sum_{t=0}^{R-1} \sum_{i=1}^{n} \sum_{\ell=0}^{K-1} \left(\eta_{i,\ell}^t\right)^2$$

$$- \frac{8}{nKR} \sum_{t=0}^{R-1} \sum_{i=1}^{n} \left(\frac{1}{2} - \eta_g nLK - \frac{5}{2}\left(\sum_{\ell=0}^{K-1} \left(\eta_{i,\ell}^t\right)^2\right) L^2 K\right) \sum_{j=0}^{K-1} \mathbb{E}\left[\left\langle \nabla f\left(z_{i,j}^t\right), z_{i,j}^t - x^*\right\rangle\right].$$

### C.2.2 MAIN RESULTS

**Theorem C.1** (Upper bound for Dual Local SGD). *Under Assumptions 1.2 to 1.4, Dual Local SGD (Algorithm 4) with $\eta_g = \min\left\{\frac{\varepsilon}{20\sigma^2}, \frac{1}{10nL}\right\}$ and $\eta_\ell \leq \sqrt{n}\eta_g$ finds an $\varepsilon$–solution after at most $R = \left\lceil 160LB^2/\varepsilon \right\rceil$ communication rounds with $K = \max\left\{\left\lceil \sigma^2/\varepsilon nL \right\rceil, 1\right\}$. Additionally, under Assumption 1.1, it requires at most*

$$\tau \frac{320LB^2}{\varepsilon} + 320h\left(\frac{LB^2}{\varepsilon} + \frac{\sigma^2 B^2}{n\varepsilon^2}\right) \tag{21}$$

*seconds to find an $\varepsilon$–solution.*

*Proof.* In Algorithm 4 the local stepsize is constant so it does not depend on the communication round $t$, neither on the worker index $i \in [n]$ nor on the number of local steps $\ell \in \{0, \dots, K-1\}$ thus the inequality from Lemma C.8 can be simplified to

$$
\frac{1}{R} \sum_{t=0}^{R-1} \left( \mathbb{E}\left[ f\left(x^t\right)\right] - f^{\inf} \right)
$$

$$
\leq \frac{4B^2}{\eta_g nKR} + 8\eta_g \sigma^2 + 20\eta_\ell^2 \sigma^2 LK
$$

$$
- \frac{8}{nKR} \sum_{t=0}^{R-1} \sum_{i=1}^{n} \left( \frac{1}{2} - \eta_g nLK - \frac{5}{2}\eta_\ell^2 L^2 K^2 \right) \sum_{j=0}^{K-1} \mathbb{E}\left[ \left\langle \nabla f\left(z_{i,j}^t\right), z_{i,j}^t - x^* \right\rangle \right],
$$

for any integer $R \geq 1$. Then, let us check the non-negativity of $\frac{1}{2} - \eta_g nLK - \frac{5}{2}\eta_\ell^2 L^2 K^2$ using the conditions $\eta_g = \min\left\{ \frac{\varepsilon}{20L\sigma^2}, \frac{1}{10nL} \right\}$, $\eta_\ell \leq \sqrt{n}\eta_g$ and $K = \max\left\{ \lceil \sigma^2/\varepsilon nL \rceil, 1 \right\}$. We distinguish two cases: if $\sigma^2/nL\varepsilon \leq 1$ then $K = 1$ and

$$
\eta_g nLK = \eta_g nL \leq \frac{1}{10}, \tag{22}
$$

since $\eta_g \leq \frac{1}{10nL}$. Otherwise, if $\sigma^2/nL\varepsilon > 1$ then $K = \lceil \sigma^2/\varepsilon nL \rceil$ and using $\eta_g \leq \frac{\varepsilon}{20\sigma^2}$

$$
\eta_g nLK = \eta_g nL \left\lceil \frac{\sigma^2}{nL\varepsilon} \right\rceil \leq \frac{nL\varepsilon}{20\sigma^2} \left( 1 + \frac{\sigma^2}{nL\varepsilon} \right) \leq \frac{nL\varepsilon}{20\sigma^2} + \frac{1}{20} \leq \frac{1}{10}. \tag{23}
$$

Moreover, since $\eta_\ell \leq \sqrt{n}\eta_g$ we have

$$
\eta_\ell^2 K^2 L^2 \leq \eta_g^2 nL^2 K^2 \leq \left( \eta_g nLK \right)^2 \overset{(22)+(23)}{\leq} \frac{1}{100}, \tag{24}
$$

because $n \geq 1$. Combining the upper bounds (22), (23) and (24) we obtain

$$
\frac{1}{2} - \eta_g nLK - \frac{5}{2}\eta_\ell^2 L^2 K^2 \geq 0,
$$

hence,

$$
\frac{1}{R} \sum_{t=0}^{R-1} \left( \mathbb{E}\left[ f\left(x^t\right)\right] - f^{\inf} \right) \leq \frac{4B^2}{\eta_g nKR} + 8\eta_g \sigma^2 + 20\eta_\ell^2 \sigma^2 LK
$$

$$
\leq \frac{4B^2}{\eta_g nKR} + 8\eta_g \sigma^2 \left( 1 + \frac{5}{2}\eta_g nLK \right)
$$

$$
\overset{(22)+(23)}{\leq} \frac{4B^2}{\eta_g nKR} + 8\eta_g \sigma^2 \left( 1 + \frac{1}{4} \right)
$$

$$
= \frac{4B^2}{\eta_g nKR} + 10\eta_g \sigma^2 L
$$

$$
\overset{(a)}{\leq} \frac{4B^2}{\eta_g nKR} + \frac{\varepsilon}{2},
$$

where in (a) we use $\eta_g \leq \frac{\varepsilon}{20\sigma^2}$. Then, it remains to choose $R \geq 1$ such that $\frac{4B^2}{\eta_g nKR} \leq \frac{\varepsilon}{2}$, that is,

$$
R \geq \frac{8LB^2}{\eta_g nLK\varepsilon} = \frac{8LB^2}{\varepsilon} \max\left\{ \frac{20\sigma^2}{\varepsilon nLK}, \frac{10nL}{nLK} \right\} = \frac{8LB^2}{\varepsilon} \max\left\{ \frac{20\sigma^2}{\varepsilon nLK}, \frac{10}{K} \right\}.
$$

Now, observe that $K \geq 1$ thus $\frac{10}{K} \leq 10$. Moreover, if $\sigma^2/nL\varepsilon \leq 1$ then

$$
\frac{20\sigma^2}{\varepsilon nLK} \leq \frac{20}{K} \leq 20, \tag{25}
$$

while, if $\sigma^2/nL\varepsilon > 1$ we have $K = \lceil \sigma^2/nL\varepsilon \rceil \geq \sigma^2/nL\varepsilon$ so

$$
\frac{20\sigma^2}{nL\varepsilon K} \leq 20. \tag{26}
$$

Combining the upper bounds (25) and (26) we have $\max\left\{\frac{20\sigma^2}{\varepsilon nLK}, \frac{10}{K}\right\} \leq 20$ hence, it is enough to take

$$R \geq \max\left\{1, \frac{160L\Delta}{\varepsilon}\right\},$$

so as to guarantee Dual Local SGD to find an $\varepsilon$–stationary point.

To derive the time complexity (21), we know that there are $R$ communication rounds and in any of these rounds each worker performs $K$ local steps thus under Assumption 1.1 it requires at most

$$\tau R + hKR \leq \tau R + hR\left(1 + \frac{\sigma^2}{n\varepsilon}\right) \leq \tau\frac{320LB^2}{\varepsilon} + 320h\left(\frac{LB^2}{\varepsilon} + \frac{\sigma^2 B^2}{n\varepsilon^2}\right)$$

seconds for Dual Local SGD to find an $\varepsilon$–stationary point, as long as $\varepsilon \lesssim L\Delta$[13]. $\qquad\square$

**Theorem C.2** (Upper bound for Decaying Local SGD). *Under Assumptions 1.2 to 1.4,* Decaying Local SGD *(Algorithm 5) with $\eta_g = \min\{\frac{\varepsilon}{20\sigma^2}, \frac{1}{10nL}\}$ and $b = \max\{\frac{\sigma^2}{\varepsilon L}, n\}$ finds an $\varepsilon$–stationary point after at most $R = \lceil 160LB^2/\varepsilon \rceil$ communication rounds with $K = \max\left\{\lceil \sigma^2/\varepsilon nL \rceil, 1\right\}$. Additionally, under Assumption 1.1, it requires at most*

$$\tau\frac{320LB^2}{\varepsilon} + 320h\left(\frac{LB^2}{\varepsilon} + \frac{\sigma^2 B^2}{n\varepsilon^2}\right)$$

*seconds to find an $\varepsilon$–stationary point.*

*Proof.* The same way as we did in the proof of Theorem 4.1, we have

$$\frac{1}{nR}\sum_{t=0}^{R-1}\sum_{i=1}^{n}\sum_{\ell=0}^{K-1}\left(\eta_{i,\ell}^t\right)^2 = \sum_{\ell=0}^{K-1}\eta_\ell^2 \leq \eta_g^2 b,$$

thus, using Lemma C.8 we obtain

$$\frac{1}{R}\sum_{t=0}^{R-1}\left(\mathbb{E}\left[f\left(x^t\right)\right] - f^{\inf}\right)$$

$$\leq \frac{4B^2}{\eta_g nKR} + 8\eta_g\sigma^2 + 20\eta_\ell^2\sigma^2 Lb$$

$$- \frac{8}{nKR}\sum_{t=0}^{R-1}\sum_{i=1}^{n}\left(\frac{1}{2} - \eta_g nLK - \frac{5}{2}\eta_\ell^2 L^2 Kb\right)\sum_{j=0}^{K-1}\mathbb{E}\left[\left\langle\nabla f\left(z_{i,j}^t\right), z_{i,j}^t - x^*\right\rangle\right],$$

for any integer $R \geq 1$. Then, let us check the non-negativity of $\frac{1}{2} - \eta_g nLK - \frac{5}{2}\eta_g^2 L^2 Kb$ using the conditions $\eta_g = \min\left\{\frac{\varepsilon}{20\sigma^2}, \frac{1}{10nL}\right\}$, $K = \max\left\{\lceil \sigma^2/\varepsilon nL \rceil, 1\right\}$ and $b = \max\left\{\sigma^2/\varepsilon L, n\right\}$. We distinguish two cases: if $\sigma^2/nL\varepsilon \leq 1$ then $K = 1$ and $b = n$ so

$$\eta_g nLK = \eta_g nL \leq \frac{1}{10}, \tag{27}$$

and

$$\eta_g^2 L^2 Kb = \eta_g^2 nL^2 \leq \frac{1}{100n} \leq \frac{1}{10},$$

since $n \geq 1$ and $\eta_g \leq \frac{1}{10nL}$. Otherwise, if $\sigma^2/nL\varepsilon > 1$ then $K = \lceil \sigma^2/\varepsilon nL \rceil$ and $b = \frac{\sigma^2}{\varepsilon L}$ and using $\eta_g \leq \frac{\varepsilon}{20\sigma^2}$ we have

$$\eta_g nLK = \eta_g nL\left\lceil\frac{\sigma^2}{nL\varepsilon}\right\rceil \leq \frac{nL\varepsilon}{20\sigma^2}\left(1 + \frac{\sigma^2}{nL\varepsilon}\right) \leq \frac{nL\varepsilon}{20\sigma^2} + \frac{1}{20} \leq \frac{1}{10}, \tag{28}$$

and

$$\eta_g^2 L^2 Kb = \eta_g^2 nL^2\left\lceil\frac{\sigma^2}{nL\varepsilon}\right\rceil\frac{\sigma^2}{nL\varepsilon} \leq \frac{1}{n}\left(\eta_g nL\left\lceil\frac{\sigma^2}{nL\varepsilon}\right\rceil\right)^2 \overset{(28)}{\leq} \frac{1}{100n} \leq \frac{1}{10}, \tag{29}$$

---

[13]See the footnote at the end of the proof of Theorem 3.1

because $n \geq 1$. Combining the upper bounds (27), (28) and (29) we obtain

$$\frac{1}{2} - \eta_g n L K - \frac{5}{2} \eta_g^2 L^2 K b \geq 0.$$

Moreover, we have

$$
\begin{aligned}
8\eta_g \sigma^2 + 20\eta_g^2 \sigma^2 L b &= 8\eta_g \sigma^2 \left(1 + \frac{5}{2} \eta_g L b\right) \\
&\overset{\text{(a)}}{\leq} 8\eta_g \sigma^2 \left(1 + \frac{5}{2} \eta_g n L K\right) \\
&\overset{(27)+(28)}{\leq} 8\eta_g \left(1 + \frac{1}{4}\right) \\
&= 10\eta_g \sigma^2 \\
&\overset{\text{(b)}}{\leq} \frac{\varepsilon}{2},
\end{aligned}
$$

where in (a) we use the fact that $b \leq nK$ and in (b) we use $\eta_g \leq \frac{\varepsilon}{20\sigma^2}$. Thus $8\eta_g \sigma^2 + 20\eta_g^2 \sigma^2 L b \leq \frac{\varepsilon}{2}$. Now, notice that the global step size $\eta_g$ and the number of local steps $K$ are the same in Theorem 3.1 and in Theorem 4.1 thus as done in the proof of Theorem C.1, it is enough to take

$$R \geq \max\left\{1, \frac{160 L \Delta}{\varepsilon}\right\},$$

in order to ensure Decaying Local SGD finds an $\varepsilon$–stationary point. Moreover, under Assumption 1.1 Decaying Local SGD requires at most

$$\tau \frac{320 L \Delta}{\varepsilon} + 320 h \left(\frac{L\Delta}{\varepsilon} + \frac{L\sigma^2 \Delta}{n\varepsilon^2}\right)$$

second to find such an $\varepsilon$–stationary point. This achieves the proof of the result. $\qquad \square$

# D  PROOFS FOR THE RESULTS IN SECTION C

## D.1  NONCONVEX SETUP

### D.1.1  PROOF OF THE DESCENT LEMMA

**Lemma C.1.** *Under Assumptions 1.2 and 1.3 the sequences of iterates $\{x^t\}_{t \geq 0}$ and $\{z_{i,j}^t\}$ in Algorithms 4 and 5 satisfy for any integer $t \geq 0$*

$$
\begin{aligned}
\mathbb{E}\left[f\left(x^{t+1}\right)\right] \leq{} & \mathbb{E}\left[f\left(x^t\right)\right] - \frac{\eta_g n K}{2} \mathbb{E}\left[\left\|\nabla f\left(x^t\right)\right\|^2\right] \\
& - \eta_g \left(\frac{1}{2} - \eta_g n L K\right) \sum_{i=1}^{n} \sum_{j=0}^{K-1} \mathbb{E}\left[\left\|\nabla f\left(z_{i,j}^t\right)\right\|^2\right] \\
& + \frac{\eta_g L^2}{2} \sum_{i=1}^{n} \sum_{j=0}^{K-1} \mathbb{E}\left[\left\|z_{i,j}^t - x^t\right\|^2\right] \\
& + \eta_g^2 \sigma^2 n L K,
\end{aligned}
$$

*Proof.* According to Assumption 1.2 we know that the function $f$ is $L$–smooth (Nesterov, 2018) thus it holds

$$
\begin{aligned}
f\left(x^{t+1}\right) &\overset{\text{Ass. 1.2}}{\leq} f\left(x^t\right) + \left\langle \nabla f\left(x^t\right), x^{t+1} - x^t\right\rangle + \frac{L}{2}\left\|x^{t+1} - x^t\right\|^2 \\
&\overset{\text{(a)}}{=} f\left(x^t\right) - \eta_g \left\langle \nabla f\left(x^t\right), \sum_{i=1}^{n} \sum_{j=0}^{K-1} \nabla f\left(z_{i,j}^t; \xi_{i,j}^t\right)\right\rangle \qquad (30) \\
&\quad + \frac{\eta_g^2 L}{2} \left\|\sum_{i=1}^{n} \sum_{j=0}^{K-1} \nabla f\left(z_{i,j}^t; \xi_{i,j}^t\right)\right\|^2,
\end{aligned}
$$

where in (a) we use the relation

$$x^{t+1} = x^t - \eta_g \sum_{i=1}^{n} \sum_{j=0}^{K-1} \nabla f\left(z_{i,j}^t; \xi_{i,j}^t\right),$$

from lines 8 and 9 of Algorithm 4 and Algorithm 5 respectively. According to Corollary C.1, we know that conditionally on $z_{i,j}^t$, the random point $\xi_{i,j}^t$ is independent from all the past iterates and randomness so in particular, it is independent from $x^t$ (unless $j = 0$ because $z_{i,0}^t = x^t$). Hence by Corollary C.1 we have

$$\mathbb{E}\left[\nabla f\left(z_{i,j}^t; \xi_{i,j}^t\right) \mid x^t, z_{i,j}^t\right] = \mathbb{E}\left[\nabla f\left(z_{i,j}^t; \xi_{i,j}^t\right) \mid z_{i,j}^t\right] \stackrel{\text{Ass. 1.3}}{=} \nabla f\left(z_{i,j}^t\right), \tag{31}$$

which still holds for $j = 0$. Now, taking expectation conditionally on $(x^t)$ in both sides of (30) gives

$$\mathbb{E}\left[f\left(x^{t+1}\right) \mid x^t\right] \stackrel{(30)}{\leq} f\left(x^t\right) - \eta_g \mathbb{E}\left[\left\langle \nabla f\left(x^t\right), \sum_{i=1}^{n} \sum_{j=0}^{K-1} \nabla f\left(z_{i,j}^t; \xi_{i,j}^t\right) \right\rangle \Bigg| x^t\right] \tag{32}$$

$$+ \frac{\eta_g^2 L}{2} \mathbb{E}\left[\left\|\sum_{i=1}^{n} \sum_{j=0}^{K-1} \nabla f\left(z_{i,j}^t; \xi_{i,j}^t\right)\right\|^2 \Bigg| x^t\right],$$

and, for the inner product above we have

$$\mathbb{E}\left[\left\langle \nabla f\left(x^t\right), \sum_{i=1}^{n} \sum_{j=0}^{K-1} \nabla f\left(z_{i,j}^t; \xi_{i,j}^t\right) \right\rangle \Bigg| x^t\right]$$

$$= \sum_{i=1}^{n} \sum_{j=0}^{K-1} \mathbb{E}\left[\left\langle \nabla f\left(x^t\right), \nabla f\left(z_{i,j}^t; \xi_{i,j}^t\right)\right\rangle \mid x^t\right]$$

$$\stackrel{(a)}{=} \sum_{i=1}^{n} \sum_{j=0}^{K-1} \mathbb{E}\left[\mathbb{E}\left[\left\langle \nabla f\left(x^t\right), \nabla f\left(z_{i,j}^t; \xi_{i,j}^t\right)\right\rangle \mid x^t, z_{i,j}^t\right] \mid x^t\right] \tag{33}$$

$$\stackrel{(b)}{=} \sum_{i=1}^{n} \sum_{j=0}^{K-1} \mathbb{E}\left[\left\langle \nabla f\left(x^t\right), \mathbb{E}\left[\nabla f\left(z_{i,j}^t; \xi_{i,j}^t\right) \mid x^t, z_{i,j}^t\right]\right\rangle \mid x^t\right]$$

$$\stackrel{(30)}{=} \sum_{i=1}^{n} \sum_{j=0}^{K-1} \mathbb{E}\left[\left\langle \nabla f\left(x^t\right), \nabla f\left(z_{i,j}^t\right)\right\rangle \mid x^t\right],$$

where in (a) we use the tower property of expectation, in (b) we put the conditional expectation $\mathbb{E}\left[\cdot \mid x^t, z_{i,j}^t\right]$ inside the inner product since $\mathbb{E}\left[\nabla f\left(x^t\right) \mid x^t, z_{i,j}^t\right] = \nabla f\left(x^t\right)$.

Then, using identity (67) and Assumption 1.2 we obtain

$$\sum_{i=1}^{n} \sum_{j=0}^{K-1} \mathbb{E}\left[\left\langle \nabla f\left(x^t\right), \nabla f\left(z_{i,j}^t\right)\right\rangle \mid x^t\right]$$

$$\stackrel{(67)}{=} \sum_{i=1}^{n} \sum_{j=0}^{K-1} \left(\frac{1}{2}\left\|\nabla f\left(x^t\right)\right\|^2 + \frac{1}{2}\mathbb{E}\left[\left\|\nabla f\left(z_{i,j}^t\right)\right\|^2 \mid x^t\right] - \frac{1}{2}\mathbb{E}\left[\left\|\nabla f\left(z_{i,j}^t\right) - \nabla f\left(x^t\right)\right\|^2 \mid x^t\right]\right)$$

$$\stackrel{\text{Ass. 1.2}}{\geq} \sum_{i=1}^{n} \sum_{j=0}^{K-1} \left(\frac{1}{2}\left\|\nabla f\left(x^t\right)\right\|^2 + \frac{1}{2}\mathbb{E}\left[\left\|\nabla f\left(z_{i,j}^t\right)\right\|^2 \mid x^t\right] - \frac{L^2}{2}\mathbb{E}\left[\left\|z_{i,j}^t - x^t\right\|^2 \mid x^t\right]\right)$$

$$= \frac{nK}{2}\left\|\nabla f\left(x^t\right)\right\|^2 + \frac{1}{2}\sum_{i=1}^{n} \sum_{j=0}^{K-1} \mathbb{E}\left[\left\|\nabla f\left(z_{i,j}^t\right)\right\|^2 \mid x^t\right]$$

$$- \frac{L^2}{2}\sum_{i=1}^{n} \sum_{j=0}^{K-1} \frac{L^2}{2}\mathbb{E}\left[\left\|z_{i,j}^t - x^t\right\|^2 \mid x^t\right] \tag{34}$$

Next, we bound the squared norm (last term) in (32) as

$$
\mathbb{E}\left[\left\|\sum_{i=1}^{n}\sum_{j=0}^{K-1}\nabla f\left(z_{i,j}^{t};\xi_{i,j}^{t}\right)\right\|^{2}\ \middle|\ x^{t}\right]
$$

$$
=\mathbb{E}\left[\left\|\sum_{i=1}^{n}\sum_{j=0}^{K-1}\left(\nabla f\left(z_{i,j}^{t};\xi_{i,j}^{t}\right)-\nabla f\left(z_{i,j}^{t}\right)\right)+\sum_{i=1}^{n}\sum_{j=0}^{K-1}\nabla f\left(z_{i,j}^{t}\right)\right\|^{2}\ \middle|\ x^{t}\right]
$$

$$
\overset{\text{Lem. E.4}}{\leq}2\,\mathbb{E}\left[\left\|\sum_{i=1}^{n}\sum_{j=0}^{K-1}\left(\nabla f\left(z_{i,j}^{t};\xi_{i,j}^{t}\right)-\nabla f\left(z_{i,j}^{t}\right)\right)\right\|^{2}\ \middle|\ x^{t}\right]+2\,\mathbb{E}\left[\left\|\sum_{i=1}^{n}\sum_{j=0}^{K-1}\nabla f\left(z_{i,j}^{t}\right)\right\|^{2}\ \middle|\ x^{t}\right]
$$

$$
\overset{\text{Lem. E.6}}{\leq}2\,\mathbb{E}\left[\left\|\sum_{i=1}^{n}\sum_{j=0}^{K-1}\left(\nabla f\left(z_{i,j}^{t};\xi_{i,j}^{t}\right)-\nabla f\left(z_{i,j}^{t}\right)\right)\right\|^{2}\ \middle|\ x^{t}\right]+2nK\sum_{i=1}^{n}\sum_{j=0}^{K-1}\mathbb{E}\left[\left\|\nabla f\left(z_{i,j}^{t}\right)\right\|^{2}\ \middle|\ x^{t}\right],
$$

$$(35)$$

and for the remaining variance term above, we have

$$
2\,\mathbb{E}\left[\left\|\sum_{i=1}^{n}\sum_{j=0}^{K-1}\left(\nabla f\left(z_{i,j}^{t};\xi_{i,j}^{t}\right)-\nabla f\left(z_{i,j}^{t}\right)\right)\right\|^{2}\ \middle|\ x^{t}\right]
$$

$$
\overset{\text{(a)}}{=}2\sum_{i=1}^{n}\mathbb{E}\left[\left\|\sum_{j=0}^{K-1}\left(\nabla f\left(z_{i,j}^{t};\xi_{i,j}^{t}\right)-\nabla f\left(z_{i,j}^{t}\right)\right)\right\|^{2}\ \middle|\ x^{t}\right],\qquad(36)
$$

where in (a) we use the fact that, conditionally on $x^{t}$ the clients $1, 2, \ldots, n$ are working independently so notably this shows that conditionally on $x^{t}$ the random variables $\{X_{i}\}_{i\in[n]}$ where for any $i\in[n]$

$$
X_{i}:=\sum_{j=0}^{K-1}\left(\nabla f\left(z_{i,j}^{t};\xi_{i,j}^{t}\right)-\nabla f\left(z_{i,j}^{t}\right)\right),\qquad(37)
$$

are mutually independent hence $\mathbb{E}\left[\langle X_{i},X_{j}\rangle\mid x^{t}\right]=\langle\mathbb{E}\left[X_{i}\mid x^{t}\right],\mathbb{E}\left[X_{j}\mid x^{t}\right]\rangle$ and since they all have zero mean it follows

$$
\mathbb{E}\left[\left\|\sum_{i=1}^{n}X_{i}\right\|^{2}\ \middle|\ x^{t}\right]=\sum_{i=1}^{n}\mathbb{E}\left[\left\|X_{i}\right\|^{2}\ \middle|\ x^{t}\right]+2\sum_{1\leq i<j\leq n}\mathbb{E}\left[\langle X_{i},X_{j}\rangle\mid x^{t}\right]
$$

$$
=\sum_{i=1}^{n}\mathbb{E}\left[\left\|X_{i}\right\|^{2}\ \middle|\ x^{t}\right]+2\sum_{1\leq i<j\leq n}\langle\mathbb{E}\left[X_{i}\mid x^{t}\right],\mathbb{E}\left[X_{j}\mid x^{t}\right]\rangle
$$

$$
\overset{\text{(a)}}{=}\sum_{i=1}^{n}\mathbb{E}\left[\left\|X_{i}\right\|^{2}\ \middle|\ x^{t}\right],\qquad(38)
$$

where in (a) we use the identity

$$
\mathbb{E}\left[X_{i}\mid x^{t}\right]\overset{(37)}{=}\sum_{j=0}^{K-1}\mathbb{E}\left[\nabla f\left(z_{i,j}^{t};\xi_{i,j}^{t}\right)-\nabla f\left(z_{i,j}^{t}\right)\mid x^{t}\right]
$$

$$
\overset{\text{Lem. E.2}}{=}\sum_{j=0}^{K-1}\mathbb{E}\left[\mathbb{E}\left[\nabla f\left(z_{i,j}^{t};\xi_{i,j}^{t}\right)-\nabla f\left(z_{i,j}^{t}\right)\mid x^{t},z_{i,j}^{t}\right]\mid x^{t}\right]
$$

$$\overset{(31)}{=} \sum_{j=0}^{K-1} \mathbb{E}\left[\nabla f\left(z_{i,j}^t\right) - \nabla f\left(z_{i,j}^t\right) \mid x^t\right]$$

$$= 0,$$

which holds for all $i \in [n]$. Continuing from (36) and taking full expectation we obtain

$$2\mathbb{E}\left[\left\|\sum_{i=1}^n \sum_{j=0}^{K-1}\left(\nabla f\left(z_{i,j}^t;\xi_{i,j}^t\right) - \nabla f\left(z_{i,j}^t\right)\right)\right\|^2\right]$$

$$\overset{(36)+(38)}{=} 2\sum_{i=1}^n \sum_{j=0}^{K-1} \mathbb{E}\left[\left\|\nabla f\left(z_{i,j}^t;\xi_{i,j}^t\right) - \nabla f\left(z_{i,j}^t\right)\right\|^2\right]$$

$$+ 2\sum_{i=1}^n 2 \sum_{0 \le j < \ell \le K-1} \underbrace{\mathbb{E}\left[\left\langle \nabla f\left(z_{i,j}^t;\xi_{i,j}^t\right) - \nabla f\left(z_{i,j}^t\right), \nabla f\left(z_{i,\ell}^t;\xi_{i,\ell}^t\right) - \nabla f\left(z_{i,\ell}^t\right)\right\rangle\right]}_{:=A_{j,\ell}},$$

$$(39)$$

and to simplify the expectation term $A_{j,\ell}$ we use Corollary C.1. To do so, observe that $j < \ell$ hence conditionally on $z_{i,\ell}^t$, the random point $\xi_{i,\ell}^t$ is statistically independent from all past iterates and random points so in particular $\xi_{i,\ell}^t$ is independent from the pair $(z_{i,j}^t, \xi_{i,j}^t)$ hence, it follows that

$$\mathbb{E}\left[\nabla f\left(z_{i,\ell}^t;\xi_{i,\ell}^t\right) \mid z_{i,j}^t, \xi_{i,j}^t, z_{i,\ell}^t\right] = \mathbb{E}\left[\nabla f\left(z_{i,\ell}^t;\xi_{i,\ell}^t\right) \mid z_{i,\ell}^t\right] \overset{\text{Ass. 1.3}}{=} \nabla f\left(z_{i,\ell}^t\right), \quad (40)$$

and then, using the tower property of the expectation, we obtain

$$A_{j,\ell} = \mathbb{E}\left[\left\langle \nabla f\left(z_{i,j}^t;\xi_{i,j}^t\right) - \nabla f\left(z_{i,j}^t\right), \nabla f\left(z_{i,\ell}^t;\xi_{i,\ell}^t\right) - \nabla f\left(z_{i,\ell}^t\right)\right\rangle\right]$$

$$\overset{\text{Lem. E.2}}{=} \mathbb{E}\left[\mathbb{E}\left[\left\langle \nabla f\left(z_{i,j}^t;\xi_{i,j}^t\right) - \nabla f\left(z_{i,j}^t\right), \nabla f\left(z_{i,\ell}^t;\xi_{i,\ell}^t\right) - \nabla f\left(z_{i,\ell}^t\right)\right\rangle \mid z_{i,j}^t, \xi_{i,j}^t, z_{i,\ell}^t\right]\right]$$

$$\overset{(a)}{=} \mathbb{E}\left[\left\langle \nabla f\left(z_{i,j}^t;\xi_{i,j}^t\right) - \nabla f\left(z_{i,j}^t\right), \mathbb{E}\left[\nabla f\left(z_{i,\ell}^t;\xi_{i,\ell}^t\right) - \nabla f\left(z_{i,\ell}^t\right) \mid z_{i,j}^t, \xi_{i,j}^t, z_{i,\ell}^t\right]\right\rangle\right]$$

$$\overset{(40)}{=} \mathbb{E}\left[\left\langle \nabla f\left(z_{i,j}^t;\xi_{i,j}^t\right) - \nabla f\left(z_{i,j}^t\right), \mathbb{E}\left[\nabla f\left(z_{i,\ell}^t;\xi_{i,\ell}^t\right) \mid z_{i,\ell}^t\right] - \nabla f\left(z_{i,\ell}^t\right)\right\rangle\right]$$

$$\overset{\text{Ass. 1.3}}{=} 0,$$

where in (a) we use the fact that

$$\mathbb{E}\left[\nabla f\left(z_{i,j}^t;\xi_{i,j}^t\right) - \nabla f\left(z_{i,j}^t\right) \mid z_{i,j}^t, \xi_{i,j}^t, z_{i,\ell}^t\right] = \nabla f\left(z_{i,j}^t;\xi_{i,j}^t\right) - \nabla f\left(z_{i,j}^t\right),$$

which allows us to take the conditional expectation $\mathbb{E}\left[\cdot \mid z_{i,j}^t, \xi_{i,j}^t, z_{i,\ell}^t\right]$ inside the inner product. That being said, we can write

$$2\mathbb{E}\left[\left\|\sum_{i=1}^n \sum_{j=0}^{K-1}\left(\nabla f\left(z_{i,j}^t;\xi_{i,j}^t\right) - \nabla f\left(z_{i,j}^t\right)\right)\right\|^2 \mid x^t\right]$$

$$= 2\sum_{i=1}^n \sum_{j=0}^{K-1} \mathbb{E}\left[\left\|\nabla f\left(z_{i,j}^t;\xi_{i,j}^t\right) - \nabla f\left(z_{i,j}^t\right)\right\|^2 \mid x^t\right], \quad (41)$$

and taking full expectation in the above equality (41) yields, thanks to the tower property (Lemma E.2),

$$2\mathbb{E}\left[\left\|\sum_{i=1}^n \sum_{j=0}^{K-1}\left(\nabla f\left(z_{i,j}^t;\xi_{i,j}^t\right) - \nabla f\left(z_{i,j}^t\right)\right)\right\|^2\right]$$

$$\overset{(41)}{=} 2\sum_{i=1}^n \sum_{j=0}^{K-1} \mathbb{E}\left[\left\|\nabla f\left(z_{i,j}^t;\xi_{i,j}^t\right) - \nabla f\left(z_{i,j}^t\right)\right\|^2\right]$$

$$\overset{\text{Ass. 1.3}}{\le} 2\sum_{i=1}^n \sum_{j=0}^{K-1} \sigma^2$$

$$= 2\sigma^2 nK. \quad (42)$$

Finally combining (34), (35) and (42) and taking full expectation in (32) gives

$$\mathbb{E}\left[f\left(x^{t+1}\right)\right] \overset{(32)}{\leq} \mathbb{E}\left[f\left(x^t\right)\right] - \eta_g \mathbb{E}\left[\left\langle \nabla f\left(x^t\right), \sum_{i=1}^{n}\sum_{j=0}^{K-1} \nabla f\left(z_{i,j}^t; \xi_{i,j}^t\right)\right\rangle\right]$$

$$+ \frac{\eta_g^2 L}{2}\mathbb{E}\left[\left\|\sum_{i=1}^{n}\sum_{j=0}^{K-1} \nabla f\left(z_{i,j}^t; \xi_{i,j}^t\right)\right\|^2\right]$$

$$\overset{(34)+(35)+(42)}{\leq} \mathbb{E}\left[f\left(x^t\right)\right] - \frac{\eta_g n K}{2}\mathbb{E}\left[\left\|\nabla f\left(x^t\right)\right\|^2\right] - \frac{\eta_g}{2}\sum_{i=1}^{n}\sum_{k=0}^{K-1}\mathbb{E}\left[\left\|\nabla f\left(z_{i,j}^t\right)\right\|^2\right],$$

$$+ \frac{\eta_g L^2}{2}\sum_{i=1}^{n}\sum_{j=0}^{K-1}\mathbb{E}\left[\left\|z_{i,j}^t - x^t\right\|^2\right]$$

$$+ \eta_g^2 n L K\sum_{i=1}^{n}\sum_{j=0}^{K-1}\mathbb{E}\left[\left\|\nabla f\left(z_{i,j}^t\right)\right\|^2\right]$$

$$+ \eta_g^2 \sigma^2 n L K,$$

and reshuffling the above expression leads to the inequality

$$\mathbb{E}\left[f\left(x^{t+1}\right)\right] \leq \mathbb{E}\left[f\left(x^t\right)\right] - \frac{\eta_g n K}{2}\mathbb{E}\left[\left\|\nabla f\left(x^t\right)\right\|^2\right] - \eta_g\left(\frac{1}{2} - \eta_g n L K\right)\sum_{i=1}^{n}\sum_{k=0}^{K-1}\mathbb{E}\left[\left\|\nabla f\left(z_{i,j}^t\right)\right\|^2\right]$$

$$+ \frac{\eta_g L^2}{2}\sum_{i=1}^{n}\sum_{j=0}^{K-1}\mathbb{E}\left[\left\|z_{i,j}^t - x^t\right\|^2\right]$$

$$+ \eta_g^2 \sigma^2 n L K,$$

which proves the desired inequality. $\qquad\square$

### D.1.2 PROOF OF THE RESIDUAL ESTIMATION

**Lemma C.2.** *Under Assumption 1.3 the sequences of iterates $\{x^t\}_{t\geq 0}$ and $\{z_{i,j}^t\}_{t\geq 0}$ in Algorithms 4 and 5 satisfy for any integers $t \geq 0$, $i \in [n]$ and $j \in \{0, \ldots, K-1\}$*

$$\mathbb{E}\left[\left\|z_{i,j}^t - x^t\right\|^2\right] \leq 2\left(\sum_{\ell=0}^{j-1}\left(\eta_{i,\ell}^t\right)^2\right)\sum_{\ell=0}^{j-1}\mathbb{E}\left[\left\|\nabla f\left(z_{i,\ell}^t\right)\right\|^2\right] + 2\sigma^2\sum_{\ell=0}^{j-1}\left(\eta_{i,\ell}^t\right)^2.$$

*Proof.* Fix $t \geq 0$, $i \in [n]$ and $j \in \{0, \ldots, K-1\}$ then according to the update rule from the local steps we have

$$z_{i,\ell+1}^t = z_{i,\ell}^t - \eta_{i,\ell}^t \nabla f\left(z_{i,\ell}^t; \xi_{i,\ell}^t\right),$$

and if we unroll this equality from $j$ down to 0 we obtain for any $\ell \in \{0, \ldots, K-1\}$

$$z_{i,j}^t = z_{i,0}^t - \sum_{\ell=0}^{j-1}\eta_{i,\ell}^t \nabla f\left(z_{i,\ell}^t; \xi_{i,\ell}^t\right), \tag{43}$$

and using the fact that $z_{i,0}^t = x^t$ (see line 2 of Algorithms 4 and 5) we have

$$z_{i,j}^t - x^t \overset{(43)}{=} -\sum_{\ell=0}^{j-1}\eta_{i,\ell}^t \nabla f\left(z_{i,\ell}^t; \xi_{i,\ell}^t\right). \tag{44}$$

Now, taking expectation of the squared norm in (44) leads to

$$\mathbb{E}\left[\left\|z_{i,j}^t - x^t\right\|^2\right] = \mathbb{E}\left[\left\|\sum_{\ell=0}^{j-1}\eta_{i,\ell}^t \nabla f\left(z_{i,\ell}^t; \xi_{i,\ell}^t\right)\right\|^2\right]$$

$$= \mathbb{E}\left[\left\|\sum_{\ell=0}^{j-1} \eta_{i,\ell}^t \left(\nabla f\left(z_{i,\ell}^t; \xi_{i,\ell}^t\right) - \nabla f\left(z_{i,\ell}^t\right)\right) + \sum_{\ell=0}^{j-1} \eta_{i,\ell}^t \nabla f\left(z_{i,\ell}^t\right)\right\|^2\right]$$

$$\overset{(a)}{\leq} 2\,\mathbb{E}\left[\left\|\sum_{\ell=0}^{j-1} \eta_{i,\ell}^t \left(\nabla f\left(z_{i,\ell}^t; \xi_{i,\ell}^t\right) - \nabla f\left(z_{i,\ell}^t\right)\right)\right\|^2\right]$$

$$+ 2\,\mathbb{E}\left[\left\|\sum_{\ell=0}^{j-1} \eta_{i,\ell}^t \nabla f\left(z_{i,\ell}^t\right)\right\|^2\right], \tag{45}$$

where in (a) we use Young's inequality with $\alpha = 1$ (Lemma E.4). Now, we bound the variance term (first one) and the second term from (45). As for the variance term, similarly as we did in (39) (but in our case here up to $j$ instead of $K$) for the proof of the descent lemma we have

$$2\,\mathbb{E}\left[\left\|\sum_{\ell=0}^{j-1} \eta_{i,\ell}^t \left(\nabla f\left(z_{i,\ell}^t; \xi_{i,\ell}^t\right) - \nabla f\left(z_{i,\ell}^t\right)\right)\right\|^2\right]$$

$$= 2\sum_{\ell=0}^{j-1} \mathbb{E}\left[\left\|\eta_{i,\ell}^t \nabla f\left(z_{i,\ell}^t; \xi_{i,\ell}^t\right) - \nabla f\left(z_{i,\ell}^t\right)\right\|^2\right]$$

$$4\sum_{0\leq\ell<k\leq j-1} \left(\eta_{i,\ell}^t\right)^2 \mathbb{E}\left[\left\langle\nabla f\left(z_{i,\ell}^t; \xi_{i,\ell}^t\right) - \nabla f\left(z_{i,\ell}^t\right), \nabla f\left(z_{i,k}^t; \xi_{i,k}^t\right) - \nabla f\left(z_{i,k}^t\right)\right\rangle\right]$$

$$\overset{(39)}{=} 2\sum_{\ell=0}^{j-1} \left(\eta_{i,\ell}^t\right)^2 \mathbb{E}\left[\left\|\nabla f\left(z_{i,\ell}^t; \xi_{i,\ell}^t\right) - \nabla f\left(z_{i,\ell}^t\right)\right\|^2\right]$$

$$\overset{\text{Ass. }1.3}{\leq} 2\sum_{\ell=0}^{j-1} \left(\eta_{i,\ell}^t\right)^2 \sigma^2$$

$$= 2\sigma^2 \sum_{\ell=0}^{j-1} \left(\eta_{i,\ell}^t\right)^2. \tag{46}$$

Concerning the second term in (45) we use Lemma E.8 which leads to

$$2\,\mathbb{E}\left[\left\|\sum_{\ell=0}^{j-1} \eta_{i,\ell}^t \nabla f\left(z_{i,\ell}^t\right)\right\|^2\right] \overset{\text{Lem. E.8}}{\leq} 2\left(\sum_{\ell=0}^{j-1} \left(\eta_{i,\ell}^t\right)^2\right) \sum_{\ell=0}^{j-1} \mathbb{E}\left[\left\|\nabla f\left(z_{i,\ell}^t\right)\right\|^2\right], \tag{47}$$

and, combining bounds (46) and (47) gives

$$\mathbb{E}\left[\left\|z_{i,j}^t - x^t\right\|^2\right] \overset{(45)}{\leq} 2\,\mathbb{E}\left[\left\|\sum_{\ell=0}^{j-1} \eta_{i,\ell}^t \left(\nabla f\left(z_{i,\ell}^t; \xi_{i,\ell}^t\right) - \nabla f\left(z_{i,\ell}^t\right)\right)\right\|^2\right]$$

$$+ 2\,\mathbb{E}\left[\left\|\sum_{\ell=0}^{j-1} \eta_{i,\ell}^t \nabla f\left(z_{i,\ell}^t\right)\right\|^2\right]$$

$$\overset{(46)+(47)}{\leq} 2\left(\sum_{\ell=0}^{j-1} \left(\eta_{i,\ell}^t\right)^2\right) \sum_{\ell=0}^{j-1} \mathbb{E}\left[\left\|\nabla f\left(z_{i,\ell}^t\right)\right\|^2\right] + 2\sigma^2 \sum_{\ell=0}^{j-1} \left(\eta_{i,\ell}^t\right)^2,$$

as desired: this achieves the proof of the lemma. □

### D.1.3 PROOF OF LEMMA C.3

**Lemma C.3.** *Under Assumptions 1.2 and 1.3 the sequences of iterates $\{x^t\}_{t\geq 0}$ and $\{z_{i,j}^t\}_{t\geq 0}$ in Algorithms 4 and 5 satisfy for any integer $R \geq 1$*

$$\frac{1}{R}\sum_{t=0}^{R-1} \mathbb{E}\left[\left\|\nabla f\left(x^t\right)\right\|^2\right] \leq \frac{2\Delta}{\eta_g nKR} + 2\eta_g\sigma^2 L + \frac{2\sigma^2 L^2}{nR}\sum_{t=0}^{R-1}\sum_{i=1}^{n}\sum_{\ell=0}^{K-1} \left(\eta_{i,\ell}^t\right)^2$$

$$-\frac{2}{nKR}\sum_{t=0}^{R-1}\sum_{i=1}^{n}\left(\frac{1}{2}-\eta_g nLK-\left(\sum_{\ell=0}^{K-1}\left(\eta_{i,\ell}^t\right)^2\right)L^2K\right)\sum_{j=0}^{K-1}\mathbb{E}\left[\left\|\nabla f\left(z_{i,j}^t\right)\right\|^2\right].$$

*Proof.* First, for any integer $t \geq 0$ by Lemma C.1 we have

$$\mathbb{E}\left[f\left(x^{t+1}\right)\right]\leq\mathbb{E}\left[f\left(x^t\right)\right]-\frac{\eta_g nK}{2}\mathbb{E}\left[\left\|\nabla f\left(x^t\right)\right\|^2\right]$$

$$-\eta_g\left(\frac{1}{2}-\eta_g nLK\right)\sum_{i=1}^{n}\sum_{j=0}^{K-1}\mathbb{E}\left[\left\|\nabla f\left(z_{i,j}^t\right)\right\|^2\right]$$

$$+\frac{\eta_g L^2}{2}\sum_{i=1}^{n}\sum_{j=0}^{K-1}\mathbb{E}\left[\left\|z_{i,j}^t-x^t\right\|^2\right]$$

$$+\eta_g^2\sigma^2 nLK,$$

which after rearranging the terms and multiplying both sides by $\frac{2}{\eta_g nK}$ gives

$$\mathbb{E}\left[\left\|\nabla f\left(x^t\right)\right\|^2\right]\leq\frac{2}{\eta_g nK}\left(\mathbb{E}\left[f\left(x^t\right)\right]-\mathbb{E}\left[f\left(x^{t+1}\right)\right]\right)$$

$$-2\left(\frac{1}{2}-\eta_g nLK\right)\frac{1}{nK}\sum_{i=1}^{n}\sum_{j=0}^{K-1}\mathbb{E}\left[\left\|\nabla f\left(z_{i,j}^t\right)\right\|^2\right]$$

$$+\frac{L^2}{nK}\sum_{i=1}^{n}\sum_{j=0}^{K-1}\mathbb{E}\left[\left\|z_{i,j}^t-x^t\right\|^2\right]$$

$$+2\eta_g\sigma^2 L,$$

and using Lemma C.2 we obtain the inequality

$$\mathbb{E}\left[\left\|\nabla f\left(x^t\right)\right\|^2\right]\leq\frac{2}{\eta_g nK}\left(\mathbb{E}\left[f\left(x^t\right)\right]-\mathbb{E}\left[f\left(x^{t+1}\right)\right]\right)$$

$$-2\left(\frac{1}{2}-\eta_g nLK\right)\frac{1}{nK}\sum_{i=1}^{n}\sum_{j=0}^{K-1}\mathbb{E}\left[\left\|\nabla f\left(z_{i,j}^t\right)\right\|^2\right]$$

$$+\frac{2L^2}{nK}\sum_{i=1}^{n}\sum_{j=0}^{K-1}\left(\left(\sum_{\ell=0}^{j-1}\left(\eta_{i,\ell}^t\right)^2\right)\sum_{\ell=0}^{j-1}\mathbb{E}\left[\left\|\nabla f\left(z_{i,\ell}^t\right)\right\|^2\right]+\sigma^2\sum_{\ell=0}^{j-1}\left(\eta_{i,\ell}^t\right)^2\right)$$

$$+2\eta_g\sigma^2 L. \tag{48}$$

Now, we further upper bound the above expression, notably we have

$$\frac{2L^2}{nK}\sum_{i=1}^{n}\sum_{j=0}^{K-1}\left(\left(\sum_{\ell=0}^{j-1}\left(\eta_{i,\ell}^t\right)^2\right)\sum_{\ell=0}^{j-1}\mathbb{E}\left[\left\|\nabla f\left(z_{i,\ell}^t\right)\right\|^2\right]+\sigma^2\sum_{\ell=0}^{j-1}\left(\eta_{i,\ell}^t\right)^2\right)$$

$$\overset{(a)}{\leq}\frac{2L^2}{nK}\sum_{i=1}^{n}\left(\sum_{\ell=0}^{K-1}\left(\eta_{i,\ell}^t\right)^2\right)\sum_{j=0}^{K-1}\sum_{\ell=0}^{j-1}\mathbb{E}\left[\left\|\nabla f\left(z_{i,\ell}^t\right)\right\|^2\right]+\frac{2\sigma^2 L^2}{n}\sum_{i=1}^{n}\sum_{\ell=0}^{K-1}\left(\eta_{i,\ell}^t\right)^2$$

where in (a) we use the fact that $j \leq K$ to upper bound the two sums over the local step sizes. Moreover, using again the fact that $j \leq K$ to upper bound the sum over $\ell$ of the squared norm of the gradients at $z_{i,\ell}^t$ we obtain

$$\frac{2L^2}{nK}\sum_{i=1}^{n}\left(\sum_{\ell=0}^{K-1}\left(\eta_{i,\ell}^t\right)^2\right)\sum_{j=0}^{K-1}\sum_{\ell=0}^{j-1}\mathbb{E}\left[\left\|\nabla f\left(z_{i,\ell}^t\right)\right\|^2\right]+\frac{2\sigma^2 L^2}{n}\sum_{i=1}^{n}\sum_{\ell=0}^{K-1}\left(\eta_{i,\ell}^t\right)^2$$

$$\leq\frac{2L^2 K}{nK}\sum_{i=1}^{n}\left(\sum_{\ell=0}^{K-1}\left(\eta_{i,\ell}^t\right)^2\right)\sum_{\ell=0}^{K-1}\mathbb{E}\left[\left\|\nabla f\left(z_{i,\ell}^t\right)\right\|^2\right]+\frac{2\sigma^2 L^2}{n}\sum_{i=1}^{n}\sum_{\ell=0}^{K-1}\left(\eta_{i,\ell}^t\right)^2,$$

hence injecting this bound in (48) yields

$$\mathbb{E}\left[\left\|\nabla f\left(x^t\right)\right\|^2\right] \overset{(48)}{\leq} \frac{2}{\eta_g nK}\left(\mathbb{E}\left[f\left(x^t\right)\right] - \mathbb{E}\left[f\left(x^{t+1}\right)\right]\right)$$

$$- 2\left(\frac{1}{2} - \eta_g nLK\right)\frac{1}{nK}\sum_{i=1}^{n}\sum_{j=0}^{K-1}\mathbb{E}\left[\left\|\nabla f\left(z_{i,j}^t\right)\right\|^2\right]$$

$$+ \frac{2L^2 K}{nK}\sum_{i=1}^{n}\left(\sum_{\ell=0}^{K-1}\left(\eta_{i,\ell}^t\right)^2\right)\sum_{\ell=0}^{K-1}\mathbb{E}\left[\left\|\nabla f\left(z_{i,\ell}^t\right)\right\|^2\right]$$

$$+ 2\eta_g\sigma^2 L + \frac{2\sigma^2 L^2}{n}\sum_{i=1}^{n}\sum_{\ell=0}^{K-1}\left(\eta_{i,\ell}^t\right)^2$$

$$\overset{(a)}{=} \frac{2}{\eta_g nK}\left(\mathbb{E}\left[f\left(x^t\right)\right] - \mathbb{E}\left[f\left(x^{t+1}\right)\right]\right) + 2\eta_g\sigma^2 L + \frac{2\sigma^2 L^2}{n}\sum_{i=1}^{n}\sum_{\ell=0}^{K-1}\left(\eta_{i,\ell}^t\right)^2$$

$$- \frac{2}{nK}\sum_{i=1}^{n}\left(\frac{1}{2} - \eta_g nLK - \left(\sum_{\ell=0}^{K-1}\left(\eta_{i,\ell}^t\right)^2\right)L^2 K\right)\sum_{j=0}^{K-1}\mathbb{E}\left[\left\|\nabla f\left(z_{i,j}^t\right)\right\|^2\right].$$

where in (a) we merged the two sums with the gradient terms. It remains to sum this inequality over $t \in \{0, \ldots, R-1\}$ for a fixed integer $R \geq 1$, this gives

$$\frac{1}{R}\sum_{t=0}^{R-1}\mathbb{E}\left[\left\|\nabla f\left(x^t\right)\right\|^2\right] \leq \frac{2}{\eta_g nKR}\left(f\left(x^0\right) - \mathbb{E}\left[f\left(x^R\right)\right]\right) + 2\eta_g\sigma^2 L + \frac{2\sigma^2 L^2}{nR}\sum_{t=0}^{R-1}\sum_{i=1}^{n}\sum_{\ell=0}^{K-1}\left(\eta_{i,\ell}^t\right)^2$$

$$- \frac{2}{nKR}\sum_{t=0}^{R-1}\sum_{i=1}^{n}\left(\frac{1}{2} - \eta_g nLK - \left(\sum_{\ell=0}^{K-1}\left(\eta_{i,\ell}^t\right)^2\right)L^2 K\right)\sum_{j=0}^{K-1}\mathbb{E}\left[\left\|\nabla f\left(z_{i,j}^t\right)\right\|^2\right]$$

$$\overset{\text{Ass. 1.2}}{\leq} \frac{2\Delta}{\eta_g nKR} + 2\eta_g\sigma^2 L + \frac{2\sigma^2 L^2}{nR}\sum_{t=0}^{R-1}\sum_{i=1}^{n}\sum_{\ell=0}^{K-1}\left(\eta_{i,\ell}^t\right)^2$$

$$- \frac{2}{nKR}\sum_{t=0}^{R-1}\sum_{i=1}^{n}\left(\frac{1}{2} - \eta_g nLK - \left(\sum_{\ell=0}^{K-1}\left(\eta_{i,\ell}^t\right)^2\right)L^2 K\right)\sum_{j=0}^{K-1}\mathbb{E}\left[\left\|\nabla f\left(z_{i,j}^t\right)\right\|^2\right],$$

and this establishes the desired inequality. $\qquad\square$

### D.2 CONVEX SETUP

#### D.2.1 PROOF OF THE DESCENT LEMMA

**Lemma C.4.** *Under Assumptions 1.2 to 1.4 the sequences of iterates $\{x^t\}_{t\geq 0}$ and $\{z_{i,j}^t\}$ in Algorithms 4 and 5 satisfy for any integer $t \geq 0$*

$$\mathbb{E}\left[\left\|x^{t+1} - x^*\right\|^2\right] \leq \mathbb{E}\left[\left\|x^t - x^*\right\|^2\right] + 2\eta_g L\sum_{i=1}^{n}\sum_{j=0}^{K-1}\mathbb{E}\left[\left\|z_{i,j}^t - x^*\right\|^2\right]$$

$$- 2\eta_g\left(\frac{3}{4} - \eta_g nLK\right)\sum_{i=1}^{n}\sum_{j=0}^{K-1}\mathbb{E}\left[\left\langle\nabla f\left(z_{i,j}^t\right), z_{i,j}^t - x^*\right\rangle\right] + 2\eta_g^2\sigma^2 nK.$$

*Proof.* Let $k \geq 0$ and $x^* \in \arg\min_{x\in\mathbb{R}^d} f(x)$ (which exists by Assumption 1.1), then we have

$$\left\|x^{t+1} - x^*\right\|^2 = \left\|\left(x^{t+1} - x^t\right) + \left(x^t - x^*\right)\right\|^2$$

$$= \left\|x^{t+1} - x^t\right\|^2 + 2\left\langle x^{t+1} - x^t, x^t - x^*\right\rangle + \left\|x^t - x^*\right\|^2$$

$$\overset{(a)}{=} \left\|x^{t+1} - x^t\right\|^2 - 2\eta_g\left\langle\sum_{i=1}^{n}\sum_{j=0}^{K-1}\nabla f\left(z_{i,j}^t; \xi_{i,j}^t\right), x^t - x^*\right\rangle + \left\|x^t - x^*\right\|^2$$

$$\stackrel{(b)}{=} \eta_g^2 \left\| \sum_{i=1}^{n} \sum_{j=0}^{K-1} \nabla f \left( z_{i,j}^t; \xi_{i,j}^t \right) \right\|^2 - 2\eta_g \left\langle \sum_{i=1}^{n} \sum_{j=0}^{K-1} \nabla f \left( z_{i,j}^t; \xi_{i,j}^t \right), x^t - x^* \right\rangle$$

$$+ \left\| x^t - x^* \right\|^2, \tag{49}$$

in (a) and (b) we use the relation

$$x^{t+1} = x^t - \eta_g \sum_{i=1}^{n} \sum_{j=0}^{K-1} \nabla f \left( z_{i,j}^t; \xi_{i,j}^t \right),$$

from lines 8 and 9 of Algorithm 4 and Algorithm 5 respectively. As we did earlier in the proof of the descent lemma (see Lemma C.4), by Corollary C.1, we know that conditionally on $z_{i,j}^t$, the random point $\xi_{i,j}^t$ is independent from all the past iterates and randomness so in particular, it is independent from $x^t$ (unless $j = 0$ because $z_{i,0}^t = x^t$). Hence by Corollary C.1 we have

$$\mathbb{E} \left[ \nabla f \left( z_{i,j}^t; \xi_{i,j}^t \right) \mid x^t, z_{i,j}^t \right] = \mathbb{E} \left[ \nabla f \left( z_{i,j}^t; \xi_{i,j}^t \right) \mid z_{i,j}^t \right] \stackrel{\text{Ass. 1.3}}{=} \nabla f \left( z_{i,j}^t \right), \tag{50}$$

which still holds for $j = 0$. Now, taking expectation conditionally on $(x^t)$ in both sides of (49) gives

$$\mathbb{E} \left[ \left\| x^{t+1} - x^* \right\|^2 \mid x^t \right] \stackrel{(49)}{\leq} \left\| x^t - x^* \right\|^2 - 2\eta_g \mathbb{E} \left[ \left\langle \sum_{i=1}^{n} \sum_{j=0}^{K-1} \nabla f \left( z_{i,j}^t; \xi_{i,j}^t \right), x^t - x^* \right\rangle \, \middle| \, x^t \right] \tag{51}$$

$$+ \eta_g^2 \mathbb{E} \left[ \left\| \sum_{i=1}^{n} \sum_{j=0}^{K-1} \nabla f \left( z_{i,j}^t; \xi_{i,j}^t \right) \right\|^2 \, \middle| \, x^t \right],$$

and, for the inner product above, the same way as we did in Lemma C.4 (see (33)) we obtain

$$\mathbb{E} \left[ \left\langle \sum_{i=1}^{n} \sum_{j=0}^{K-1} \nabla f \left( z_{i,j}^t; \xi_{i,j}^t \right), x^t - x^* \right\rangle \, \middle| \, x^t \right] = \sum_{i=1}^{n} \sum_{j=0}^{K-1} \mathbb{E} \left[ \left\langle \nabla f \left( z_{i,j}^t \right), x^t - x^* \right\rangle \mid x^t \right], \tag{52}$$

while, for the variance term in (51), still following what we did in the proof of Lemma C.4, we have

$$\mathbb{E} \left[ \left\| \sum_{i=1}^{n} \sum_{j=0}^{K-1} \nabla f \left( z_{i,j}^t; \xi_{i,j}^t \right) \right\|^2 \, \middle| \, x^t \right] \leq 2nK \sum_{i=1}^{n} \sum_{j=0}^{K-1} \mathbb{E} \left[ \left\| \nabla f \left( z_{i,j}^t \right) \right\|^2 \mid x^t \right] + 2\sigma^2 nK. \tag{53}$$

Hence, combining both (52) and (53) and injecting in (51) leads to

$$\mathbb{E} \left[ \left\| x^{t+1} - x^* \right\|^2 \mid x^t \right]$$

$$\stackrel{(51)}{\leq} \left\| x^t - x^* \right\|^2 - 2\eta_g \mathbb{E} \left[ \left\langle \sum_{i=1}^{n} \sum_{j=0}^{K-1} \nabla f \left( z_{i,j}^t; \xi_{i,j}^t \right), x^t - x^* \right\rangle \, \middle| \, x^t \right]$$

$$+ \eta_g^2 \mathbb{E} \left[ \left\| \sum_{i=1}^{n} \sum_{j=0}^{K-1} \nabla f \left( z_{i,j}^t; \xi_{i,j}^t \right) \right\|^2 \, \middle| \, x^t \right]$$

$$\stackrel{(52)+(53)}{\leq} \left\| x^t - x^* \right\|^2 - 2\eta_g \sum_{i=1}^{n} \sum_{j=0}^{K-1} \mathbb{E} \left[ \left\langle \nabla f \left( z_{i,j}^t \right), x^t - x^* \right\rangle \mid x^t \right]$$

$$+ 2\eta_g^2 nK \sum_{i=1}^{n} \sum_{j=0}^{K-1} \mathbb{E} \left[ \left\| \nabla f \left( z_{i,j}^t \right) \right\|^2 \, \middle| \, x^t \right] + 2\eta_g^2 \sigma^2 nK. \tag{54}$$

Next, for any $i \in [n]$ and any $j \in \{0, \dots, K-1\}$, to upper bound the gradient term $\eta_g^2 \left\| \nabla f \left( z_{i,j}^t \right) \right\|^2$ we use Lemma E.11 with $x = z_{i,j}^t$ and $y^* = x^* \in \arg\min_{x \in \mathbb{R}^d} f(x)$ which leads to

$$\eta_g^2 \left\| \nabla f \left( z_{i,j}^t \right) \right\|^2 \leq \eta_g^2 L \left\langle \nabla f \left( z_{i,j}^t \right), z_{i,j}^t - x^* \right\rangle. \tag{55}$$

Then, it remains to upper bound the inner products $-2\eta_g \left\langle \nabla f\left(z_{i,j}^t\right), x^t - x^* \right\rangle$ for any $i \in [n]$ and any $j \in \{0, \dots, K-1\}$. To do so, we split it in two as follows

$$-2\eta_g \left\langle \nabla f\left(z_{i,j}^t\right), x^t - x^* \right\rangle = -2\eta_g \left\langle \nabla f\left(z_{i,j}^t\right), x^t - z_{i,j}^t \right\rangle - 2\eta_g \left\langle \nabla f\left(z_{i,j}^t\right), z_{i,j}^t - x^* \right\rangle, \quad (56)$$

and injecting the upper bound (55) and equality (56) in (54) gives

$$
\mathbb{E}\left[ \left\| x^{t+1} - x^* \right\|^2 \mid x^t \right]
$$

$$
\overset{(55)+(56)}{\leq} \left\| x^t - x^* \right\|^2 - 2\eta_g \sum_{i=1}^n \sum_{j=0}^{K-1} \mathbb{E}\left[ \left\langle \nabla f\left(z_{i,j}^t\right), x^t - z_{i,j}^t \right\rangle \mid x^t \right]
$$

$$
- 2\eta_g \sum_{i=1}^n \sum_{j=0}^{K-1} \mathbb{E}\left[ \left\langle \nabla f\left(z_{i,j}^t\right), z_{i,j}^t - x^* \right\rangle \mid x^t \right]
$$

$$
+ 2\eta_g^2 nLK \sum_{i=1}^n \sum_{j=0}^{K-1} \mathbb{E}\left[ \left\langle \nabla f\left(z_{i,j}^t\right), z_{i,j}^t - x^* \right\rangle \mid x^t \right] + 2\eta_g^2 \sigma^2 nK
$$

$$
= \left\| x^t - x^* \right\|^2 - 2\eta_g \sum_{i=1}^n \sum_{j=0}^{K-1} \mathbb{E}\left[ \left\langle \nabla f\left(z_{i,j}^t\right), x^t - z_{i,j}^t \right\rangle \mid x^t \right]
$$

$$
- 2\eta_g \left(1 - \eta_g nLK\right) \sum_{i=1}^n \sum_{j=0}^{K-1} \mathbb{E}\left[ \left\langle \nabla f\left(z_{i,j}^t\right), z_{i,j}^t - x^* \right\rangle \mid x^t \right] + 2\eta_g^2 \sigma^2 nK. \quad (57)
$$

It remains to upper bound the inner product $-2\eta_g \left\langle \nabla f\left(z_{i,j}^t\right), x^t - z_{i,j}^t \right\rangle$ for any $i \in [n]$ and any $j \in \{0, \dots, K-1\}$. To do so, we will use the Young's inequality (Lemma E.5) with the scalar $\alpha = 1/2L$ and then, we will use Lemma E.11 with, again, $x = z_{i,j}^t$ and $y^\star = x^* \in \arg\min_{x \in \mathbb{R}^d} f(x)$. We have

$$
-2\eta_g \left\langle \nabla f\left(z_{i,j}^t\right), x^t - z_{i,j}^t \right\rangle \leq 2\eta_g \left| \left\langle \nabla f\left(z_{i,j}^t\right), x^t - z_{i,j}^t \right\rangle \right|
$$

$$
\overset{\text{Lem. E.5}}{\leq} \eta_g \left( \alpha \left\| \nabla f\left(z_{i,j}^t\right) \right\|^2 + \frac{1}{\alpha} \left\| x^t - z_{i,j}^t \right\|^2 \right)
$$

$$
\overset{(a)}{=} \eta_g \left( \frac{1}{2L} \left\| \nabla f\left(z_{i,j}^t\right) \right\|^2 + 2L \left\| x^t - z_{i,j}^t \right\|^2 \right)
$$

$$
\overset{\text{Lem. E.11}}{\leq} \eta_g \left( \frac{1}{2} \left\langle \nabla f\left(z_{i,j}^t\right), z_{i,j}^t - x^* \right\rangle + 2L \left\| x^t - z_{i,j}^t \right\|^2 \right)
$$

$$
= \frac{\eta_g}{2} \left\langle \nabla f\left(z_{i,j}^t\right), z_{i,j}^t - x^* \right\rangle + 2\eta_g L \left\| x^t - z_{i,j}^t \right\|^2, \quad (58)
$$

where in (a) we use $\alpha = \frac{1}{2L} > 0$. Thus, injecting the previous bounds in (57) for all $i \in [n]$ and all $j \in \{0, \dots, K-1\}$ we obtain

$$
\mathbb{E}\left[ \left\| x^{t+1} - x^* \right\|^2 \mid x^t \right]
$$

$$\overset{(57)}{\leq} \left\|x^t - x^*\right\|^2 - 2\eta_g \sum_{i=1}^{n} \sum_{j=0}^{K-1} \mathbb{E}\left[\left\langle \nabla f\left(z_{i,j}^t\right), x^t - z_{i,j}^t\right\rangle \mid x^t\right]$$

$$- 2\eta_g\left(1 - \eta_g nLK\right) \sum_{i=1}^{n} \sum_{j=0}^{K-1} \mathbb{E}\left[\left\langle \nabla f\left(z_{i,j}^t\right), z_{i,j}^t - x^*\right\rangle \mid x^t\right] + 2\eta_g^2 \sigma^2 nK$$

$$\overset{(58)}{\leq} \left\|x^t - x^*\right\|^2 + \frac{\eta_g}{2} \sum_{i=1}^{n} \sum_{j=0}^{K-1} \mathbb{E}\left[\left\langle \nabla f\left(z_{i,j}^t\right), z_{i,j}^t - x^*\right\rangle \mid x^t\right]$$

$$+ 2\eta_g L \sum_{i=1}^{n} \sum_{j=0}^{K-1} \mathbb{E}\left[\left\|z_{i,j}^t - x^*\right\|^2 \mid x^t\right]$$

$$- 2\eta_g\left(1 - \eta_g nLK\right) \sum_{i=1}^{n} \sum_{j=0}^{K-1} \mathbb{E}\left[\left\langle \nabla f\left(z_{i,j}^t\right), z_{i,j}^t - x^*\right\rangle \mid x^t\right] + 2\eta_g^2 \sigma^2 nK,$$

and rearranging the above right-hand side leads to the inequality

$$\mathbb{E}\left[\left\|x^{t+1} - x^*\right\|^2 \mid x^t\right]$$

$$\leq \left\|x^t - x^*\right\|^2 + 2\eta_g L \sum_{i=1}^{n} \sum_{j=0}^{K-1} \mathbb{E}\left[\left\|z_{i,j}^t - x^*\right\|^2 \mid x^t\right]$$

$$- 2\eta_g\left(\frac{3}{4} - \eta_g nLK\right) \sum_{i=1}^{n} \sum_{j=0}^{K-1} \mathbb{E}\left[\left\langle \nabla f\left(z_{i,j}^t\right), z_{i,j}^t - x^*\right\rangle \mid x^t\right] + 2\eta_g^2 \sigma^2 nK, \quad (59)$$

then, taking full expectation in both sides of (59) along with the tower property (Lemma E.2) leads to the desired inequality. $\qquad\square$

### D.2.2 PROOF OF THE DESCENT LEMMA ON $\{f(z_{i,j}^t)\}$

**Lemma C.5.** *Under Assumptions 1.2 to 1.4 the sequences of iterates $\{x^t\}_{t\geq 0}$ and $\{z_{i,j}^t\}$ in Algorithms 4 and 5 satisfy for any integer $t \geq 0$*

$$\frac{\eta_g}{2} \sum_{i=1}^{n} \sum_{j=0}^{K-1} \left(\mathbb{E}\left[f\left(z_{i,j}^t\right)\right] - f^{\mathrm{inf}}\right)$$

$$\leq \mathbb{E}\left[\left\|x^t - x^*\right\|^2\right] - \mathbb{E}\left[\left\|x^{t+1} - x^*\right\|^2\right] + 2\eta_g L \sum_{i=1}^{n} \sum_{j=0}^{K-1} \mathbb{E}\left[\left\|z_{i,j}^t - x^*\right\|^2\right]$$

$$- 2\eta_g\left(\frac{1}{2} - \eta_g nLK\right) \sum_{i=1}^{n} \sum_{j=0}^{K-1} \mathbb{E}\left[\left\langle \nabla f\left(z_{i,j}^t\right), z_{i,j}^t - x^*\right\rangle\right] + 2\eta_g^2 \sigma^2 nK.$$

*Proof.* Starting from our previous descent lemma (Lemma C.4), for any integer $t \geq 0$

$$\mathbb{E}\left[\left\|x^{t+1} - x^*\right\|^2\right] \leq \mathbb{E}\left[\left\|x^t - x^*\right\|^2\right] + 2\eta_g L \sum_{i=1}^{n} \sum_{j=0}^{K-1} \mathbb{E}\left[\left\|z_{i,j}^t - x^*\right\|^2\right]$$

$$- 2\eta_g\left(\frac{3}{4} - \eta_g nLK\right) \sum_{i=1}^{n} \sum_{j=0}^{K-1} \mathbb{E}\left[\left\langle \nabla f\left(z_{i,j}^t\right), z_{i,j}^t - x^*\right\rangle\right] + 2\eta_g^2 \sigma^2 nK,$$

$$(60)$$

and, using Lemma E.9, inequality (70), for $x = z_{i,j}^t$ and $x^\star = x^* \in \arg\min_{x \in \mathbb{R}^d} f(x)$, with the fact that $\frac{\eta_g}{2} > 0$ we obtain

$$-\frac{\eta_g}{2} \left\langle \nabla f\left(z_{i,j}^t\right), z_{i,j}^t - x^*\right\rangle \leq -\frac{\eta_g}{2}\left(f\left(z_{i,j}^t\right) - f^{\mathrm{inf}}\right), \quad (61)$$

and, injecting the bound (61) in (60) leads to

$$\mathbb{E}\left[\left\|x^{t+1} - x^*\right\|^2\right] \leq \mathbb{E}\left[\left\|x^t - x^*\right\|^2\right] + 2\eta_g L \sum_{i=1}^{n} \sum_{j=0}^{K-1} \mathbb{E}\left[\left\|z_{i,j}^t - x^*\right\|^2\right]$$

$$- 2\eta_g \left(\frac{1}{2} - \eta_g nLK\right) \sum_{i=1}^{n} \sum_{j=0}^{K-1} \mathbb{E}\left[\left\langle \nabla f\left(z_{i,j}^t\right), z_{i,j}^t - x^*\right\rangle\right]$$

$$- \frac{\eta_g}{2} \sum_{i=1}^{n} \sum_{j=0}^{K-1} \left(\mathbb{E}\left[f\left(z_{i,j}^t\right)\right] - f^{\inf}\right) + 2\eta_g^2 \sigma^2 nK,$$

which, after reshuffling the above expression, gives the desired inequality. $\qquad\square$

### D.2.3 PROOF OF THE DESCENT LEMMA ON $\{f(x^t)\}$

**Lemma C.6.** *Under Assumptions 1.2 to 1.4 the sequences of iterates $\{x^t\}_{t\geq 0}$ and $\{z_{i,j}^t\}$ in Algorithms 4 and 5 satisfy for any integer $t \geq 0$*

$$\frac{\eta_g}{4}\left(\mathbb{E}\left[f\left(x^t\right)\right] - f^{\inf}\right)$$

$$\leq \frac{1}{nK}\left(\mathbb{E}\left[\left\|x^t - x^*\right\|^2\right] - \mathbb{E}\left[\left\|x^{t+1} - x^*\right\|^2\right]\right) + \frac{5\eta_g L}{2nK} \sum_{i=1}^{n} \sum_{j=0}^{K-1} \mathbb{E}\left[\left\|z_{i,j}^t - x^*\right\|^2\right]$$

$$- 2\eta_g \left(\frac{1}{2} - \eta_g nLK\right) \frac{1}{nK} \sum_{i=1}^{n} \sum_{j=0}^{K-1} \mathbb{E}\left[\left\langle \nabla f\left(z_{i,j}^t\right), z_{i,j}^t - x^*\right\rangle\right] + 2\eta_g^2 \sigma^2.$$

*Proof.* From the previous descent lemma (Lemma C.5), we can rewrite the left-hand side as

$$\frac{\eta_g}{2} \sum_{i=1}^{n} \sum_{j=0}^{K-1} \left(\mathbb{E}\left[f\left(z_{i,j}^t\right)\right] - f^{\inf}\right)$$

$$= \frac{\eta_g nK}{2}\left(\mathbb{E}\left[f\left(x^t\right)\right] - f^{\inf}\right) + \frac{\eta_g}{2} \sum_{i=1}^{n} \sum_{j=0}^{K-1} \left(\mathbb{E}\left[f\left(z_{i,j}^t\right)\right] - f\left(x^t\right)\right), \qquad (62)$$

and, for any $i \in [n]$ and any $j \in \{0, \ldots, K-1\}$, using Lemma E.12 on the last term of (62) for $x = x^t$ and $y = z_{i,j}^t$, with the fact that $\frac{\eta_g}{2} > 0$ we obtain

$$\frac{\eta_g}{2}\left(f\left(x^t\right) - f\left(z_{i,j}^t\right)\right) \leq \frac{\eta_g}{4}\left(f\left(x^t\right) - f^{\inf}\right) + \frac{\eta_g L}{2}\left\|x^t - z_{i,j}^t\right\|^2, \qquad (63)$$

hence

$$\frac{\eta_g}{2} \sum_{i=1}^{n} \sum_{j=0}^{K-1} \left(\mathbb{E}\left[f\left(x^t\right) - f\left(z_{i,j}^t\right)\right]\right)$$

$$\overset{(63)}{\leq} \frac{\eta_g nK}{4}\left(\mathbb{E}\left[f\left(x^t\right)\right] - f^{\inf}\right) + \frac{\eta_g L}{2} \sum_{i=1}^{n} \sum_{j=0}^{K-1} \mathbb{E}\left[\left\|x^t - z_{i,j}^t\right\|^2\right], \qquad (64)$$

and, injecting (62) with the inequality (64) in Lemma C.5 gives

$$\frac{\eta_g nK}{2}\left(\mathbb{E}\left[f\left(x^t\right)\right] - f^{\inf}\right)$$

$$\leq \mathbb{E}\left[\left\|x^t - x^*\right\|^2\right] - \mathbb{E}\left[\left\|x^{t+1} - x^*\right\|^2\right] + \frac{5\eta_g L}{2} \sum_{i=1}^{n} \sum_{j=0}^{K-1} \mathbb{E}\left[\left\|z_{i,j}^t - x^*\right\|^2\right]$$

$$- 2\eta_g \left(\frac{1}{2} - \eta_g nLK\right) \sum_{i=1}^{n} \sum_{j=0}^{K-1} \mathbb{E}\left[\left\langle \nabla f\left(z_{i,j}^t\right), z_{i,j}^t - x^*\right\rangle\right] + 2\eta_g^2 \sigma^2 nK$$

$$+ \frac{\eta_g n K}{4} \left( \mathbb{E}\left[ f\left( x^t \right) \right] - f^{\mathrm{inf}} \right),$$

which, after reshuffling the above expression and dividing both sides by $nK$, gives the desired inequality. $\qquad\square$

### D.2.4 PROOF OF THE RESIUAL ESTIMATION

**Lemma C.7.** *Under Assumptions 1.3 and 1.4 the sequences of iterates $\{x^t\}_{t \geq 0}$ and $\{z_{i,j}^t\}_{t \geq 0}$ in Algorithms 4 and 5 satisfy for any integers $t \geq 0$, $i \in [n]$ and $j \in \{0, \ldots, K-1\}$*

$$\mathbb{E}\left[ \left\| z_{i,j}^t - x^t \right\|^2 \right] \leq 2L \left( \sum_{\ell=0}^{j-1} \left( \eta_{i,\ell}^t \right)^2 \right) \sum_{\ell=0}^{j-1} \mathbb{E}\left[ \left\langle \nabla f\left( z_{i,\ell}^t \right), z_{i,\ell}^t - x^* \right\rangle \right] + 2\sigma^2 \sum_{\ell=0}^{j-1} \left( \eta_{i,\ell}^t \right)^2.$$

*Proof.* According to Assumption 1.3 we can apply our first residual estimation (Lemma C.2) hence, for any $t \geq 0$, any $i \in [n]$ and any $j \in \{0, \ldots, K-1\}$ we have

$$\mathbb{E}\left[ \left\| z_{i,j}^t - x^t \right\|^2 \right] \leq 2 \left( \sum_{\ell=0}^{j-1} \left( \eta_{i,\ell}^t \right)^2 \right) \sum_{\ell=0}^{j-1} \mathbb{E}\left[ \left\| \nabla f\left( z_{i,\ell}^t \right) \right\|^2 \right] + 2\sigma^2 \sum_{\ell=0}^{j-1} \left( \eta_{i,\ell}^t \right)^2. \qquad (65)$$

Now, thanks to the convexity of $f$ (Assumption 1.4) and Assumption 1.2 we can use Lemma E.11 on each squared norm $\|\nabla f(z_{i,\ell}^t)\|^2$ for $\ell \in \{0, \ldots, K-1\}$. Choosing $y^\star = x^* \in \arg\min_{x \in \mathbb{R}^d} f(x)$ in Lemma E.11 and $x$ as the corresponding argument appearing in all gradients from (65), we obtain the new upper bound

$$\mathbb{E}\left[ \left\| z_{i,j}^t - x^t \right\|^2 \right] \leq 2L \left( \sum_{\ell=0}^{j-1} \left( \eta_{i,\ell}^t \right)^2 \right) \sum_{\ell=0}^{j-1} \mathbb{E}\left[ \left\langle \nabla f\left( z_{i,\ell}^t \right), z_{i,\ell}^t - x^* \right\rangle \right] + 2\sigma^2 \sum_{\ell=0}^{j-1} \left( \eta_{i,\ell}^t \right)^2,$$

as stated. $\qquad\square$

### D.2.5 PROOF OF LEMMA C.8

**Lemma C.8.** *Under Assumptions 1.2 to 1.4 the sequences of iterates $\{x^t\}_{t \geq 0}$ and $\{z_{i,j}^t\}_{t \geq 0}$ in Algorithms 4 and 5 satisfy for any integer $R \geq 1$*

$$\frac{1}{R} \sum_{t=0}^{R-1} \left( \mathbb{E}\left[ f\left( x^t \right) \right] - f^{\mathrm{inf}} \right)$$

$$\leq \frac{4B^2}{\eta_g n K R} + 8\eta_g \sigma^2 + \frac{20\sigma^2 L}{nR} \sum_{t=0}^{R-1} \sum_{i=1}^{n} \sum_{\ell=0}^{K-1} \left( \eta_{i,\ell}^t \right)^2$$

$$- \frac{8}{nKR} \sum_{t=0}^{R-1} \sum_{i=1}^{n} \left( \frac{1}{2} - \eta_g n L K - \frac{5}{2} \left( \sum_{\ell=0}^{K-1} \left( \eta_{i,\ell}^t \right)^2 \right) L^2 K \right) \sum_{j=0}^{K-1} \mathbb{E}\left[ \left\langle \nabla f\left( z_{i,j}^t \right), z_{i,j}^t - x^* \right\rangle \right].$$

*Proof.* First, for any integer $t \geq 0$ by Lemma C.6 we have

$$\frac{\eta_g}{4} \left( \mathbb{E}\left[ f\left( x^t \right) \right] - f^{\mathrm{inf}} \right)$$

$$\leq \frac{1}{nK} \left( \mathbb{E}\left[ \left\| x^t - x^* \right\|^2 \right] - \mathbb{E}\left[ \left\| x^{t+1} - x^* \right\|^2 \right] \right) + \frac{5\eta_g L}{2nK} \sum_{i=1}^{n} \sum_{j=0}^{K-1} \mathbb{E}\left[ \left\| z_{i,j}^t - x^* \right\|^2 \right]$$

$$- 2\eta_g \left( \frac{1}{2} - \eta_g n L K \right) \frac{1}{nK} \sum_{i=1}^{n} \sum_{j=0}^{K-1} \mathbb{E}\left[ \left\langle \nabla f\left( z_{i,j}^t \right), z_{i,j}^t - x^* \right\rangle \right] + 2\eta_g^2 \sigma^2.$$

which after rearranging the terms and multiplying both sides by $\frac{4}{\eta_g}$ gives

$$\mathbb{E}\left[ f\left( x^t \right) \right] - f^{\mathrm{inf}}$$

$$\leq \frac{4}{\eta_g nK}\left(\mathbb{E}\left[\left\|x^t - x^*\right\|^2\right] - \mathbb{E}\left[\left\|x^{t+1} - x^*\right\|^2\right]\right) + \frac{10L}{nK}\sum_{i=1}^{n}\sum_{j=0}^{K-1}\mathbb{E}\left[\left\|z_{i,j}^t - x^*\right\|^2\right]$$

$$- 8\left(\frac{1}{2} - \eta_g nLK\right)\frac{1}{nK}\sum_{i=1}^{n}\sum_{j=0}^{K-1}\mathbb{E}\left[\left\langle \nabla f\left(z_{i,j}^t\right), z_{i,j}^t - x^*\right\rangle\right] + 8\eta_g\sigma^2.$$

and using Lemma C.7 we obtain the inequality

$$\mathbb{E}\left[f\left(x^t\right)\right] - f^{\inf}$$

$$\leq \frac{4}{\eta_g nK}\left(\mathbb{E}\left[\left\|x^t - x^*\right\|^2\right] - \mathbb{E}\left[\left\|x^{t+1} - x^*\right\|^2\right]\right) + 8\eta_g\sigma^2$$

$$- 8\left(\frac{1}{2} - \eta_g nLK\right)\frac{1}{nK}\sum_{i=1}^{n}\sum_{j=0}^{K-1}\mathbb{E}\left[\left\langle \nabla f\left(z_{i,j}^t\right), z_{i,j}^t - x^*\right\rangle\right]$$

$$+ \frac{20L}{nK}\sum_{i=1}^{n}\sum_{j=0}^{K-1}\left(L\left(\sum_{\ell=0}^{j-1}\left(\eta_{i,\ell}^t\right)^2\right)\sum_{\ell=0}^{j-1}\mathbb{E}\left[\left\langle \nabla f\left(z_{i,\ell}^t\right), z_{i,\ell}^t - x^*\right\rangle\right] + \sigma^2\sum_{\ell=0}^{j-1}\left(\eta_{i,\ell}^t\right)^2\right).$$

$$(66)$$

Now, we further upper bound the above expression, notably we have

$$\frac{20L}{nK}\sum_{i=1}^{n}\sum_{j=0}^{K-1}\left(L\left(\sum_{\ell=0}^{j-1}\left(\eta_{i,\ell}^t\right)^2\right)\sum_{\ell=0}^{j-1}\mathbb{E}\left[\left\langle \nabla f\left(z_{i,\ell}^t\right), z_{i,\ell}^t - x^*\right\rangle\right] + \sigma^2\sum_{\ell=0}^{j-1}\left(\eta_{i,\ell}^t\right)^2\right)$$

$$\overset{(a)}{\leq} \frac{20L^2}{nK}\sum_{i=1}^{n}\left(\sum_{\ell=0}^{K-1}\left(\eta_{i,\ell}^t\right)^2\right)\sum_{j=0}^{K-1}\sum_{\ell=0}^{j-1}\mathbb{E}\left[\left\langle \nabla f\left(z_{i,\ell}^t\right), z_{i,\ell}^t - x^*\right\rangle\right] + \frac{20\sigma^2 L}{n}\sum_{i=1}^{n}\sum_{\ell=0}^{K-1}\left(\eta_{i,\ell}^t\right)^2$$

where in (a) we use the fact that $j \leq K$ to upper bound the two sums over the local step sizes. Using again $j \leq K$ to upper bound the sum over $\ell$ of the non-negative[14] inner products we obtain

$$\frac{20L^2}{nK}\sum_{i=1}^{n}\left(\sum_{\ell=0}^{K-1}\left(\eta_{i,\ell}^t\right)^2\right)\sum_{j=0}^{K-1}\sum_{\ell=0}^{j-1}\mathbb{E}\left[\left\langle \nabla f\left(z_{i,\ell}^t\right), z_{i,\ell}^t - x^*\right\rangle\right] + \frac{20\sigma^2 L}{n}\sum_{i=1}^{n}\sum_{\ell=0}^{K-1}\left(\eta_{i,\ell}^t\right)^2$$

$$\leq \frac{20L^2 K}{nK}\sum_{i=1}^{n}\left(\sum_{\ell=0}^{K-1}\left(\eta_{i,\ell}^t\right)^2\right)\sum_{\ell=0}^{K-1}\mathbb{E}\left[\left\langle \nabla f\left(z_{i,\ell}^t\right), z_{i,\ell}^t - x^*\right\rangle\right] + \frac{20\sigma^2 L}{n}\sum_{i=1}^{n}\sum_{\ell=0}^{K-1}\left(\eta_{i,\ell}^t\right)^2,$$

hence injecting this bound in (66) yields

$$\mathbb{E}\left[f\left(x^t\right)\right] - f^{\inf}$$

$$\overset{(66)}{\leq} \frac{4}{\eta_g nK}\left(\mathbb{E}\left[\left\|x^t - x^*\right\|^2\right] - \mathbb{E}\left[\left\|x^{t+1} - x^*\right\|^2\right]\right) + 8\eta_g\sigma^2$$

$$- 8\left(\frac{1}{2} - \eta_g nLK\right)\frac{1}{nK}\sum_{i=1}^{n}\sum_{j=0}^{K-1}\mathbb{E}\left[\left\langle \nabla f\left(z_{i,j}^t\right), z_{i,j}^t - x^*\right\rangle\right]$$

$$+ \frac{20L^2 K}{nK}\sum_{i=1}^{n}\left(\sum_{\ell=0}^{K-1}\left(\eta_{i,\ell}^t\right)^2\right)\sum_{\ell=0}^{K-1}\mathbb{E}\left[\left\langle \nabla f\left(z_{i,\ell}^t\right), z_{i,\ell}^t - x^*\right\rangle\right]$$

$$+ \frac{20\sigma^2 L}{n}\sum_{i=1}^{n}\sum_{\ell=0}^{K-1}\left(\eta_{i,\ell}^t\right)^2$$

$$\overset{(a)}{=} \frac{4}{\eta_g nK}\left(\mathbb{E}\left[\left\|x^t - x^*\right\|^2\right] - \mathbb{E}\left[\left\|x^{t+1} - x^*\right\|^2\right]\right) + 8\eta_g\sigma^2 + \frac{20\sigma^2 L}{n}\sum_{i=1}^{n}\sum_{\ell=0}^{K-1}\left(\eta_{i,\ell}^t\right)^2$$

$$- \frac{8}{nK}\sum_{i=1}^{n}\left(\frac{1}{2} - \eta_g nLK - \frac{5}{2}\left(\sum_{\ell=0}^{K-1}\left(\eta_{i,\ell}^t\right)^2\right)L^2 K\right)\sum_{j=0}^{K-1}\mathbb{E}\left[\left\langle \nabla f\left(z_{i,j}^t\right), z_{i,j}^t - x^*\right\rangle\right].$$

---

[14]The non-negativity of the inner product $\langle \nabla f(x), x - x^*\rangle$ for any $x^* \in \arg\min_{x \in \mathbb{R}^d} f(x)$ follows from Lemma E.11.

where in (a) we merged the two sums with the inner products. It remains to sum this inequality over $t \in \{0, \ldots, R-1\}$ for a fixed integer $R \geq 1$, this gives

$$\frac{1}{R} \sum_{t=0}^{R-1} \left( \mathbb{E}\left[ f\left(x^t\right)\right] - f^{\mathrm{inf}} \right)$$

$$\leq \frac{4}{\eta_g nKR} \left( \left\| x^0 - x^* \right\|^2 - \mathbb{E}\left[ \left\| x^R - x^* \right\|^2 \right]\right) + 8\eta_g \sigma^2 L + \frac{20\sigma^2 L}{nR} \sum_{t=0}^{R-1} \sum_{i=1}^{n} \sum_{\ell=0}^{K-1} \left(\eta_{i,\ell}^t\right)^2$$

$$- \frac{8}{nKR} \sum_{t=0}^{R-1} \sum_{i=1}^{n} \left( \frac{1}{2} - \eta_g nLK - \frac{5}{2} \left( \sum_{\ell=0}^{K-1} \left(\eta_{i,\ell}^t\right)^2 \right) L^2 K \right) \sum_{j=0}^{K-1} \mathbb{E}\left[ \left\langle \nabla f\left(z_{i,j}^t\right), z_{i,j}^t - x^* \right\rangle \right]$$

$$\overset{\text{Ass. 1.4}}{\leq} \frac{4B^2}{\eta_g nKR} + 8\eta_g \sigma^2 L + \frac{20\sigma^2 L}{nR} \sum_{t=0}^{R-1} \sum_{i=1}^{n} \sum_{\ell=0}^{K-1} \left(\eta_{i,\ell}^t\right)^2$$

$$- \frac{8}{nKR} \sum_{t=0}^{R-1} \sum_{i=1}^{n} \left( \frac{1}{2} - \eta_g nLK - \frac{5}{2} \left( \sum_{\ell=0}^{K-1} \left(\eta_{i,\ell}^t\right)^2 \right) L^2 K \right) \sum_{j=0}^{K-1} \mathbb{E}\left[ \left\langle \nabla f\left(z_{i,j}^t\right), z_{i,j}^t - x^* \right\rangle \right],$$

and this establishes the desired inequality. $\qquad\square$

# E  USEFUL RESULTS

For any vectors $x, y \in \mathbb{R}^d$, we have

$$2 \langle x, y \rangle = \|x\|^2 + \|y\|^2 - \|x - y\|^2. \tag{67}$$

**Lemma E.1** (Variance Decomposition). *For any random vector $X \in \mathbb{R}^d$ and any non-random vector $c \in \mathbb{R}^d$ we have*

$$\mathbb{E}\left[ \|X - c\|^2 \right] = \mathbb{E}\left[ \|X - \mathbb{E}\left[X\right]\|^2 \right] + \|\mathbb{E}\left[X\right] - c\|^2.$$

**Lemma E.2** (Tower Property of the Expectation). *For any random variables $X \in \mathbb{R}^d$ and $Y_1, \ldots, Y_n$ we have*

$$\mathbb{E}\left[ \mathbb{E}\left[ X \mid Y_1, \ldots, Y_n \right]\right] = \mathbb{E}\left[X\right].$$

**Lemma E.3** (Cauchy Schwarz's Inequality). *For any vectors $a, b \in \mathbb{R}^d$ we have*

$$\langle a, b \rangle \leq |\langle a, b \rangle| \leq \|a\| \|b\|.$$

**Lemma E.4** (Young's Inequality (Norm Form)). *For any vectors $a, b \in \mathbb{R}^d$ and any scalar $\alpha > 0$ we have*

$$\|a + b\|^2 \leq (1 + \alpha) \|x\|^2 + \left( 1 + \frac{1}{\alpha} \right) \|y\|^2.$$

**Lemma E.5** (Young's Inequality (Inner Product Form)). *For any vectors $a, b \in \mathbb{R}^d$ and any scalar $\alpha > 0$ we have*

$$2 \langle a, b \rangle \leq 2 |\langle a, b \rangle| \leq \alpha \|x\|^2 + \frac{1}{\alpha} \|y\|^2. \tag{68}$$

*Proof.* It's enough to prove inequality (68) when $d = 1$. Hence, consider $a, b \in \mathbb{R}$, we have given $\alpha > 0$

$$2ab \leq 2|ab| = 2|a| \cdot |b| = 2 \left| \sqrt{\alpha}\, a \right| \cdot \left| \frac{b}{\sqrt{\alpha}} \right| \overset{\text{(a)}}{\leq} \alpha |a|^2 + \frac{1}{\alpha} |b|^2 = \alpha\, a^2 + \frac{b^2}{\alpha},$$

where in (a) we use the arithmetic-geometric inequality in $n = 2$ variables $\left( \sqrt{\alpha}\,|a|, \frac{1}{\sqrt{\alpha}}\,|b| \right)$. $\qquad\square$

**Lemma E.6** (Jensen's Inequality). *Let $f \colon \mathbb{R}^d \to \mathbb{R}$ be a convex function then*

1. *(Probabilistic Form) for any random vector $X \in \mathbb{R}^d$ we have*

$$\mathbb{E}\left[ f(X) \right] \geq f\left( \mathbb{E}\left[X\right] \right).$$

2. *(Deterministic Form) for any vectors $v_1, \ldots, v_n \in \mathbb{R}^d$ and scalars $\lambda_1, \ldots, \lambda_n \in \mathbb{R}_+$ we have*

$$\sum_{i=1}^{n} \lambda_i f(v_i) \geq f\left(\sum_{i=1}^{n} \lambda_i v_i\right),$$

*provided $\lambda_i \geq 0$ for all $i \in [n]$ and $\sum_{i=1}^{n} \lambda_i = 1$.*

**Lemma E.7.** *For any vectors $v_1, \ldots, v_n \in \mathbb{R}^d$ we have*

$$\left\|\sum_{i=1}^{n} v_i\right\|^2 \leq n \sum_{i=1}^{n} \|v_i\|^2.$$

*Proof.* The function $\|\cdot\|^2 : \mathbb{R}^d \to \mathbb{R}$ is $\mu$–strongly convex with $\mu = 2$ so is convex thus applying Jensen's inequality E.6 with $\lambda_1 = \cdots = \lambda_n = \frac{1}{n}$ gives

$$\left\|\sum_{i=1}^{n} \frac{v_i}{n}\right\|^2 \leq \frac{1}{n} \sum_{i=1}^{n} \|v_i\|^2,$$

and multiplying both sides by $n^2$ gives the desired inequality. $\qquad\square$

**Lemma E.8.** *For any vectors $v_1, \ldots, v_n \in \mathbb{R}^d$ and any scalars $\lambda_1, \ldots, \lambda_n \in \mathbb{R}$ we have*

$$\left\|\sum_{i=1}^{n} \lambda_i v_i\right\|^2 \leq \left(\sum_{i=1}^{n} \lambda_i^2\right) \sum_{i=1}^{n} \|v_i\|^2.$$

*Proof.* Using the triangle inequality followed by the Cauchy-Schwarz inequality (Lemma E.3) we have

$$\left\|\sum_{i=1}^{n} \lambda_i v_i\right\|^2 \leq \left(\sum_{i=1}^{n} |\lambda_i| \, \|v_i\|\right)^2$$

$$\overset{\text{Lem. E.3}}{\leq} \left(\sqrt{\sum_{i=1}^{n} \lambda_i^2} \cdot \sqrt{\sum_{i=1}^{n} \|v_i\|^2}\right)^2$$

$$= \left(\sum_{i=1}^{n} \lambda_i^2\right) \sum_{i=1}^{n} \|v_i\|^2,$$

as claimed. $\qquad\square$

**Lemma E.9** (Nonnegativity of the Bregman Divergence). *Let $f : \mathbb{R}^d \to \mathbb{R}$ be convex and continuously differentiable over $\mathbb{R}^d$ then, for any $x, y \in \mathbb{R}^d$*

$$D_f(y, x) := f(y) - f(x) - \langle \nabla f(x), y - x \rangle \geq 0 \tag{69}$$

*In particular, under Assumption 1.4, applying (69) for $y = x^* \in \arg\min_{x \in \mathbb{R}^d} f(x)$ we obtain*

$$D_f(x^*, x) = f^{\text{inf}} - f(x) - \langle \nabla f(x), x^* - x \rangle \geq 0,$$

*or, said differently*

$$-\left(f(x) - f^{\text{inf}}\right) \geq -\langle \nabla f(x), x - x^* \rangle. \tag{70}$$

**Lemma E.10.** *Let $f : \mathbb{R}^d \to \mathbb{R}$ be a function satisfying Assumption 1.2 then, for all $x \in \mathbb{R}^d$*

$$\|\nabla f(x)\|^2 \leq 2L(f(x) - f^{\text{inf}}). \tag{71}$$

*In particular, if $f$ admits at least one global minimizer $x^* \in \arg\min_{x \in \mathbb{R}^d} f(x)$ then the inequality (71) holds for $f^{\text{inf}} := \min_{x \in \mathbb{R}^d} f(x)$.*

*Remark* E.1. Let $f \colon \mathbb{R}^d \to \mathbb{R}$ be a continuously differentiable function defined everywhere on $\mathbb{R}^d$, then any global minimizer $x^* \in \mathbb{R}^d$ of $f$ on its domain (which is $\mathbb{R}^d$) satisfies the first-order optimality condition, that is $\nabla f(x^*) = 0$. Moreover, under Assumption 1.4 for any $x \in \mathbb{R}^d$ (Nesterov, 2018, Theorem 2.1.1 on p. 81) the equivalence

$$x \in \arg\min_{x \in \mathbb{R}^d} f(x) \text{ if, and only if } \nabla f(x) = 0, \tag{72}$$

holds.

**Lemma E.11.** *Let* $f \colon \mathbb{R}^d \to \mathbb{R}$ *be a function satisfying Assumptions 1.2 and 1.4 then, for all* $x \in \mathbb{R}^d$ *and all* $y^\star \in \arg\min_{x \in \mathbb{R}^d} f(x) \neq \varnothing$

$$\|\nabla f(x)\|^2 \leq L \langle \nabla f(x), x - y^\star \rangle. \tag{73}$$

*Proof.* Let $x \in \mathbb{R}^d$, since $y^\star \in \arg\min_{x \in \mathbb{R}^d} f(x)$ then, using (72) and (Nesterov, 2018, Theorem 2.1.5 on p. 87) we have

$$\|\nabla f(x) - \nabla f(y^\star)\|^2 \overset{(72)}{=} \|\nabla f(x)\|^2$$
$$\leq L \langle \nabla f(x) - \nabla f(y^\star), x - y^\star \rangle$$
$$\overset{(72)}{=} L \langle \nabla f(x), x - y^\star \rangle,$$

and the lemma follows. $\qquad\square$

**Lemma E.12.** *Let* $f \colon \mathbb{R}^d \to \mathbb{R}$ *be a function satisfying Assumptions 1.2 and 1.4 then, for all* $x, y \in \mathbb{R}^d$, *with* $f^{\text{inf}} := \min_{x \in \mathbb{R}^d} f(x)$ *we have*

$$f(x) - f(y) \leq \frac{1}{2} \left( f(x) - f^{\text{inf}} \right) + L \|x - y\|^2. \tag{74}$$

*Proof.* Let $x, y \in \mathbb{R}^d$ then, using Lemma E.9 since $f$ is continuously differentiable and convex on $\mathbb{R}^d$ we have

$$D_f(y, x) := f(y) - f(x) - \langle \nabla f(x), y - x \rangle \geq 0,$$

hence, reshuffling the above inequality yields

$$\langle \nabla f(x), x - y \rangle \geq f(x) - f(y). \tag{75}$$

Next, from inequality (75) we apply Young's inequality (Lemma E.5) with parameter $\alpha = 1/2L$ and then we use Lemma E.10 using the fact that $f$ is lower bounded by $f^{\text{inf}}$. Hence, we obtain

$$f(x) - f(y) \overset{(75)}{\leq} \langle \nabla f(x), x - y \rangle$$
$$\overset{\text{Lem. E.5}}{\leq} \frac{\alpha}{2} \|\nabla f(x)\|^2 + \frac{1}{2\alpha} \|x - y\|^2$$
$$\overset{(a)}{=} \frac{1}{4L} \|\nabla f(x)\|^2 + L \|x - y\|^2$$
$$\overset{\text{Lem. E.10}}{\leq} \frac{1}{2} \left( f(x) - f^{\text{inf}} \right) + L \|x - y\|^2,$$

where in (a) we use the $\alpha = \frac{1}{2L}$. This achieves the proof of the lemma. $\qquad\square$

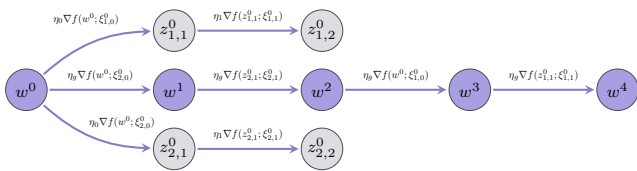

Figure 3: An example of the first round in Decaying Local SGD with $n = 2$ and $K = 2$. In this tree, $x^0 = w^0$ and $x^1 = w^4$, where $x^0, x^1$ are defined in Alg. 5. Every edge has a weight, which represents a stochastic gradient and the step size used to obtain a new point.

## F  EXTENSION TO OTHER ASYNCHRONOUS AND LOCAL METHODS

Minibatch SGD and Local SGD are not the only methods for accelerating distributed optimization. Many other techniques exist, including Asynchronous SGD (Recht et al., 2011; Maranjyan et al., 2025), as well as various combinations of these approaches. Building on our progress, we now aim to extend our new insights from Section 4 to other distributed methods. It turns out that all these methods can be analyzed using a unified analysis and technique, and our new step size rules can be incorporated not only into Local SGD but also into other methods.

Here we will be brief, and we delegate details to Section G. The idea is to represent any method with a computation tree (Tyurin and Sivtsov, 2025). Initially, the tree is $G = (V, E)$ with $V = \{x^0\}$ and $E = \emptyset$. Then, every method can be represented by the following procedure: take two points $w_{\text{base}}$ and $w_{\text{grad}}$ from $V$ (in the first step, the only choice is $x^0$), choose a step size $\eta$, find a new point $w_{\text{new}} = w_{\text{base}} - \eta \nabla f(w_{\text{grad}}; \xi)$, add it to $V$, and add the weighted directed edge $(w_{\text{base}}, w_{\text{new}}, \eta \nabla f(w_{\text{grad}}; \xi))$ to $E$, then start the procedure again. For instance, Decaying Local SGD (Algorithm 5) can represented by Figure 3. The work by Tyurin and Sivtsov (2025) provides a general framework for analyzing virtually any local and asynchronous methods via computation graphs. However, we noticed that their theory can be improved in the aspect discussed in Section 3.1: the local step sizes used in their framework can be significantly increased, making the methods considered by the framework more practical[15]. In their version, $\eta_\ell = \eta_g$, but $\eta_\ell$ can be increased, as we explain in Section 4. We can also take $\eta_j = \sqrt{b/(j+1)(\log K + 1)} \times \eta_g$, with the only difference that $j$ is the tree distance between the main branch ($w^0 \to \cdots \to w^4$ in Fig.3) and the point where the stochastic gradient was calculated (e.g., $j = 0$ for $w^0$; $j = 1$ for $z^0_{1,1}$ and $z^0_{2,1}$; and $j = 2$ for $z^0_{1,2}$ and $z^0_{2,2}$). Moreover, this choice of step size is adaptive to the length of the local branch, which is important for asynchronous methods when we do not know *a priori* the length of the local branch. See details in Section G.

---

[15]This observation was, in fact, the starting point of this project.

---

**Algorithm 6:** Birch SGD framework

**Input:** starting point $w^0 \in \mathbb{R}^d$, step size $\gamma \geq 0$
Initialize the set of computed points: $V = \{w^0\}$
(and the set of directed edges $E = \emptyset$)
**for** $k = 0, 1, 2, \dots$ **do**
 Choose any point $w_{\text{base}} \in V$ from which to compute a new point
 Choose any point $w_{\text{grad}} \in V$ at which to compute a stochastic gradient
 Choose any step size $\gamma > 0$
 Compute the new point: $w^{k+1} = w_{\text{base}} - \gamma \nabla f(w_{\text{grad}}; \zeta), \zeta \sim \mathcal{D}$,
 where $\zeta$ might be reused (not necessarily a i.i.d. sequence is generated)
 Add $w^{k+1}$ to the set of computed points $V$
 (and add the edge with weight $(w_{\text{base}}, w^{k+1}, \gamma \nabla f(w_{\text{grad}}; \zeta))$ to the set of edges $E$)
**end for**

---

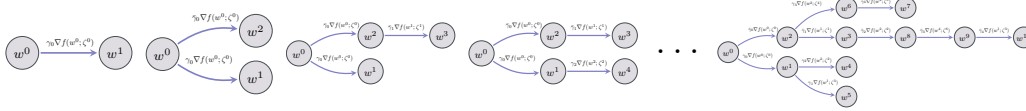

Figure 4: A possible computation tree $G$ for an SGD method after four steps and beyond.

## G  An Improved Birch SGD Theory

### G.1  Preliminaries

Let us briefly recall the Birch SGD framework introduced by Tyurin and Sivtsov (2025). The core idea is that a broad class of SGD methods, including Vanilla SGD, Asynchronous SGD, Local SGD, can all be described using a unified graph-based view.

More precisely, any SGD method can be constructed as in Algorithm 6. The procedure begins at an initial point $w^0 \in \mathbb{R}^d$ and computes a sequence of iterates by selecting, at each step, a *base point* $w_{\text{base}}$ and a (possibly different) point $w_{\text{grad}}$ at which to evaluate the stochastic gradient. The next iterate is then computed as

$$w^{k+1} = w_{\text{base}} - \gamma \nabla f(w_{\text{grad}}; \zeta), \quad \zeta \sim \mathcal{D}.$$

The step sizes $\gamma$ are also selected in every iteration. This new point $w^{k+1}$ is added to the set of computed points $V$, and the directed edge

$$\left(w_{\text{base}}, w^{k+1}, \gamma \nabla f(w_{\text{grad}}; \zeta)\right)$$

is added to the set of edges $E$. After $k$ steps, the entire process can be represented as a weighted directed tree $G = (V, E)$, called a *computation tree*.

Initially, the method starts from $w^0$ and computes a stochastic gradient there, generating $w^1 = w^0 - \gamma_0 \nabla f(w^0; \cdot)$. In subsequent steps, there are several choices for how to form $w^2$:

$$w^2 = w^i - \gamma_1 \nabla f(w^j; \cdot),$$

for any $i, j \in \{0, 1\}$. In general, the number of possible ways to construct future iterates grows exponentially, leading to different computation trees (see Figure 4).

We have to define a *main branch* and its associated *auxiliary sequence*.

**Definition G.1** (*Main Branch* and *Auxiliary Sequence*). For a given computation tree $G$, we call a sequence $\{x^k\}_{k \geq 0}$ a *main branch* if it forms a path in $G$ starting at the initial node $w^0 \equiv x^0$. That is, for each $k \geq 0$, the node $x^{k+1}$ is a direct successor of $x^k$ in $G$. By the construction of tree $G$, if $\{x^k\}_{k \geq 0}$ is a *main branch*, then for each $k \geq 0$ there exists a unique triple $(\gamma_k, z^k, \xi^k)$, where $\gamma_k > 0$, $z^k \in V$ and $\xi^k \sim \mathcal{D}$, such that $x^{k+1} = x^k - \gamma_k \nabla f(z^k; \xi^k)$. The sequence $\{(\gamma_k, z^k, \xi^k)\}_{k \geq 0}$, which generates the main branch $\{x^k\}_{k \geq 0}$, is called an *auxiliary sequence*.

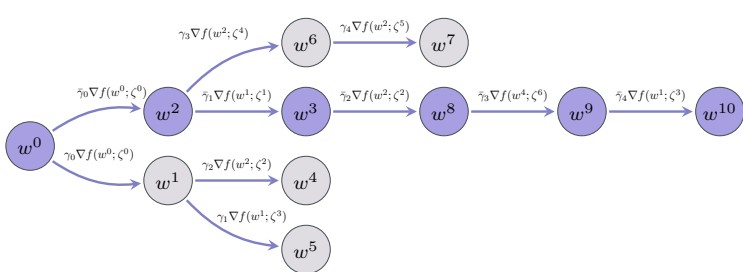

Figure 5: A natural choice of a main branch from the example in Figure 4.

Although multiple main branches may exist in principle, for all practical SGD methods, there is usually a unique and natural choice. In Figure 5, we can take a main branch $\{x^k\}_{k\geq 0}$ as follows: $x^0 = w^0, x^1 = w^2, x^2 = w^3, x^3 = w^8, x^4 = w^9, x^5 = w^{10}$. The auxiliary sequence is accordingly defined by $(\gamma_0, z^0, \xi^0) = (\bar{\gamma}_0, w^0, \zeta^0)$, $(\gamma_1, z^1, \xi^1) = (\bar{\gamma}_1, w^1, \zeta^1)$, $(\gamma_2, z^2, \xi^2) = (\bar{\gamma}_2, w^2, \zeta^2)$, $(\gamma_3, z^3, \xi^3) = (\bar{\gamma}_3, w^4, \zeta^6)$, $(\gamma_4, z^4, \xi^4) = (\bar{\gamma}_4, w^1, \zeta^3)$.

Next, we have to define the *tree distance* between two points in $V$:

**Definition G.2.** For all $y, z \in V$, the tree distance $\mathrm{dist}(y, z)$ is the maximal number of edges separating $y$ and $z$ from their closest common ancestor in $G$.

For example, in Figure 5, the distance $\mathrm{dist}(w^9, w^4) = \max\{4, 2\} = 4$ because the closest common ancestor is $w^0$, and the respective depths of $w^9$ and $w^4$ from $w^0$ are 4 and 2. We generalize this definition to the distance between a node and a main branch:

**Definition G.3.** For all $y \in V$ and a main branch $\{x^k\}_{k\geq 0}$, we define the distance from the node $y$ to the main branch as

$$\mathrm{dist}(y, \{x^k\}) = \min_{k\geq 0} \mathrm{dist}(y, x^k).$$

For instance, $\mathrm{dist}(w^7, \{x^k\}) = 2$ in Figure 5, where $\{x^1 \equiv w^0, x^2 \equiv w^2, x^3 \equiv w^3, x^4 \equiv w^8, x^5 \equiv w^9, x^6 \equiv w^{10}\}$ is the main branch.

We also define the *representation* of a point to capture which stochastic gradients have been used to generate it.

**Definition G.4.** For all $y \in V$, the representation $\mathrm{repr}(y)$ is the multiset of stochastic gradients applied to $w^0$ to get $y$. In other words, there exist $\{(\gamma_1, m^1, \kappa^1), \ldots, (\gamma_p, m^p, \kappa^p)\}$ for some $p \geq 0$ such that $y = w^0 - \sum_{j=1}^{p} \gamma_j \nabla f(m^j, \kappa^j)$. Then, we define $\mathrm{repr}(y) := \{(m^1, \kappa^1), \ldots, (m^p, \kappa^p)\}$ (ignoring the step sizes).

We define the representation of points to understand how all points are related. An important relation that we need is that $\mathrm{repr}(x) \subseteq \mathrm{repr}(y)$, which essentially means that all stochastic gradients used to compute $x$ are also used to compute $y$. For instance, in Figure 5, we have:

$$\mathrm{repr}(w^9) = \{(w^0, \zeta^0), (w^1, \zeta^1), (w^2, \zeta^2), (w^4, \zeta^6)\}$$

and

$$\mathrm{repr}(w^4) = \{(w^0, \zeta^0), (w^2, \zeta^2)\}.$$

Thus, $\mathrm{repr}(w^4) \subseteq \mathrm{repr}(w^9)$.

### G.2 MAIN RESULT

The only difference between our framework and the framework by Tyurin and Sivtsov (2025) is that we allow different step sizes in Algorithm 6. We are ready to state our main result:

**Theorem G.1** (Main Theorem). *Let Assumptions 1.2 and 1.3 hold. Consider any* SGD *method represented by* computation tree $G = (V, E)$. *Let* $\{x^k\}_{k\geq 0}$ *be a* main branch *of $G$ and* $\{(\gamma_k, z^k, \xi^k)\}_{k\geq 0}$ *be the corresponding* auxiliary sequence *(see Def. G.1) that satisfy the following conditions:*
**Condition 1:** *For all $k \geq 0$, $\xi^k$ is statistically independent of* $\{(x^{i+1}, z^{i+1}, \xi^i)\}_{i=0}^{k-1}$.

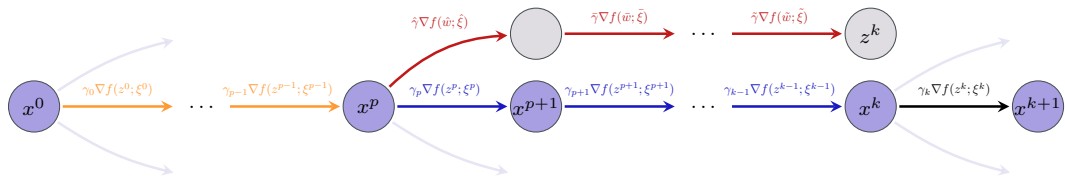

Figure 6: A general representation of the step $x^{k+1} = x^k - \gamma \nabla f(z^k; \xi^k)$ that shows how $x^k$ and $z^k$ are graph-geometrically related.

**Condition 2:** *The representation of $z^k$ is contained within that of $x^k$, i.e., $\mathrm{repr}(z^k) \subseteq \mathrm{repr}(x^k)$ for all $k \geq 0$. Equivalently, all stochastic gradients used in the computation of $z^k$ are also utilized in calculating $x^k$.*

**Condition 3:** *There exists a constant $R \in [0, \infty]$ such that $\mathrm{dist}(x^k, z^k) \leq R$ for all $k \geq 0$.*

**Condition 4:** *The step sizes along the main branch satisfy $\gamma_k = \gamma_g := \min\{\frac{1}{2L}, \frac{1}{4RL}, \frac{\varepsilon}{8\sigma^2 L}\}$ for all $k \geq 0$. Any other step size $\gamma$ can be taken as large as[16] $\sqrt{\frac{R}{(j+1)(\log R+1)}}\gamma_g$, where $j = \mathrm{dist}(y, \{x^k\})$ (from Def. G.3) and $y$ is the node at which the step $y - \gamma \nabla f(\cdot; \cdot)$ with this step size $\gamma$ is applied to find a new node.*

*Then $\frac{1}{K} \sum_{k=0}^{K-1} \mathbb{E}\left[\|\nabla f(x^k)\|^2\right] \leq \varepsilon$ for all*

$$K \geq \frac{8(R+1)L\Delta}{\varepsilon} + \frac{16\sigma^2 L\Delta}{\varepsilon^2}.$$

Assumptions 1.2 and 1.3 are standard in the analysis of stochastic methods. Conditions 1, 2, and 3 are the same as in the original paper by (Tyurin and Sivtsov, 2025); we refer to Section 2.1 for a detailed explanation and intuition.

### G.3 ON THE NEW CONDITION 4

Let us clarify the new Condition 4. As we explained previously, Tyurin and Sivtsov (2025) assume that all step sizes are the same in computation graphs. Here, we relax this assumption in the following way. Along the chosen main branch of the computation graph, we assume that all step sizes are equal to $\gamma_g$. Thus, in Figure 6, $\gamma_0 = \gamma_1 = \cdots = \gamma_k = \gamma_g$. However, all other step sizes are allowed to be larger (up to a logarithmic factor). Consider Figure 6 and an arbitrary path. In Figure 6, we take the path from $x^p$ to $z^k$ with the step sizes $\hat{\gamma}, \bar{\gamma}, \ldots, \tilde{\gamma}$. According to the rule from Condition 4, we are allowed to take any $\hat{\gamma} \leq \sqrt{\frac{R}{(0+1)(\log R+1)}}\gamma_g$ because $j = \mathrm{dist}(x^p, \{x^k\}) = 0$ (from Def. G.3), and $x^p$ is the node at which the step $x^p - \hat{\gamma}\nabla f(\hat{x}; \hat{\xi})$ is applied with the step size $\hat{\gamma}$. Similarly, $\bar{\gamma} \leq \sqrt{\frac{R}{(1+1)(\log R+1)}}\gamma_g$ because $j = \mathrm{dist}(y, \{x^k\}) = 1$, where $y$ is the next point generated by the step $x^p - \bar{\gamma}\nabla f(\bar{x}; \bar{\xi})$, and so on.

### G.4 ON THE LARGER STEP SIZES

The larger the distance between a point and the main branch, the smaller we should take the step. However, up to the logarithmic factor, it will never be smaller than in (Tyurin and Sivtsov, 2025):

$$\sqrt{\frac{R}{(j+1)(\log R+1)}}\gamma_g \geq \sqrt{\frac{1}{\log R+1}}\gamma_g$$

because $j \leq R - 1$ due to Condition 3. Moreover, it can be arbitrarily larger: if we take $j = 0$, then $\sqrt{\frac{R}{(j+1)(\log R+1)}}\gamma_g = \sqrt{\frac{R}{\log R+1}}\gamma_g$. In virtually all optimal algorithms, $R = \sigma^2/\varepsilon$ (e.g., Section G.5.2 or (Tyurin and Sivtsov, 2025)); thus, $\sqrt{\frac{R}{(j+1)(\log R+1)}}\gamma_g = \tilde{\Theta}\left(\sqrt{\frac{\sigma^2}{\varepsilon}}\gamma_g\right)$ and the increase can be $\tilde{\Theta}\left(\sqrt{\sigma^2/\varepsilon}\right)$ times.

---

[16]For $R = 0$, we use the standard convention $\frac{0}{\log 0+1} = 0$.

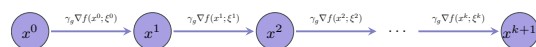

Figure 7: The computation tree of Vanilla SGD

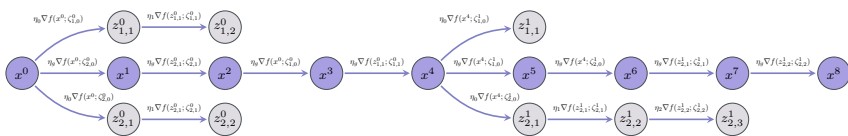

Figure 8: An example of a Decaying Local SGD (asynchronous version) computation tree with $b = 4$ and 2 workers, each performing local steps over 2 global steps. While in the first round, they perform the same number of local steps, they have different numbers in the second round. Each stochastic gradient is used 2 times in the tree of this method.

### G.5 EXAMPLES OF ALGORITHMS

Since we do not change Conditions 1, 2, and 3, all the results, theorems, and proofs from Tyurin and Sivtsov (2025) hold (up to universal constants), with the only difference being that we have to use the new step size rule from Condition 4, which does not interfere with the previous derivations. Let us consider some examples.

#### G.5.1 Vanilla SGD

The classical stochastic gradient descent (Vanilla SGD) method is $w^{k+1} = w^k - \gamma_g \nabla f(w^k; \zeta^k)$, where $w^0$ is a starting point and $\{\zeta^k\}$ are i.i.d. random variables. Taking $x^k = z^k = w^k$ and $\xi^k = \zeta^k$ for all $k \geq 0$, we get a main branch. All conditions of Theorem G.1 hold: $\xi^k$ is independent of $\{(x^{i+1}, z^{i+1}, \xi^i)\}_{i=0}^{k-1}$, $\mathrm{repr}(x^k) = \mathrm{repr}(z^k)$ for all $k \geq 0$, and $R = 0$. We get the classical *iteration complexity* $\mathcal{O}\left(L\Delta/\varepsilon + \sigma^2 L\Delta/\varepsilon^2\right)$ (Lan, 2020; Arjevani et al., 2023). The corresponding tree is in Figure 7.

#### G.5.2 Decaying Local SGD (ASYNCHRONOUS VERSION)

Let us consider a generalization of Algorithm 5 from Section 4. Consider Algorithm 7. The only difference between Algorithm 5 and Algorithm 7 is that, in the latter, we allow the workers to run different numbers of local steps (e.g., due to random delays in computations or heterogeneous hardware).

At the same time, Algorithm 7 is the same method as Algorithm 5 from (Tyurin and Sivtsov, 2025), with the only difference that the local step sizes are larger in Algorithm 8. From this point, the convergence result of the method is a simple corollary of Theorem G.1. The proof is exactly the same as in Tyurin and Sivtsov (2025), which we include here for clarity.

Notice that we can take the following main branch:

$$
\begin{aligned}
x^0 &= w^0 \\
x^1 &= x^0 - \eta_g \nabla f(z_{1,0}^0; \zeta_{1,0}^0), \\
&\vdots \\
x^{M_1} &= x^{M_1-1} - \eta_g \nabla f(z_{1,M_1-1}^0; \zeta_{1,M_1-1}^0), \\
x^{M_1+1} &= x^{M_1} - \eta_g \nabla f(z_{2,0}^0; \zeta_{2,0}^0), \\
&\vdots \\
x^{M_1+M_2} &= x^{M_1+M_2-1} - \eta_g \nabla f(z_{2,M_2-1}^0; \zeta_{2,M_2-1}^0), \\
&\vdots \\
x^{\sum_{i=1}^n M_i} &= x^{\sum_{i=1}^n M_i-1} - \eta_g \nabla f(z_{n,M_n-1}^0; \zeta_{n,M_n-1}^0).
\end{aligned}
\tag{76}
$$

---

**Algorithm 7:** Decaying Local SGD (asynchronous version)

---

**Require:** Initial model $w^0$, step size $\eta_g$, parameter $b$
1: **for** $k = 0, 1, 2, \ldots$ **do**
2:    Broadcast $w^k$ to all workers
3:    **for** each worker $i \in [n]$ **in parallel do**
4:       Worker $i$ starts `LocalSGDWorker`$(w^k, \eta_g, b)$ from Algorithm 8
5:    **end for**
6:    Wait for the moment when $\sum_{i=1}^n M_i = b$     ($\{M_i\}$ from `LocalSGDWorker`$(w^k, \eta_g, b)$)
7:    Ask workers to stop running `LocalSGDWorker`$(w^k, \eta_g, b)$
8:    Aggregate $\eta_g \sum_{i=1}^n \sum_{j=0}^{M_i-1} \nabla f(z_{i,j}^k; \zeta_{i,j}^k)$ from the workers (e.g, via `AllReduce`)
9:    Update $w^{k+1} = w^k - \eta_g \sum_{i=1}^n \sum_{j=0}^{M_i-1} \nabla f(z_{i,j}^k; \zeta_{i,j}^k)$
10: **end for**

---

**Algorithm 8:** `LocalSGDWorker`$(w, \eta_g, b)$ in worker $i$ at round $k$

---

1: $z_{i,0}^k = w$
2: $M_i \leftarrow 0$
3: **while** True **do**
4:    Calculate step size $\eta_{M_i} = \sqrt{\frac{b-1}{(M_i+1)(\log(b-1)+1)}} \eta_g$
5:    $z_{i,M_i+1}^k = z_{i,M_i}^k - \eta_{M_i} \nabla f(z_{i,M_i}^k; \zeta_{i,M_i}^k), \quad \zeta_{i,M_i}^k \sim \mathcal{D}$
6:    $M_i = M_i + 1$
7: **end while**

---

Then, repeat the process for subsequent rounds. Notice that $x^0 = w^0, x^{\sum_{i=1}^n M_i} = w^1$, and so on.

**Theorem G.2.** *Let Assumptions 1.2 and 1.3 hold. Consider the computation tree of* Decaying Local SGD *(Algorithm 7), then* $\{x^k\}_{k \geq 0}$, *from (76) is a main branch and* $\frac{1}{K} \sum_{k=0}^{K-1} \mathbb{E}\left[\|\nabla f(x^k)\|^2\right] \leq \varepsilon$ *for all*

$$K \geq \frac{8bL\Delta}{\varepsilon} + \frac{16\sigma^2 L\Delta}{\varepsilon^2}.$$

*with step size* $\eta_g = \min\{\frac{1}{4bL}, \frac{\varepsilon}{8\sigma^2 L}\}$.

*Proof.* The corresponding auxiliary sequence can be inferred from (76): $(z^0, \xi^0) = (z_{1,0}^0, \zeta_{1,0}^0), \ldots, (z^{M_1}, \xi^{M_1}) = (z_{1,M_1}^0, \zeta_{1,M_1}^0)$, and etc. Condition 1 is satisfied because $\{\zeta_{i,j}^k\}$ are i.i.d., and by the construction (76). Condition 2 of Theorem G.1 holds because the same stochastic gradients used for computing $z^k$ are also used for $x^k$ (see Figure 8). Condition 3: notice that

$$\sup_{k \geq 0} \text{dist}(x^k, z^k) \leq b - 1$$

because the maximum number of edges to the common closest ancestor can not exit $b - 1$ (see Figure 8). Thus, $R = b - 1$ in Theorem G.1. Condition 4 holds due to the construction of the algorithm: $M_i$ is exactly the distance between the current point and the main branch.     $\square$

Notice that $b$, the total number of local steps, is a parameter. The question is how to choose it. Following the main part of the paper, we choose it to get the optimal time complexity (up to a logarithmic factor):

**Theorem G.3** (Proof in (Tyurin and Sivtsov, 2025))**.** *Consider Theorem G.2 and its conditions. Under Assumption 1.1, the total time complexity of* Decaying Local SGD *(Alg. 7) is*

$$\mathcal{O}\left(\tau \frac{L\Delta}{\varepsilon} + h\left(\frac{L\Delta}{\varepsilon} + \frac{L\sigma^2\Delta}{n\varepsilon^2}\right)\right)$$

*with* $b = \max\left\{\left\lceil \frac{\sigma^2}{\varepsilon} \right\rceil, 1\right\}$.

The choice of $R = \Theta\left(\frac{\sigma^2}{\varepsilon}\right)$ in Theorem G.1 seems to be a universal rule in all asynchronous and parallel algorithms for achieving optimal time complexities (Tyurin and Sivtsov, 2025).

### G.6 OTHER LOCAL AND ASYNCHRONOUS ALGORITHMS

Tyurin and Sivtsov (2025) provide many other algorithms with different computation and communication properties (see their Table 1). All these algorithms can be improved in the aspect we have previously discussed. Their local steps, not related to the main branches, can be increased according to the rule described in Condition 4. Then, nothing else needs to be changed, and the derived results still hold.

## G.7 PROOF OF THEOREM G.1

**Theorem G.1** (Main Theorem). *Let Assumptions 1.2 and 1.3 hold. Consider any* SGD *method represented by* computation tree $G = (V, E)$. *Let* $\{x^k\}_{k \geq 0}$ *be a* main branch *of* $G$ *and* $\{(\gamma_k, z^k, \xi^k)\}_{k \geq 0}$ *be the corresponding* auxiliary sequence *(see Def. G.1) that satisfy the following conditions:*
**Condition 1:** *For all* $k \geq 0$, $\xi^k$ *is statistically independent of* $\{(x^{i+1}, z^{i+1}, \xi^i)\}_{i=0}^{k-1}$.
**Condition 2:** *The representation of* $z^k$ *is contained within that of* $x^k$, *i.e.,* $\mathrm{repr}(z^k) \subseteq \mathrm{repr}(x^k)$ *for all* $k \geq 0$. *Equivalently, all stochastic gradients used in the computation of* $z^k$ *are also utilized in calculating* $x^k$.
**Condition 3:** *There exists a constant* $R \in [0, \infty]$ *such that* $\mathrm{dist}(x^k, z^k) \leq R$ *for all* $k \geq 0$.
**Condition 4:** *The step sizes along the main branch satisfy* $\gamma_k = \gamma_g := \min\{\frac{1}{2L}, \frac{1}{4RL}, \frac{\varepsilon}{8\sigma^2 L}\}$ *for all* $k \geq 0$. *Any other step size* $\gamma$ *can be taken as large as[17]* $\sqrt{\frac{R}{(j+1)(\log R+1)}}\gamma_g$, *where* $j = \mathrm{dist}(y, \{x^k\})$ *(from Def. G.3) and* $y$ *is the node at which the step* $y - \gamma \nabla f(\cdot; \cdot)$ *with this step size* $\gamma$ *is applied to find a new node.*
*Then* $\frac{1}{K} \sum_{k=0}^{K-1} \mathbb{E}\left[\|\nabla f(x^k)\|^2\right] \leq \varepsilon$ *for all*

$$K \geq \frac{8(R+1)L\Delta}{\varepsilon} + \frac{16\sigma^2 L\Delta}{\varepsilon^2}.$$

The following proof closely follows the proof of Theorem 2.4 from (Tyurin and Sivtsov, 2025), with the difference that we have to work with different step sizes and some essential changes that we will highlight.

*Proof.* Using Assumption 1.2 and Condition 4, we have $\gamma_k = \gamma_g$ and

$$f(x^{k+1}) \leq f(x^k) - \gamma_g \left\langle \nabla f(x^k), \nabla f(z^k; \xi^k) \right\rangle + \frac{L\gamma_g^2}{2} \left\| \nabla f(z^k; \xi^k) \right\|^2$$

for $x^{k+1} = x^k - \gamma_k \nabla f(z^k; \xi^k) = x^k - \gamma_g \nabla f(z^k; \xi^k)$. Due to Condition 1 and the variance decomposition equality, $\xi^k$ is statistically independent of $(x^k, z^k)$ and

$$\mathbb{E}_k\left[f(x^{k+1})\right] \leq f(x^k) - \gamma_g \left\langle \nabla f(x^k), \nabla f(z^k) \right\rangle + \frac{L\gamma_g^2}{2}\mathbb{E}_k\left[\left\|\nabla f(z^k; \xi^k)\right\|^2\right]$$

$$= f(x^k) - \gamma_g \left\langle \nabla f(x^k), \nabla f(z^k) \right\rangle + \frac{L\gamma_g^2}{2} \left\|\nabla f(z^k)\right\|^2 + \frac{L\gamma_g^2}{2}\mathbb{E}_k\left[\left\|\nabla f(z^k; \xi^k) - \nabla f(z^k)\right\|^2\right]$$

$$\leq f(x^k) - \gamma_g \left\langle \nabla f(x^k), \nabla f(z^k) \right\rangle + \frac{L\gamma_g^2}{2} \left\|\nabla f(z^k)\right\|^2 + \frac{L\gamma_g^2\sigma^2}{2},$$

where $\mathbb{E}_k\left[\cdot\right]$ is the expectation conditioned on $(x^k, z^k)$. In the last inequality, we use Assumption 1.3. Rewriting the dot product and using $\gamma_g \leq \frac{1}{2L}$, we obtain

$$\mathbb{E}_k\left[f(x^{k+1})\right]$$

$$\leq f(x^k) - \frac{\gamma_g}{2}\left(\left\|\nabla f(x^k)\right\|^2 + \left\|\nabla f(z^k)\right\|^2 - \left\|\nabla f(x^k) - \nabla f(z^k)\right\|^2\right) + \frac{L\gamma_g^2}{2}\left\|\nabla f(z^k)\right\|^2 + \frac{L\gamma_g^2\sigma^2}{2}$$

$$\leq f(x^k) - \frac{\gamma_g}{2}\left\|\nabla f(x^k)\right\|^2 - \frac{\gamma_g}{4}\left\|\nabla f(z^k)\right\|^2 + \frac{\gamma_g}{2}\left\|\nabla f(x^k) - \nabla f(z^k)\right\|^2 + \frac{L\gamma_g^2\sigma^2}{2}. \tag{77}$$

We now focus on $\left\|\nabla f(x^k) - \nabla f(z^k)\right\|^2$. Using Assumption 1.2, we obtain

$$\left\|\nabla f(x^k) - \nabla f(z^k)\right\|^2 \leq L^2 \left\|x^k - z^k\right\|^2. \tag{78}$$

Notice that there exist $p \in \{0, \ldots, k\}$ and the closest common ancestor $x^p$ to $x^k$ and $z^k$ such that

$$x^k = x^p - \gamma_g \sum_{i=p}^{k-1} \nabla f(z^i; \xi^i)$$

---

[17]For $R = 0$, we use the standard convention $\frac{0}{\log 0 + 1} = 0$.

and

$$z^k = x^p - \sum_{(\gamma, w, \xi) \in S^k} \gamma \nabla f(w; \xi),$$

where $S^k$ is the set of points and random variables used to compute $z^k$ starting from $x^p$ (see Figure 6). Moreover, due to Condition 3, we have $\text{dist}(x^k, z^k) \leq \max\{k - p, |S^k|\} \leq R$, meaning $p \geq k - R$ and $|S^k| \leq R$. In total,

$$k \geq p \geq k - R \tag{79}$$

and

$$|S^k| \leq R, \tag{80}$$

which we use later. Condition 2 assumes

$$\text{repr}(z^k) := \underbrace{\{(z^i; \xi^i)\}_{i=0}^{p-1}}_{A} \uplus \underbrace{\{(w; \xi)\}_{(\gamma, w, \xi) \in S^k}}_{C}$$

$$\subseteq \text{repr}(x^k) := \underbrace{\{(z^i; \xi^i)\}_{i=0}^{p-1}}_{A} \uplus \underbrace{\{(z^i; \xi^i)\}_{i=p}^{k-1}}_{B},$$

where $\uplus$ is the multiset union operation. Thus

$$\underbrace{\{(w; \xi)\}_{(\gamma, w, \xi) \in S^k}}_{C} \subseteq \underbrace{\{(z^i; \xi^i)\}_{i=p}^{k-1}}_{B}. \tag{81}$$

(Starting from this point, our proof and the proof by (Tyurin and Sivtsov, 2025) diverge). Using Jensen's inequality and (78),

$$\left\| \nabla f(x^k) - \nabla f(z^k) \right\|^2 \leq L^2 \left\| \gamma_g \sum_{i=p}^{k-1} \nabla f(z^i; \xi^i) - \sum_{(\gamma, w, \xi) \in S^k} \gamma \nabla f(w; \xi) \right\|^2$$

$$\leq 4L^2 \left\| \gamma_g \sum_{i=p}^{k-1} (\nabla f(z^i; \xi^i) - \nabla f(z^i)) \right\|^2 + 4L^2 \left\| \sum_{(\gamma, w, \xi) \in S^k} \gamma (\nabla f(w; \xi) - \nabla f(w)) \right\|^2$$

$$+ 4L^2 \left\| \gamma_g \sum_{i=p}^{k-1} \nabla f(z^i) \right\|^2 + 4L^2 \left\| \sum_{(\gamma, w, \xi) \in S^k} \gamma \nabla f(w) \right\|^2.$$

Using Assumption 1.3 and since $\xi^k$ is statistically independent of $\{(z^{i+1}, \xi^i)\}_{i=0}^{k-1}$ for all $k \geq 0$ (Condition 1), we have

$$\mathbb{E}\left[ \left\| \gamma_g \sum_{i=p}^{k-1} (\nabla f(z^i; \xi^i) - \nabla f(z^i)) \right\|^2 \right] \leq \gamma_g^2 (k - p) \sigma^2.$$

Moreover, due to (81), $\sum_{(\gamma, w, \xi) \in S^k} \gamma (\nabla f(w; \xi) - \nabla f(w))$ is a subtotal of $\sum_{i=p}^{k-1} (\nabla f(z^i; \xi^i) - \nabla f(z^i))$ and we can use the same reasoning as in the previous inequality:

$$\mathbb{E}\left[ \left\| \sum_{(\gamma, w, \xi) \in S^k} \gamma (\nabla f(w; \xi) - \nabla f(w)) \right\|^2 \right] \leq \sigma^2 \sum_{(\gamma, w, \xi) \in S^k} \gamma^2.$$

In total,

$$\mathbb{E}\left[ \left\| \nabla f(x^k) - \nabla f(z^k) \right\|^2 \right] \leq 4L^2 \gamma_g^2 \sigma^2 (k - p) + 4L^2 \sigma^2 \sum_{(\gamma, w, \xi) \in S^k} \gamma^2$$

$$+ 4L^2 \mathbb{E}\left[\left\|\gamma_g \sum_{i=p}^{k-1} \nabla f(z^i)\right\|^2\right] + 4L^2 \mathbb{E}\left[\left\|\sum_{(\gamma,w,\xi)\in S^k} \gamma \nabla f(w)\right\|^2\right].$$

Using Lemma E.8,

$$\mathbb{E}\left[\left\|\nabla f(x^k) - \nabla f(z^k)\right\|^2\right] \leq 4L^2\gamma_g^2\sigma^2(k-p) + 4L^2\sigma^2 \sum_{(\gamma,w,\xi)\in S^k} \gamma^2$$

$$+ 4L^2\gamma_g^2(k-p)\sum_{i=p}^{k-1}\mathbb{E}\left[\left\|\nabla f(z^i)\right\|^2\right] + 4L^2\left(\sum_{(\gamma,w,\xi)\in S^k}\gamma^2\right)\sum_{(\gamma,w,\xi)\in S^k}\mathbb{E}\left[\left\|\nabla f(w)\right\|^2\right]. \tag{82}$$

We now bound the sum $\sum_{(\gamma,w,\xi)\in S^k} \gamma^2$. Notice that

$$\sum_{(\gamma,w,\xi)\in S^k} \gamma^2 \leq \sum_{j=0}^{|S^k|-1} \frac{R}{(j+1)(\log R + 1)}\gamma_g^2$$

due to Condition 4. See also Figure 6, which visualizes the set $\{\gamma\}_{(\gamma,w,\xi)\in S^k} = \{\hat{\gamma}, \bar{\gamma}, \ldots, \tilde{\gamma}\}$, where $j = \text{dist}(x^p, \{x^k\}) = 0$ corresponds to $\hat{\gamma}$, $j = 1$ corresponds to $\bar{\gamma}, \ldots, j = |S^k| - 1$ corresponds to $\tilde{\gamma}$. Thus,

$$\sum_{(\gamma,w,\xi)\in S^k} \gamma^2 \leq \frac{\gamma_g^2 R}{(\log R + 1)}\sum_{j=1}^{|S^k|}\frac{1}{j} \leq \frac{\gamma_g^2 R(\log|S^k| + 1)}{(\log R + 1)} \overset{(80)}{\leq} \gamma_g^2 R,$$

where the second inequality due to the standard inequality $\sum_{j=1}^m \frac{1}{j} \leq \log m + 1$ for all $m \geq 1$. For the corner case $R = 0$, the inequalities also hold under the standard convention $\frac{0}{\log 0 + 1} = 0$. Due to the last bound and (79), (82) yields

$$\mathbb{E}\left[\left\|\nabla f(x^k) - \nabla f(z^k)\right\|^2\right] \leq 4L^2\gamma_g^2\sigma^2 R + 4L^2\sigma^2\gamma_g^2 R$$

$$+ 4L^2\gamma_g^2 R\sum_{i=p}^{k-1}\mathbb{E}\left[\left\|\nabla f(z^i)\right\|^2\right] + 4L^2\gamma_g^2 R\sum_{(\gamma,w,\xi)\in S^k}\mathbb{E}\left[\left\|\nabla f(w)\right\|^2\right]$$

$$= 8L^2\gamma_g^2\sigma^2 R$$

$$+ 4L^2\gamma_g^2 R\sum_{i=p}^{k-1}\mathbb{E}\left[\left\|\nabla f(z^i)\right\|^2\right] + 4L^2\gamma_g^2 R\sum_{(\gamma,w,\xi)\in S^k}\mathbb{E}\left[\left\|\nabla f(w)\right\|^2\right].$$

Since (81), $\sum_{(\gamma,w,\xi)\in S^k}\mathbb{E}\left[\left\|\nabla f(w)\right\|^2\right] \leq \sum_{i=p}^{k-1}\mathbb{E}\left[\left\|\nabla f(z^i)\right\|^2\right]$ and

$$\mathbb{E}\left[\left\|\nabla f(x^k) - \nabla f(z^k)\right\|^2\right] \leq 8L^2\gamma_g^2\sigma^2 R + 8L^2\gamma_g^2 R\sum_{i=p}^{k-1}\mathbb{E}\left[\left\|\nabla f(z^i)\right\|^2\right]$$

$$\leq 8L^2\gamma_g^2\sigma^2 R + 8L^2\gamma_g^2 R\sum_{i=k-R}^{k-1}\mathbb{E}\left[\left\|\nabla f(z^i)\right\|^2\right],$$

where the last inequality due to (79). Substituting this inequality to (77) and taking the full expectation, we obtain

$$\mathbb{E}\left[f(x^{k+1})\right] \leq \mathbb{E}\left[f(x^k)\right] - \frac{\gamma_g}{2}\mathbb{E}\left[\left\|\nabla f(x^k)\right\|^2\right] - \frac{\gamma_g}{4}\mathbb{E}\left[\left\|\nabla f(z^k)\right\|^2\right] + \frac{L\gamma_g^2\sigma^2}{2}$$

$$+ \frac{\gamma_g}{2}\left(8L^2\gamma_g^2 R\sum_{j=k-R}^{k-1}\mathbb{E}\left[\left\|\nabla f(z^j)\right\|^2\right] + 8L^2\gamma_g^2 R\sigma^2\right)$$

$$\leq \mathbb{E}\left[f(x^k)\right] - \frac{\gamma_g}{2}\mathbb{E}\left[\left\|\nabla f(x^k)\right\|^2\right] - \frac{\gamma_g}{4}\mathbb{E}\left[\left\|\nabla f(z^k)\right\|^2\right] + 2L\gamma_g^2\sigma^2$$

$$+ 4L^2\gamma_g^3 R \sum_{j=k-R}^{k-1}\mathbb{E}\left[\left\|\nabla f(z^j)\right\|^2\right] \tag{83}$$

because $\gamma_g \leq \frac{1}{4RL}$. Note that $\sum_{k=0}^{K-1}\sum_{j=k-R}^{k-1}\mathbb{E}\left[\left\|\nabla f(z^j)\right\|^2\right] \leq R\sum_{k=0}^{K-1}\mathbb{E}\left[\left\|\nabla f(z^k)\right\|^2\right]$. Thus, summing (83) for $k = 0, \ldots, K-1$ and substituting $f^*$,

$$\mathbb{E}\left[f(x^K) - f^*\right] \leq f(x^0) - f^* - \frac{\gamma_g}{2}\sum_{k=0}^{K-1}\mathbb{E}\left[\left\|\nabla f(x^k)\right\|^2\right] - \frac{\gamma_g}{4}\sum_{k=0}^{K-1}\mathbb{E}\left[\left\|\nabla f(z^k)\right\|^2\right] + 2KL\gamma_g^2\sigma^2$$

$$+ 4L^2\gamma_g^3 R^2 \sum_{k=0}^{K-1}\mathbb{E}\left[\left\|\nabla f(z^k)\right\|^2\right]$$

$$\leq f(x^0) - f^* - \frac{\gamma_g}{2}\sum_{k=0}^{K-1}\mathbb{E}\left[\left\|\nabla f(x^k)\right\|^2\right] + 2KL\gamma_g^2\sigma^2$$

because $\gamma_g \leq \frac{1}{4LR}$. Finally, since $\mathbb{E}\left[f(x^K) - f^*\right] \geq 0$,

$$\frac{1}{K}\sum_{k=0}^{K-1}\mathbb{E}\left[\left\|\nabla f(x^k)\right\|^2\right] \leq \frac{2\Delta}{K\gamma_g} + 4L\gamma_g\sigma^2.$$

It is left to use that $\gamma_g = \min\{\frac{1}{2L}, \frac{1}{4RL}, \frac{\varepsilon}{8\sigma^2 L}\}$ and the bound on $K$ from the theorem statement. $\square$

