# OpenReview forum: "Local SGD and Federated Averaging Through the Lens of Time Complexity"
_ICLR.cc/2026/Conference — Submitted to ICLR 2026_

### Official Review · Reviewer_ZKTy · 2025-10-16

**Soundness:** 2
**Presentation:** 3
**Contribution:** 3
**Rating:** 8
**Confidence:** 4

**Summary:**

The paper studies Local SGD and its variants. It begins by showing a lower bound for the canonical Local SGD algorithm. Then, the authors proposed several new variants of Local SGD to avoid the term in the rate of the canonical Local SGD. Finally, they propose Decaying Local SGD with new stepsize rules.

**Strengths:**

The paper first establishes a lower bound on Local SGD of order $\Omega (\epsilon ^{-3/2})$. This term seems to appear in most analyses of Local SGD, and it is now justified by this lower bound result.

The authors then investigated two stepsizes and dual Local SGD, and proposed a $\sqrt {n}$-times larger global aggregating stepsize in Local SGD. Though the idea of using a different, larger global stepsize is not new and already appears for instance in (Karimireddy et al., 2020), the analysis in this paper is the refined version over (Yang et al. (2021) and Karimireddy et al. (2020)).

They finally proposed a new Decaying Local SGD method, with a new stepsize rule. This can lead to much faster convergence in some cases, and can also be generalized to asynchronous and tree-graph problems.

The paper addresses several non-trivial technical issues of Local SGD. The writing is clean and easy to follow. The results are described in a reasonable way.

I checked the proof of the lower bound of Local SGD, and I looked at the analysis of Dual Local SGD. I did not spot any proof mistake.

**Weaknesses:**

The only weakness is the experimental results seem to be less convincing. I understand that Local SGD is a very robust baseline method. But I still expect the Dual Local SGD can be faster under the conditioning that the additional $O(\epsilon ^{-3/2})$ term dominates.
It would be great to show in some experiments that, at least in some conditionings, the Local SGD algorithm really gets slowed down in practice due to this term. This would also help to better motivate the entire story of the paper. Otherwise, now the paper looks like addressing a merely theoretical issue.

**Questions:**

Is it reasonable to add a Minibatch SGD in the experiment plots? The algorithm proposed in this paper has the same theoretical rate as Minibatch SGD, but it would be great to know how they compare to each other in practice.

This is not really a question but a small suggestion. It seems clearer to rewrite the lower bound of (Tyurin and Richtárik, 2024) in Table 2 as $\Omega \left( \min \left\\{ \tau \frac {L \Delta} {\epsilon}, h \frac {L \sigma^2 \Delta} {\epsilon^2} \right\\} + h \left( \frac {L \Delta} {\epsilon} + \frac {L \sigma^2 \Delta} {n \epsilon^2} \right) \right)$. Now, it seems clearer that the red term in your lower bound is the average over the minimum terms in their algorithmic-independent lower bound.

---

> ### Author Response · Authors · 2025-11-20
>
> Dear Reviewer ZKTy,
>
> Thank you very much for your careful review and valuable feedback.
>
> We address your points below:
>
> > _The only weakness is the experimental results seem to be less convincing ..._
>
> As we explain in the paper, the canonical Local SGD method remains one of the most widely evaluated and tested algorithms in distributed and federated learning, and there is no doubt that it is a strong method. It is a very non-trivial and consuming task to find a real example where we would observe the gap. But even if we find a dataset and a problem, it would not be clear whether the gap is due to $\\varepsilon$ vs. $\\varepsilon^{3/2}$ or to some intrinsic characteristics of the dataset and problem.
>
> > _Is it reasonable to add a Minibatch SGD in the experiment plots? The algorithm proposed in this paper has the same theoretical rate as Minibatch SGD, but it would be great to know how they compare to each other in practice._
>
> We included Minibatch SGD in the experiment plots and an additional paragraph discussing the obtained results.
>
> Concerning your small suggestion, we updated table 2 accordingly.

---

> > ### Comment · Reviewer_ZKTy · 2025-11-26
> >
> > Thank you for the response. I am willing to keep my current score.

---

### Official Review · Reviewer_8mzB · 2025-11-01

**Soundness:** 3
**Presentation:** 3
**Contribution:** 3
**Rating:** 6
**Confidence:** 3

**Summary:**

This paper shows that while Local SGD (FedAvg) seems efficient in theory, it’s actually slower in real time than simpler methods like Minibatch or single-worker SGD once communication costs are included. The authors fix this with Dual Local SGD and Decaying Local SGD, which use better scaling and adaptive step sizes to reach optimal time efficiency. Experiments on MNIST and CIFAR-10 confirm these tweaks make training faster and more stable.

**Strengths:**

1. This paper analyzes Local SGD and FedAvg through time complexity, not just iteration count, offering a more realistic view of efficiency.

2. Challenges the common belief that local updates always save time, providing a clearer picture for real-world federated training.

**Weaknesses:**

1. In real distributed or federated systems, communication often overlaps with computation, and network latency can vary significantly. This makes the proposed time complexity framework mathematically elegant but not fully representative of real wall-clock behavior.

2. All experiments are conducted on small benchmarks (toy problem, MNIST, CIFAR-10) with simulated workers. The paper does not demonstrate wall-clock speedup or test on realistic federated environments (e.g., cross-device, mobile, or edge networks).

3.The “decaying local step” rule assumes that increasing local step sizes never harms stability up to logarithmic factors, but this is not proved for strongly nonconvex objectives. The bound holds only asymptotically with carefully tuned constants. In practice, large step sizes can cause divergence, contradicting the theoretical “never worse” claim.

**Questions:**

N/A

---

> ### Author Response · Authors · 2025-11-20
>
> Dear Reviewer 8mzB,
>
> Thank you for your thoughtful review and constructive feedback.
>
> As for your concerns:
>
> > _All experiments are conducted on small benchmarks (toy problem, MNIST, CIFAR-10) with simulated workers. The paper does not demonstrate wall-clock speedup or test on realistic federated environments (e.g., cross-device, mobile, or edge networks)._
>
> We updated the plots, they now show the “time” (in seconds) on the $x$-axis. Notice that our goal was to compare the methods in controlled environments. Unlike the controlled environments, the realistic environments include a lot of noise that would not allow us to see the real difference between the methods.
>
> > _3.The “decaying local step” rule assumes that increasing local step sizes never harms stability up to logarithmic factors, but this is not proved for strongly nonconvex objectives. The bound holds only asymptotically with carefully tuned constants. In practice, large step sizes can cause divergence, contradicting the theoretical “never worse” claim._
>
> Following the proof of Theorem 3.1, the local stepsizes $\\{\\eta\_{\\ell}\\}\_{0 \\le \\ell \\le K - 1}$ satisfy (see eq. (17), lines 1018)
>
> $$\\sum\_{\\ell = 0}^{K - 1} \\eta_{\\ell}^2 \\lesssim \\eta\_g^2 b, \\quad (2)$$
>
> which is enough to establish iteration and time complexity guarantees. Our claim that the new stepsize rule is _never worse up to logarithmic factors_ (l. 413-414) does not assert that increasing the local stepsizes is always harmless, but rather that within the theoretically valid regime given by (2), the adaptive rule is never worse (up to logarithmic factors) compared with the canonical choice $\\eta\_{\\ell} = \\sqrt{n}\\,\\eta\_g$ made earlier. Stepsizes outside this regime may lead to divergence, which falls beyond the conditions of our proofs.
>
> Moreover, we would like to emphasize that our analysis provides uniform convergence guarantees for a broad class of local stepsize schedules--including constant, decreasing, and oscillating schemes--as long as (2) is satisfied. While the theory of the paper does not reflect all practical behaviours of these schedules, it does establish a common set of assumptions under which all such variants remain stable and satisfy the same iteration/time complexity guarantees.

---

### Official Review · Reviewer_K8KK · 2025-11-03

**Soundness:** 2
**Presentation:** 3
**Contribution:** 2
**Rating:** 4
**Confidence:** 4

**Summary:**

This work studied local SGD in terms of time complexity. It shows that under a certain computation time framework, canonical Local SGD can be strictly suboptimal in time complexity compared to Minibatch SGD and HeroSGD. To overcome the issue, the paper proposed Dual Local SGD and proved they achieve optimal complexities (up to log factors) under the same model. Experiments demonstrate the effectiveness of the proposed algorithms.

**Strengths:**

1. Time complexity perspective is a meaningful metric in practice, which is more useful in algorithm design.
2. The observation that Local SGD is strictly worse (under the claimed computation framework) is interesting.
3. The proposed fix for Local SGD is simple and elegant, with a solid theoretical result supporting their effectiveness.
4. The writing of the work is clear in general, the flow of the story is easy to follow.

**Weaknesses:**

1. Assumption 1.1 characterized a fixed computation model, which is not fully verified yet. For example, here the gradient computation (for local SGD) always take only one sample, for small batch scenarios, some fixed time cost (for things like kernel launch) may take effect or even dominate the computation. Similarly for the communication, there should be some existing models in that community (like Postal model or other latency models) which characterized the communication/broadcast time more accurately. It would strengthen the paper if the authors could clarify the intended scope of this model.
2. Confusing lower bound results. From the proof, the results are a combination of lower iteration complexity bounds and Assumption 1.1 (lower iteration bound $\times$ execution time), but if I understand it correctly, Assumption 1.1 highlights "at most" for each execution, that means 'execution time' $\leq$ something, which I think is confusing to imply a lower bound (bound $\geq$ something). So to improve the clarity, I think Assumption 1.1 can be further clarified, e.g., the execution time are just exactly $h$ (and $\tau$).
3. The experiment part seems to still use the "rounds" in the $x$-axis, but the work highlight time complexity, maybe it is better to attach more results in terms of wallclock time. Also the experiment does not echo the observation that Local SGD is strictly worse than Minibatch-SGD and Hero-SGD, it would strengthen the work if there are corresponding empirical results.
4. Writing. Missing $O(\cdot)$ (and $\Omega) notations for many expressions throughout the work.

**Questions:**

See above.

---

> ### Author Response · Authors · 2025-11-20
>
> Dear reviewer K8KK,
>
> Thank you for your thoughtful review and valuable feedback.
>
> Concerning the points you raised:
>
> > _Assumption 1.1 characterized a fixed computation model, which is not fully verified yet. For example, here the gradient computation (for local SGD) always take only one sample, for small batch scenarios, some fixed time cost (for things like kernel launch) may take effect or even dominate the computation. Similarly for the communication, there should be some existing models in that community (like Postal model or other latency models) which characterized the communication/broadcast time more accurately. It would strengthen the paper if the authors could clarify the intended scope of this model._
>
> Of course, it does not hold in practice, where the real computation and communication times are chaotic and random, and we agree that this assumption does not capture some other subroutines.
>
> However, while it is not an exact representation, it is very close to what happens in real systems. Indeed, there is no doubt in the ML community that gradient computations (backpropagation) are the main bottleneck in local computations; all other operations are negligible compared to them, so our model is reasonable. In the communication part, there are many steps, including preparing and compressing messages by workers, sending them, receiving them, and decompressing them by the server. The time required to do all these steps can be represented by one value $\tau,$ which we believe is a reasonable and elegant approach.
>
> This and similar assumptions were very useful in numerous papers (e.g. [1,2,3,4] and many more). For instance, under a similar assumption, [1] showed that Asynchronous SGD is provably faster than Minibatch SGD. [2] proved the tight complexities for asynchronous methods, and so forth. What we aim to explain in this paper is that iteration complexity is the wrong metric, and that the appropriate one is time complexity, and this is natural as in practice, we are often interested in which methods is the _fastest_ to optimize a given objective.
>
>
> [1]: Mishchenko, K., Bach, F., Even, M., and Woodworth, B. (2022). Asynchronous SGD beats minibatch
> SGD under arbitrary delays. Advances in Neural Information Processing Systems.
>
> [2]: Tyurin, A. and Richtárik, P. (2023). Optimal time complexities of parallel stochastic optimization
> methods under a fixed computation model. Advances in Neural Information Processing Systems,
> 36.
>
> [3]: Tyurin, A. and Richtárik, P. (2024). On the optimal time complexities in decentralized stochastic
> asynchronous optimization. Advances in Neural Information Processing Systems, 37
>
> [4]: Maranjyan, A., Tyurin, A., and Richtárik, P. (2025). Ringmaster ASGD: The first asynchronous SGD
> with optimal time complexity. In International Conference on Machine Learning.
>
> > _Confusing lower bound results. From the proof, the results are a combination of lower iteration complexity bounds and Assumption 1.1 (lower iteration bound $\\times$ execution time), but if I understand it correctly, Assumption 1.1 highlights "at most" for each execution, that means 'execution time' $\\le$ something, which I think is confusing to imply a lower bound (bound $\\ge$ something). So to improve the clarity, I think Assumption 1.1 can be further clarified, e.g., the execution time are just exactly $h$ (and $\\tau$)._
>
> We agree with the reviewer that the wording of Assumption 1.1 may cause confusion regarding how the lower bounds are derived. As originally stated, the assumption allowed computation and communication times to vary up to fixed thresholds, which can obscure the interpretation of the lower bounds obtained by multiplying iteration complexity by execution time. Under that formulation, the resulting lower bounds should indeed be read as _worst-case_ lower bounds.
>
> To remove this ambiguity, we have updated the paper and clarified Assumption 1.1 by replacing the “at most" wording with an exact computation time $h$ and communication time $\\tau$. This makes the connection between iteration complexity and time complexity fully transparent. We hope this revision resolves the confusion and improves the clarity of the lower-bound argument.

---

> > ### Author Response · Authors · 2025-11-20
> >
> > > _The experiment part seems to still use the "rounds" in the $x$-axis, but the work highlight time complexity, maybe it is better to attach more results in terms of wallclock time. Also the experiment does not echo the observation that Local SGD is strictly worse than Minibatch-SGD and Hero-SGD, it would strengthen the work if there are corresponding empirical results._
> >
> > We updated the plots, they now show the “time” (in seconds) on the $x$-axis. Although this is visually more convenient, under Assumption 1.1 (fixed computation $h$ and communication $\\tau$ times), one iteration of Local SGD (and all local methods considered in the paper) takes exactly $\\tau + K h$ seconds. Therefore, up to an affine rescaling of the $y$-axis, the plots are the same.
> >
> > We want to stress the canonical Local SGD method (FedAvg) remains one of the most widely evaluated and tested algorithms in distributed and federated learning, and there is no doubt that it is a strong method for practical optimization tasks. Our experiments confirm this. The main goal of this paper is to explain that the previous **theoretical comparison** in Table 1 might be misleading, and a better one is presented in Table 2.
> >
> > > _Writing. Missing $O(\\cdot)$ (and $\\Omega(\\cdot)$) notations for many expressions throughout the work._
> >
> > We thank the reviewer for pointing this out. We have corrected the issue by adding the missing $O(\\cdot)$ and $\\Omega(\\cdot)$ notations throughout the paper, in particular in the statements of Theorems 2.1, 2.3, 2.5, and 3.1. We hope this makes the presentation clearer rather than repeatedly stating “up to constant factors.”

---

### Official Review · Reviewer_YJY4 · 2025-11-03

**Soundness:** 1
**Presentation:** 2
**Contribution:** 2
**Rating:** 2
**Confidence:** 5

**Summary:**

This paper reinterprets existing convergence analyses of Local SGD in a time complexity model, aiming to demonstrate that it cannot beat the time complexity of simple baselines, such as mini-batch SGD and single-machine SGD. The paper then proceeds to analyze a two-step-size version of Local SGD, and shows that it can match the convergence guarantee of mini-batch SGD in the homogeneous non-convex setting. Finally, it analyzes another step-size schedule for SGD with two step-sizes, showing a similar convergence rate. Experiments on MNIST and CIFAR-10 are performed to compare Local SGD and its two-step-size variants.

**Strengths:**

The point of view of time complexity is indeed interesting, as it can potentially highlight that regimes where we previously thought Local SGD could beat the baselines actually do not exist once we account for computation and communication costs. Thus, morally, the paper is asking interesting questions.

**Weaknesses:**

Some of the proofs in the paper are incorrect, and even the ones that are correct are either not entirely novel or not particularly surprising.

- Theorems 2.3 and 2.5 are wrong. First of all, since these results are derived from upper bounds on Local SGD's convergence, they do not represent time complexity lower bounds. Thus, the theorem statements themselves are wrong. The paper acknowledges this in the preceding text, but still chooses to write the incorrect version of these theorems. Furthermore, the proof of Theorem 2.5 is incorrect because it overlooks specific terms in the upper bounds when deriving the theorem. If the results being used were lower bounds, this would have been fine, but these results are upper bounds, and hence ignoring some terms would not give a correct picture: ignoring these terms makes the upper bounds invalid. Since these are the data heterogeneity terms, this is further potentially misleading in the heterogeneous setting.

- In the convex setting, i.e., when comparing Theorems 2.1 and 2.2, we do not have the above issue, as the authors do look at the lower bound due to Glasgow et al. for Local SGD. Still, I am not convinced that conclusion is correct in all regimes as the authors claim. Please answer my question below regarding this.

- The paper is inconsistent about stating and discussing the resuls for Hero SGD/single machine SGD. For instance, in Algorithm 3 we look at an algorithm with $R$ iterations, as opposed to $KR$ iterations which would be the correct baseline to compare to. But when discussing the algorithm's guarantee in Table 1, the authors mix the different terms from these two algorithms: the optimization term seems to follow Algorithm 3, while the noise term for Hero SGD seems to suggest $KR$ iterations. This is most probably a typo or misunderstanding about which single machine baseline is the correct comparison. Surprisingly in Theorem 2.2, the authors revert back to the correct baselines, with $KR$ iterations. This is all very confusing, and would likely mislead an uninformed reader. I suggest describing in the pseducode only the variant with $KR$ iterations, because this equalizes the compute each machine actually does in the parallel algorithms (Local SGD and mini-batch SGD).

- Morally, we don't need to look at time complexity to conclude that in the homogeneous setting Local SGD has a limited utility. The seminal result of [Woodworth et al.](https://arxiv.org/abs/2102.01583) already shows that accelerated mini-batch SGD and accelerated single machine SGD together are min-max optimal. Yes, this leaves open the question whether a similar dichotomy can be shown for non accelerated algorithms, but it is not a very interesting gap. Even if in principle, unaccelerated Local SGD could match the min-max optimal rate, that would still not explain its practical effectiveness. More than anything, their lower bound doesn't show that Local SGD is a bad algorithm, but that rather the homogeneous convex smooth setting is not "rich" enough to actually benefit from local update steps. Infact, this is the motivation for studying mildly heterogeneous settings, where the benefit of collaboration is not limited to just variance reduction in the noise term, but collaboration can genuinely help clients explore the loss landscape. The results of [Patel et al.](https://arxiv.org/abs/2405.11667) highlight this, as they show that the dichotomy pointed out by Woodworth et al. does not really hold in the heterogeneous setting, because single machine SGD is no longer a reasonable baseline.  Finally, this highlights that several important references are missing in this discussion. I would encourage the authors to re-write with a more through inspection of what the current results imply in the accereated and non-accelerated settings. The thesis by [Patel](https://arxiv.org/abs/2507.00195) does contain many useful missing references.

-  Theorem 3.1 is not interesting. It makes a point, which has been underlined again and again (by some of the works cited in the paper itself): Local SGD with two step-sizes can recover mini-batch SGD. It is not any more insghtful to show this for a specific choice of step-sizes. Granted, the step-sizes for which Theorem 3.1 shows this are not extreme, but even the [SCAFFOLD paper's Theorem 1](https://arxiv.org/pdf/1910.06378) already shows this. What is the novelty here? What would be interesting is if the authors could identify a theoretical setting, where Local SGD with two step-sizes not only matches but beats both mini-batch and single-machine SGD. Similarly, the fact that there is a gap between Local SGD with one and two step-sizes was also formalized again by the lower bound instances of [Patel et al.](https://arxiv.org/pdf/2405.11667) (see their Remark 5).

- It is unclear what is the point of Theorem 4.1, it attains the same convergence rate as Theorem 3.1. If the point is to show that a bigger step-size schedule does not hurt or helps in some way, then clearly the theoretical setting considered in the paper is not rich enough to demonstrate that. Empricially, previous works have already investigated this quite extensively, for instance [Charles and Konecny](https://arxiv.org/abs/2007.00878) do this on several benchmarks. So what would be interesting is to consider some theoretical setting, where such a scheme could have provable optimization benefits.

**Questions:**

While the proofs of Theorems 2.1 and 2.2 look correct to me, I am not convinced that the time complexities of mini-batch SGD/single machine SGD are always at least as good as Local SGD's time complexity. If this point were being made for accelerated algorithms, I would be convinced, given the results of Woodworth et al. in the oracle complexity model. However, for non-accelerated methods, we know that there exists a regime where the convergence rate of Local SGD is strictly better than both mini-batch and single-machine SGD. If such a regime indeed exists, there must be at least some regime of $\tau$ and $h$, where even the time complexities of Local SGD are strictly better. What am I missing? Can the authors give an explicit proof that the complexity in 2.1 can never strictly improve over the one in Theorem 2.2? This is only alluded to in the contributions section, but not explicitly shown.

For context, about which regime I am looking at, note that when $\frac{LB}{\sqrt{KR}} < \sigma < \frac{LB\sqrt{MK}}{\sqrt{R}}$, then the dominant term in the convergence rate of the three algorithms is as follows:
- Single Machine SGD: $\frac{\sigma B}{\sqrt{KR}}$ which is strictly bigger than $\frac{LB^2}{KR}$ due to the first restriction on the range of $\sigma$ above;
- Mini-batch SGD: $\frac{LB^2}{R}$ which is strictly bigger than $\frac{\sigma B}{\sqrt{MKR}}$ due to the second restriction on the range of $\sigma$ above;
- Local SGD: $\frac{LB^2}{KR} + \frac{\sigma B}{\sqrt{MKR}}$, which, based on the best two bullets, is strictly better than both the previous algorithms.

Due to this, in the above regime, there must exist some regime of $\tau, h, \epsilon$ where Local SGD's time complexity is also better. If I am missing something, I am happy to be corrected.

---

> ### Author Response · Authors · 2025-11-20
>
> Dear Reviewer YJY4,
>
> Thank you for your thoughtful review and concerns.
>
> Concerning the weaknesses and questions you raised:
>
> > _Theorems 2.3 and 2.5 are wrong. First of all, since these results are derived from upper bounds on Local SGD's convergence, they do not represent time complexity lower bounds. Thus, the theorem statements themselves are wrong. The paper acknowledges this in the preceding text, but still chooses to write the incorrect version of these theorems. Furthermore, the proof of Theorem 2.5 is incorrect because it overlooks specific terms in the upper bounds when deriving the theorem. If the results being used were lower bounds, this would have been fine, but these results are upper bounds, and hence ignoring some terms would not give a correct picture: ignoring these terms makes the upper bounds invalid. Since these are the data heterogeneity terms, this is further potentially misleading in the heterogeneous setting._
>
> We respectfully disagree. Theorems 2.3 and 2.5 are correct within the scope of the text “_... using the analysis by Koloskova et al. (2020); Luo et al. (2025)_,” which might have been overlooked. In these theorems, we intended to take the best-known iteration rate for Local SGD and transform it into the corresponding time complexity (which, unfortunately, was not done by the authors). Nevertheless, we agree that this might lead to confusion, so we now refer to these results as “Corollaries” instead of “Theorems.” (we also do not indicate anymore that the corollaries are lower bounds) **This is precisely, at the very least, what the authors of the previous papers (Koloskova et al. (2020); Luo et al. (2025)) would obtain under Assumption 1.1.**
>
> Regarding your last comment, in the proof of Theorem 2.5 (now it is Corollary 2.2), when we derive the time complexity bounds, we ignore the terms with these extra dependencies. Notice that once we start taking them into account, the bound can only increase! In other words, in Corollary 2.2, instead of
> $$\\frac{L \\Delta}{\\varepsilon} + h \\left(\\frac{L \\Delta}{\\varepsilon} + \\frac{L \\sigma^2 \\Delta}{n \\varepsilon^2}\\right),$$
> we might get
> $$\\frac{L \\Delta}{\\varepsilon} + h \\left(\\frac{L \\Delta}{\\varepsilon} + \\frac{L \\sigma^2 \\Delta}{n \\varepsilon^2}\\right) + T(\\eta, \\rho, \\dots),$$
> where $T(\\eta, \\rho, \\dots) \\geq 0$ is an extra term, which depends on other parameters from the heterogeneous setting. So, our the derived result is valid. For our work and story, it is sufficient to show that the time complexity of Local SGD is not better than $\\sqrt{\\tau h \\left(\\frac{L^2 \\sigma^2 \\Delta^2}{\\varepsilon^{3}}\\right)} + h \\left(\\frac{L \\Delta}{\\varepsilon} + \\frac{L\\sigma^2 \\Delta}{n \\varepsilon^2}\\right).$ Even with this result, the gap is clear.
>
> > _In the convex setting, i.e., when comparing Theorems 2.1 and 2.2, we do not have the above issue, as the authors do look at the lower bound due to Glasgow et al. for Local SGD. Still, I am not convinced that conclusion is correct in all regimes as the authors claim. Please answer my question below regarding this._
>
> We added a formal lemma explicitly establishing our claim (Lemma 2.1). The proof has been included in the appendix. We hope this clarifies the point that was previously alluded to in the contributions section but not explicitly stated. We also explicitly provide a regime when Local SGD is strictly worse (see the new pdf). **Thus, our claim that Local SGD is never better but can be strictly worse is correct (see the new pdf).**
>
> As for the specific regime you highlighted (for remainder, in the homogeneous convex regime), under the assumption (***) $\\frac{L B}{\\sqrt{K R}} < \\sigma < \\frac{L B \\sqrt{n K}}{\\sqrt{R}}$ (we take $M = n$ as in the paper's notation), we agree that the rates of Hero/Single Machine SGD (with $K R$ iterations) and Minibatch SGD reduce respectively to $\\frac{\\sigma B}{\\sqrt{K R}}$ and $\\frac{L B^2}{R}$. According to Table 1, this follows from the inequalities $\\frac{\\sigma B}{\\sqrt{K R}} > \\frac{L B^2}{K R}$ and $\\frac{\\sigma B}{\\sqrt{n K R}} < \\frac{L B^2}{R}$ as you stated. Therefore, the rate of Hero SGD/Minibatch SGD is (dominated by) $\\min\\left\\{\\frac{\\sigma B}{\\sqrt{K R}}, \\frac{L B^2}{R} \\right\\}$ (up to constant factors).

---

> > ### Author Response · Authors · 2025-11-20
> >
> > > _continuing the previous point_
> >
> > But, we do not agree that the rate of Local SGD is dominated by $\\frac{L B^2}{K R} + \\frac{\\sigma B}{\\sqrt{n K R}}$. In the later we assume $n \\ge 2$ (otherwise Local SGD reduces to Hero SGD) and $K \\ge 1$. First, recall that Glasgow et al. (2022) established the sharp lower bound (in this sense that it matches the upper bound from [1]): $\\frac{L B^2}{K R} + \\frac{\\sigma B}{\\sqrt{n K R}} + \\min\\left\\{\\frac{L^{\\frac{1}{3}} \\sigma^{\\frac{2}{3}} B^{\\frac{4}{3}}}{K^{\\frac{1}{3}} R^{\\frac{2}{3}}}, \\frac{\sigma B}{\\sqrt{K R}}\\right\\}$ for Local SGD in the homogeneous and convex regime. Granted this, we have, by assumption (***)
> >
> > - first, $\\frac{\\sigma B}{\\sqrt{K R}} > \\frac{L B^2}{K R}$ and $\\frac{L^{\\frac{1}{3}} \\sigma^{\\frac{2}{3}} B^{\\frac{4}{3}}}{K^{\\frac{1}{3}} R^{\\frac{2}{3}}} > \\frac{L B^2}{K^{\\frac{2}{3}} R} \\ge \frac{L B^2}{K R}$ since $K \\ge 1$ so $\\frac{L B^2}{K R}$ is not dominant in the rate of Local SGD.
> >
> > - Moreover, since $n \\ge 2$ ($n \\ge 1$ would suffice) we have $\\frac{\sigma B}{\\sqrt{K R}} \\ge \\frac{\sigma B}{\\sqrt{n K R}}$ and
> >
> > $$ \\frac{L^{\\frac{1}{3}} \\sigma^{\\frac{2}{3}} B^{\\frac{4}{3}}}{K^{\\frac{1}{3}} R^{\\frac{2}{3}}} = \\frac{\\sigma B}{\\sqrt{n K R}} \\cdot \\frac{L^{\\frac{1}{3}} \\sigma^{\\frac{2}{3} - 1} B^{\\frac{1}{3}} \\sqrt{n}}{K^{\\frac{1}{3} - \\frac{1}{2}} R^{\\frac{2}{3} - \\frac{1}{2}}} = \\frac{\\sigma B}{\\sqrt{n K R}} \\cdot \\frac{L^{\\frac{1}{3}} K^{\\frac{1}{6}} B^{\\frac{1}{3}} \\sqrt{n}}{ \\sigma^{\\frac{1}{3}} R^{\\frac{1}{6}}} \\overset{(***)}{>} \\frac{\\sigma B}{\\sqrt{n K R}} \\cdot \\frac{L^{\\frac{1}{3}} K^{\\frac{1}{6}} B^{\\frac{1}{3}} \\sqrt{n}}{ \\left( \\frac{L B \\sqrt{n K}}{\\sqrt{R}} \\right)^{\\frac{1}{3}} R^{\\frac{1}{6}}} = \\frac{\\sigma B}{\\sqrt{n K R}} \\cdot n^{\\frac{1}{2} - \\frac{1}{6}} \\ge \\frac{\\sigma B}{\\sqrt{n K R}}, $$
> >
> > so, in the given regime, the rate of Local SGD is not dominated by the sum $\\frac{L B^2}{K R} + \\frac{\\sigma B}{\\sqrt{n K R}}$ but by the $\\min\\{\\ldots\\}$ itself. Moreover, under (***) we have
> >
> > $$ \\min\\left\\{ \\frac{\\sigma B}{\\sqrt{K R}}, \\frac{L B^2}{R} \\right\\} \\le \\min\\left\\{\\frac{L^{\\frac{1}{3}} \\sigma^{\\frac{2}{3}} B^{\\frac{4}{3}}}{K^{\\frac{1}{3}} R^{\\frac{2}{3}}}, \\frac{\sigma B}{\\sqrt{K R}}\\right\\}. \\quad (AAA)$$
> >
> > To do so, we distinguish the two sub-regimes
> > - if $L B \\sqrt{\\frac{K}{R}} \le \\sigma \\overset{(***)}{<} \\frac{L B \\sqrt{n K}}{\\sqrt{R}}$ then, $\\min\\left\\{\\frac{L^{\\frac{1}{3}} \\sigma^{\\frac{2}{3}} B^{\\frac{4}{3}}}{K^{\\frac{1}{3}} R^{\\frac{2}{3}}}, \\frac{\\sigma B}{\\sqrt{K R}}\\right\\} = \\frac{L^{\\frac{1}{3}} \\sigma^{\\frac{2}{3}} B^{\\frac{4}{3}}}{K^{\\frac{1}{3}} R^{\\frac{2}{3}}}$ (first term) but then in this sub-regime we have
> >
> > $$\\frac{L^{\\frac{1}{3}} \\sigma^{\\frac{2}{3}} B^{\\frac{4}{3}}}{K^{\\frac{1}{3}} R^{\\frac{2}{3}}} \\ge \\frac{L^{\\frac{1}{3}} \\left( L B \\sqrt{\\frac{K}{R}} \\right)^{\\frac{2}{3}} B^{\\frac{4}{3}}}{K^{\\frac{1}{3}} R^{\\frac{2}{3}}} = \\frac{L B^2}{R}, $$
> > hence (AAA) holds.
> >
> > - if $\\frac{L B}{\\sqrt{K R}} \\overset{(***)}{<} \\sigma < L B \\sqrt{\\frac{K}{R}}$ we have $\\min\\left\\{\frac{L^{\\frac{1}{3}} \\sigma^{\\frac{2}{3}} B^{\\frac{4}{3}}}{K^{\\frac{1}{3}} R^{\\frac{2}{3}}}, \\frac{\\sigma B}{\\sqrt{K R}}\\right\\} = \\frac{\\sigma B}{\\sqrt{K R}}$ (second term) and (AAA) also holds.
> >
> > Hence: not only Local SGD rate is driven by the $\\min\\{\\ldots\\}$ itself under (***), but this rate is never better than the best of Single-machine SGD/Minibatch SGD rates in the prescribed regime.

---

> ### Author Response · Authors · 2025-11-20
>
> > _continuing the previous point_
>
> Actually, we have the stronger result: if we take Hero SGD with $KR$ iterations (instead of $R$ as we do in the paper), it is possible to prove that **there do not exists regimes in which the rate of Local SGD is better than the minimum of the rates of both Hero SGD / Minibatch SGD**. This can be proved the following way:
>
> - if $\\sigma \\ge L B \sqrt{\\frac{K}{R}}$ then, as proved before, the rate of Local SGD is
>
> $$\\frac{L B^2}{K R} + \\frac{\\sigma B}{\\sqrt{n K R}} + \\min\\left\\{\\frac{L^{\\frac{1}{3}} \\sigma^{\\frac{2}{3}} B^{\\frac{4}{3}}}{K^{\\frac{1}{3}} R^{\\frac{2}{3}}}, \\frac{\\sigma B}{\\sqrt{K R}}\\right\\} = \\frac{L B^2}{K R} + \\frac{\\sigma B}{\\sqrt{n K R}} + \\frac{L^{\\frac{1}{3}} \\sigma^{\\frac{2}{3}} B^{\\frac{4}{3}}}{K^{\\frac{1}{3}} R^{\\frac{2}{3}}} \\ge \\frac{L B^2}{K R} + \\frac{\\sigma B}{\\sqrt{n K R}} + \\frac{L B^2}{R},$$
>
> which is bigger than the rate of Minibatch SGD,
>
> - if $0 \\le \\sigma < L B \\sqrt{\\frac{K}{R}}$ then
>
> $$ \\frac{L B^2}{K R} + \\frac{\\sigma B}{\\sqrt{n K R}} + \\min\\left\\{\\frac{L^{\\frac{1}{3}} \\sigma^{\\frac{2}{3}} B^{\\frac{4}{3}}}{K^{\\frac{1}{3}} R^{\\frac{2}{3}}}, \\frac{\\sigma B}{\\sqrt{K R}}\\right\\} = \\frac{L B^2}{K R} + \\frac{\\sigma B}{\\sqrt{n K R}} + \\frac{\\sigma B}{\\sqrt{K R}} \\ge \\frac{L B^2}{K R} + \\frac{\\sigma B}{\\sqrt{K R}}, $$
>
> which is bigger than the rate of Hero SGD (**with $KR$ iterations!**).
>
> Therefore, whatever the choice of input parameters, the time complexity of Hero SGD/Minibatch SGD is always better than that of Local SGD **in all possible regimes**.
>
> However, once we look at Hero SGD with **only** $R$ iterations (and **this is the setting in our paper**) then the conclusion is weaker and this is what we actually looked at and proved in the paper: **there exists a choice of $K$, namely $K = \\max\\{ \\left\\lceil \\sigma^2 / (L \\varepsilon n) \\right\\rceil, 1 \\}$ for which the time complexity of Hero SGD/Minibatch SGD is always better than the time complexity of Local SGD** and this is true **for every possible choice of $K$ for Local SGD** (you can allow different values of $K$ for Minibatch SGD and Local SGD and there exists a choice of $K$ for Minibatch SGD for which, whatever the number of local steps you made in Local SGD, the resulting time complexity cannot be better than the one of Hero SGD/Minibatch SGD). So, while there may exists some $K$ for which Local SGD may be faster (this can happen if $K$ is not optimally chosen for instance), this $K$ is not a parameter of the problem (like $\\varepsilon$, $L$ and $B$ or even $n$ depending on the setup) but is a hyperparameter than can be tuned. **In practice, we do not care about having the fastest method _for all values of $K$_, but we rather want to have _the optimal choice of $K$_ to minimize the time complexity. And this is exactly what we do in the paper (with Hero SGD having $R$ iterations).**
>
> > _The paper is inconsistent about stating and discussing the results for Hero SGD/single machine SGD. For instance, in Algorithm 3 we look at an algorithm with iterations, as opposed to iterations which would be the correct baseline to compare to. But when discussing the algorithm's guarantee in Table 1, the authors mix the different terms from these two algorithms: the optimization term seems to follow Algorithm 3, while the noise term for Hero SGD seems to suggest iterations. This is most probably a typo or misunderstanding about which single machine baseline is the correct comparison. Surprisingly in Theorem 2.2, the authors revert back to the correct baselines, with iterations. This is all very confusing, and would likely mislead an uninformed reader. I suggest describing in the pseducode only the variant with iterations, because this equalizes the compute each machine actually does in the parallel algorithms (Local SGD and mini-batch SGD)._
>
> While it may seem fair to compare Hero SGD, Minibatch SGD and Local SGD by running Hero SGD for $K R$ iterations, we stick with the convention of running Hero SGD for only $R$ iterations. It does not make sense to add a ``fake'' parameter $K$, since Hero SGD does not calculate local steps or mini-batches; it only runs $R$ iterations.

---

> ### Author Response · Authors · 2025-11-20
>
> > _Morally, we don't need to look at time complexity to conclude that in the homogeneous setting Local SGD has a limited utility. The seminal result of Woodworth et al. already shows that accelerated mini-batch SGD and accelerated single machine SGD together are min-max optimal. Yes, this leaves open the question whether a similar dichotomy can be shown for non accelerated algorithms, but it is not a very interesting gap._
>
> We respectfully disagree. **The fact that Local SGD is not only no faster than accelerated mini-batch SGD and accelerated single-machine SGD, but also no faster than their non-accelerated counterparts, is an important contribution. Moreover, we also show that Local SGD is provably slower than non-accelerated mini-batch SGD and accelerated single-machine SGD, which is very surprising and interesting! Considering the popularity of Local SGD, even this result is sufficient for a publication.**
>
> **Even the reviewer was surprised by this result and could not believe it. We strongly believe that this is an indication that our work is very important for the community!**
>
> > _Even if in principle, unaccelerated Local SGD could match the min-max optimal rate, that would still not explain its practical effectiveness. More than anything, their lower bound doesn't show that Local SGD is a bad algorithm, but that rather the homogeneous convex smooth setting is not "rich" enough to actually benefit from local update steps. In fact, this is the motivation for studying mildly heterogeneous settings, where the benefit of collaboration is not limited to just variance reduction in the noise term, but collaboration can genuinely help clients explore the loss landscape. The results of Patel et al. highlight this, as they show that the dichotomy pointed out by Woodworth et al. does not really hold in the heterogeneous setting, because single machine SGD is no longer a reasonable baseline._
>
> We appreciate the results by Patel et al. These are strong and important contributions. However, their research is not directly related to ours; we consider both non-convex and convex settings. Their improvements apply in regimes where the functions are convex quadratic functions or close to them ($Q$ is very small in third-order smoothness). Real-world problems are non-quadratic and highly non-linear. Thus, explaining the superiority of Local SGD in truly non-linear settings remains an open problem.
>
> > _Finally, this highlights that several important references are missing in this discussion. I would encourage the authors to re-write with a more through inspection of what the current results imply in the accelerated and non-accelerated settings. The thesis by Patel does contain many useful missing references._
>
> We agree that some important references and discussion are missing. We've already added them.
>
> > _Theorem 3.1 is not interesting. It makes a point, which has been underlined again and again (by some of the works cited in the paper itself): Local SGD with two step-sizes can recover mini-batch SGD. It is not any more insightful to show this for a specific choice of step-sizes. Granted, the step-sizes for which Theorem 3.1 shows this are not extreme, but even the SCAFFOLD paper's Theorem 1 already shows this. What is the novelty here?_
>
> Our paper should be viewed as a single, monolithic result with many novel contributions, and this is one of them. Theorem 3.1 is not the only result in this paper, and we already acknowledge in the text that the analysis with two step sizes is not new. In this particular theorem, our goal is to show that the suboptimal time complexity of the classical Local SGD method can be easily fixed.
>
> > _What would be interesting is if the authors could identify a theoretical setting, where Local SGD with two step-sizes not only matches but beats both mini-batch and single-machine SGD. Similarly, the fact that there is a gap between Local SGD with one and two step-sizes was also formalized again by the lower bound instances of Patel et al. (see their Remark 5)._
>
> It is impossible to outperform Minibatch SGD and Hero SGD under the assumptions we consider, due to the known lower bounds on time complexity. As we explain the previous comments, the result by [1] or Patel et al. hold if the function is quadratic or close to quadratic. However, in real applications the objectives are non-quadratic, highly non-convex, and non-linear, and to the best of our knowledge, the appropriate assumptions needed to show the superiority of Local SGD in such highly non-linear settings remain unknown.

---

> ### Author Response · Authors · 2025-11-20
>
> > _It is unclear what is the point of Theorem 4.1, it attains the same convergence rate as Theorem 3.1. If the point is to show that a bigger step-size schedule does not hurt or helps in some way, then clearly the theoretical setting considered in the paper is not rich enough to demonstrate that. Empirically, previous works have already investigated this quite extensively, for instance Charles and Konecny do this on several benchmarks. So what would be interesting is to consider some theoretical setting, where such a scheme could have provable optimization benefits._
>
> Once again, this is just one result among many others. The point is precisely to show that the local step sizes can be increased even further while still achieving the optimal time complexity. To the best of our knowledge, this is the first Local SGD method with this step-size schedule and optimal time complexity. This result is theoretical, not empirical.
>
> ---
>
> [1]: B Woodworth et al. Is local SGD better than minibatch SGD?

---

### Official Review · Reviewer_14YH · 2025-11-06

**Soundness:** 3
**Presentation:** 3
**Contribution:** 2
**Rating:** 4
**Confidence:** 4

**Summary:**

The paper re-evaluates Local SGD / FedAvg through time complexity (wall-clock) rather than iteration complexity. Under a simple compute/comm model (per-grad cost $h$, per-sync cost $\tau$, it argues canonical Local SGD is never better—and can be strictly worse—than Minibatch SGD or “Hero” (single-worker) SGD. It then proposes (i) Dual Local SGD (two step sizes, correcting the global aggregation scaling) and (ii) Decaying Local SGD (larger, decaying local steps), and shows these match the optimal time complexity (up to log factors) achieved by Minibatch/Hero in nonconvex settings.

**Strengths:**

- The paper is generally well written and well structured; algorithms and tables are clear. The Contributions section and Table 2 succinctly convey the main message.

- Reframing the discussion on Local SGD in terms of time complexity—with explicit compute and communication costs ($h$, $\tau$) exposes a gap between iteration-rate folklore and wall-time behavior.

**Weaknesses:**

- The claimed contribution regarding the “correct” scaling for Canonical Local SGD—namely $1/\sqrt{n}$ rather than $1/n$—does not seem particularly novel. Several prior works (e.g., Khaled et al., 2020; Woodworth et al., 2020) already employ step sizes $\eta = \Theta(\sqrt{n})$, which, when combined with $1/n$ averaging, effectively yields an overall $O(1/\sqrt{n}​)$ scaling.

- The stated time-complexity lower bounds for Local SGD (Theorems 2.1, 2.3, 2.5, 3.1) become trivial in the edge case where the per-gradient compute cost $h \rightarrow 0$ (i.e., computation is negligible relative to communication), collapsing to zero.

- Much of the analysis appears to closely follow existing literature (e.g., Tyurin & Richtárik, 2023; Tyurin & Sivtsov, 2025), which makes the technical novelty feel incremental.

- The numerical section offers limited insight. The main toy example is a one-dimensional strongly convex problem, yet Figure 1 shows very slow convergence even with a logarithmic y-axis, and the methods converge to different objective values. Figure 2 on standard ML tasks exhibits similar issues.

**Questions:**

- Theorem 2.5 presents results for both Local SGD and SCAFFOLD. Because SCAFFOLD uses control variates, its rates typically avoid explicit dependence on data heterogeneity, whereas Local SGD usually does depend on this. In your result, this dependence is not visible. Could you clarify: (i) whether you absorb heterogeneity into other constants, and (ii) how the heterogeneity parameter would appear if stated explicitly? A version of the theorem that makes the dependence explicit would help.

- In Theorem 3.1, the phrase “when combined with Hero SGD” is ambiguous. Do you mean a hybrid algorithm that interleaves Local SGD with a single-worker (“Hero”) phase, a fallback schedule, or simply a comparison/baseline in the bound? Please spell out the precise algorithmic composition and schedule implied by “combined”.

---

> ### Author Response · Authors · 2025-11-20
>
> Dear Reviewer 14YH,
>
> Thank you for your thoughtful review and comments.
>
> Concerning the points you raised regarding weaknesses:
>
> > _The claimed contribution regarding the “correct” scaling for Canonical Local SGD--namely $1 / \\sqrt{n}$ rather than $1 / n$-- does not seem particularly novel. Several prior works (e.g., Khaled et al., 2020; Woodworth et al., 2020) already employ step sizes $\\eta =\\Theta(\\sqrt{n})$, which, when combined with $1 / n$ averaging, effectively yields an overall $O(1 / \\sqrt{n})$ scaling._
>
> We agree with the reviewer that the “correct” scaling of $1 / \\sqrt{n}$ is not new. We acknowledge this saying "_While our work is not
> the first to consider Dual Local SGD in the literature..._" However, to the best of our knowledge, previous works typically _couple_ the local and global stepsizes, making it unclear how each individually influences the convergence rate. Our analysis helps clarify this relationship. For example, in SCAFFOLD, the authors consider the update rule
>
> $$x^{t + 1} \\gets x^t - \\frac{\\widehat{\\eta}\_{\\ell} \\widehat{\\eta}\_g}{n} \\sum\_{i = 1}^n \\sum\_{j = 0}^{K - 1} \\nabla f\\left( z^t\_{i,j}; \\xi^t\_{i, j} \\right), \\quad (1)$$
>
> with local and global stepsizes $\\widehat{\\eta}\_{\\ell}$ and $\\widehat{\\eta}\_g$, which in our notation correspond to $\\eta\_{\\ell} = \\widehat{\\eta}\_{\\ell}$ and $\\eta\_g = \\widehat{\\eta}\_{\\ell} \\widehat{\\eta}\_g / n$. Decoupling the local and global stepsizes, as we do, allows for a clearer understanding of their individual roles in convergence. Moreover, our analysis naturally recovers Minibatch SGD as the special case $\\eta\_{\\ell} = 0$, in contrast to the update rule in (1).
>
> > _The stated time-complexity lower bounds for Local SGD (Theorems 2.1, 2.3, 2.5, 3.1) become trivial in the edge case where the per-gradient compute cost $h \\to 0$ (i.e., computation is negligible relative to communication), collapsing to zero._
>
> This is not a weakness of our paper or results. The fact that the time complexities collapse to zero as $h \\to 0$ is **expected!** Notice that the time complexity of Hero SGD is $h \\left(\\frac{L B^2}{\\varepsilon} + \\frac{\\sigma^2 B^2}{\\varepsilon^2}\\right)$ because it does not communicate and only calculates stochastic gradients, which takes $h$ seconds. So, if $h \to 0,$ then the time complexity tends to zero.
>
> In the regime, Hero SGD is the best strategy, and Local SGD reduces to Hero SGD. For Local SGD in Theorem 2.1, the optimal choice is $K \\to \\infty$ when $h \to 0,$ the method effectively reduces to Hero SGD.
>
> There are not contradictions and this is not a weakness of the results.
>
> > _Much of the analysis appears to closely follow existing literature (e.g., Tyurin, Richtárik, 2023; Tyurin, Sivtsov, 2025), which makes the technical novelty feel incremental._
>
> Similarly to virtually all previous optimization works, we prove descent lemmas, variance bounds, and so forth. However, there are many new details. For instance, to prove the result for Decaying Local SGD, which employs larger step sizes, we use a new and important Lemma C.2. Moreover, our Theorem 2.1 and Theorem 2.2 are new.
>
> > _The numerical section offers limited insight. The main toy example is a one-dimensional strongly convex problem, yet Figure 1 shows very slow convergence even with a logarithmic y-axis, and the methods converge to different objective values. Figure 2 on standard ML tasks exhibits similar issues._
>
> The methods converge after at most 20–50 global iterations in Figure 1. Why is that considered slow convergence? Dual Local SGD converges faster to lower function values, which clearly underscores its superiority.
>
> We want to stress what we wrote in the paper: the canonical Local SGD method (FedAvg) remains one of the most widely evaluated and tested algorithms in distributed and federated learning, and there is no doubt that it is a strong method for practical optimization tasks. Our experiments confirm this. The main goal of this paper is to explain that the previous **theoretical comparison** in Table 1 might be misleading, and a better one is presented in Table 2. We analyze the theoretical properties of the methods.

---

> ### Author Response · Authors · 2025-11-20
>
> Regarding your questions,
>
> > _Theorem 2.5 presents results for both Local SGD and SCAFFOLD. Because SCAFFOLD uses control variates, its rates typically avoid explicit dependence on data heterogeneity, whereas Local SGD usually does depend on this. In your result, this dependence is not visible. Could you clarify: (i) whether you absorb heterogeneity into other constants, and (ii) how the heterogeneity parameter would appear if stated explicitly? A version of the theorem that makes the dependence explicit would help._
>
> In the proof of Corollary 2.2 of the updated PDF, when we derive the time complexities, we ignore the terms with these extra dependencies. Notice that once we start taking them into account, the time complexity can only increase. In other words, in Corollary 2.2, instead of
> $$\\frac{L \\Delta}{\\varepsilon} + h \\left(\\frac{L \\Delta}{\\varepsilon} + \\frac{L \sigma^2 \Delta}{n \\varepsilon^2}\\right),$$
> we might get
> $$\frac{L \Delta}{\varepsilon} + h \left(\frac{L \Delta}{\varepsilon} + \frac{L \sigma^2 \Delta}{n \varepsilon^2}\\right) + T(\\eta, \\rho, \\dots),$$
> where $T(\\eta, \\rho, \\dots)$ is an extra term.
>
> The reason why we ignore $T(\\eta, \\rho, \\dots)$ is that even without this term, we can show a strict gap between the methods.
>
> > _In Theorem 3.1, the phrase “when combined with Hero SGD” is ambiguous. Do you mean a hybrid algorithm that interleaves Local SGD with a single-worker (“Hero”) phase, a fallback schedule, or simply a comparison/baseline in the bound? Please spell out the precise algorithmic composition and schedule implied by “combined”._
>
> Thank you for pointing out this ambiguous term. The idea we had in mind, when saying “combined”, is taking the _best_ of the two methods Local SGD and Hero SGD (single-worker SGD). Concretely, as we are dealing with time complexities, this means running both methods in parallel (from the same starting point), and outputting the result of whichever finishes first. We updated the PDF document with a footnote.
>
> We hope this clarifies the intended meaning.

---

### Author Response · Authors · 2025-12-03

We thank the reviewers for their careful reading and constructive feedback. In response to the reviews, we have revised the manuscript to address the questions, suggestions, and technical concerns raised across the reviews, especially those of Reviewer YJY4, whose detailed critique helped us significantly strengthen the clarity, rigour, and presentation of our main theoretical results. Below, we summarize the key changes.

---

**Reviewer YJY4.**  We take this concern very seriously.  To remove ambiguity and address every correctness point you raised, we performed the following changes:

- (i) renamed ambiguous statements (Theorems\,2.3/2.5) as "Corollaries" instead of "Theorems", and we also do not indicate anymore that the corollaries are lower bounds. Yet, the resulting time complexities would be **at the very least, what the authors of the previous papers (Koloskova et al. (2020); Luo et al. (2025)) would obtain under our Assumption 1.1.**

- (ii) added a new *Lemma 2.1*, with full proof in the appendix that gives an explicit, case-by-case algebraic proof showing that under our time model, Local SGD is not strictly better than the best of Minibatch/Hero SGD. We also explicitly provide in the main part of our manuscript a regime when Local SGD is arbitrarily worse Minibatch/Hero SGD.

- (iii) to answer the question raised by reviewer YJY4, we specifically analyze rigorously the regime highlighted and, more generally, we show that if Hero SGD runs for $KR$ iterations (as suggested by the reviewer), there **do not** exist a regime where Local SGD offers a strict improvement. We strengthen this result further by considering Hero SGD with $R$ iterations only, and we show that **there exists a choice of $K$, namely $K = \\max\\{ \\left\\lceil \\sigma^2 / (L \\varepsilon n) \\right\\rceil, 1 \\}$ for which the time complexity of Hero SGD/Minibatch SGD is always better than the time complexity of Local SGD** and this is true **for every possible choice of $K$ for Local SGD**: this constitutes the main part of our work. For overall clarity, both results, with detailed proofs, can be found in the rebuttal below.


Overall, as we argued in the rebuttal the fact that Local SGD is not only no faster than accelerated mini-batch SGD and accelerated single-machine SGD, but also no faster than their non-accelerated counterparts, **is an important contribution**. Moreover, we also show that Local SGD is provably slower than non-accelerated mini-batch SGD and non-accelerated single-machine SGD, **which is very surprising and interesting!** Especially considering the popularity of Local SGD.

We observed that even the **reviewer was surprised by this result and could not believe it:** in the review comment (in "Questions"), the reviewer tried to explain that Local SGD can be better, which is false, and we prove it in detail in our revision of the paper and in the rebuttal comments. **We strongly believe that this is an indication that our work is very important for the community.**

---

**Reviewer 14YH.**  We acknowledged prior work on local and global step-size choices for Local SGD, and clarified our contribution: *decoupling* local and global step-sizes to make each step-size's role explicit. We also carefully explained that there is not need to display the full heterogeneity terms in our time complexity as we can show a strict gap between the methods without it. We also clarified the SCAFFOLD comparison as asked.

---

**Reviewer K8KK.**  We clarified and tightened the modelling and empirical presentation: Assumption 1.1 now specifies **exact** execution times ($h, \tau$) to remove the "at most" ambiguity; we updated the experiments to show wall-clock time (seconds) on the $x$-axis and added Minibatch SGD to the plots. We have also corrected missing $O(\\cdot)$ and $\\Omega(\\cdot)$ notation for clarity, where appropriate.

---

**Reviewers 8mzB and ZKTy.**  We incorporated practical suggestions: all experiment plots now use wall-clock time. We also improved the clarity of our manuscript following your helpful comments.

---

**Concluding remark.**  We sincerely appreciate the time and effort the reviewers devoted to engaging with our submission. While this year reviewing process has been affected by unusual external circumstances, we are deeply grateful for the thoughtful critiques, which helped us substantially strengthen the clarity, correctness, and presentation of our work.

We hope that the revised version makes the contributions clearer, highlights the conceptual importance of our theoretical findings, and reflects the care we put into addressing each reviewer's points.

Authors

---

### Meta-Review · Area_Chair_9smp · 2026-01-05

**Summary:**

## Summary
This paper studies the time complexity of local SGD, minibatch SGD, hero SGD (where SGD is run on a single device), and other new variants of local SGD such as dual local SGD and decaying local SGD. The paper argues that the iteration complexity (the number of communication rounds) of local SGD is “better” than minibatch SGD but if we consider the time complexity, i.e., taking into account the time required for gradient evaluations and synchronization, then the picture changes: local SGD is no better than minibatch SGD and could be significantly worse in both convex and nonconvex setups. The authors then propose to study dual local SGD, where the updates use an inner step size $\eta_\ell$ and an outer step size $\eta_g$, and show that if the inner step size $\eta_\ell \leq \sqrt{n}\eta_g$ ($n$ being the number of devices), then dual local SGD is optimal in terms of time complexity, as minibatch SGD is. The authors also propose a time-varying version decaying local SGD with a similar optimal time complexity.

## Reviewer Concerns
Major concerns raised by reviewers can be summarized as follows.
- **Incorrect claims on lower bounds**. Reviewer YJY4 pointed out that Theorems 2.3 and 2.5 (now Corollaries 2.1 and 2.2) are incorrect, as the authors derive lower bounds from existing upper bounds. Reviewer K8KK also pointed out that Assumption 1.1 states the execution time as “at most,” hence making claims on lower bounds invalid.
- **Claims that “local SGD is always no better”**. Reviewer YJY4 argued that, since there are known regimes where local SGD outperforms (in terms of iteration complexity) minibatch SGD and Hero SGD, at least for some $\tau$ and $h$ there should be regimes where local SGD is better in terms of time complexity.
- **Motivation and novelty of Theorems 3.1 and 4.1**. Reviewer YJY4 questioned the novelty and motivation for proving Theorem 3.1 and 4.1, as the SCAFFOLD paper already shows convergence under the considered step size regime. Reviewer 14YH also pointed out that the claimed correct $1/\sqrt{n}$ scaling is not novel.
- **Limited empirical validation**. Most reviewers pointed out issues with the experiment section: 1) the experiments are small-scale, 2) experiments do not agree well with theoretical results, 3) no wall-clock time comparison is given.
- **Minor writing issues**: Reviewers pointed out minor clarity issues such as inconsistency when stating Hero SGD results, missing $O(\cdot)$ and $\Omega(\cdot)$ notation, “when combined with Hero SGD” is ambiguous, the difference between SCAFFOLD and local SGD is not revealed in Corollary 2.2 statement, etc.

**Reviewer Concerns:**

Unfortunately, the discussion period closed before we received feedback from four out of the five reviewers. Based on my reading, the authors’ responses can be summarized as follows.
- **Incorrect claims on lower bounds**.
  - Regarding the issue with Theorems 2.3 and 2.5, the authors renamed them as Corollaries 2.1 and 2.2, but claimed that they take the best-known iteration rate for local SGD and transform it into the corresponding time complexity. **In my judgment, I do not find this explanation convincing, as the paper concerns provable gaps between local SGD and other methods**. It is only rigorous to compare between lower bounds of local SGD and upper bounds of other methods when we want to claim that local SGD is no better. In Theorem 2.5, the authors also take an upper bound, treat it as a lower bound, and then exclude an additive factor from it to derive a “lower bound,” which I find to be **unfounded**.
  - As for the “at most $\tau$” assumptions, the authors revised them to “exactly”.

- **Claims that “local SGD is always no better”**. For the counterargument of Reviewer YJY4, the authors claimed that the dominant term in Local SGD lower bound in fact comes from the third term. I did not have the chance to fully validate the authors’ arguments, but to my understanding, the apparent contradiction with existing results identifying regimes where local SGD is better (Woodworth et al 2020) may have come from the fact that this submission treats $K$ as a hyperparameter one can choose whereas the existing papers treated $K$ as given. (The choice of $K$ by the authors leads to $K \propto R$, and it is known from Woodworth et al 2020 that minibatch SGD does better in this regime.) While this sounds reasonable to take the best $K$ for each algorithm, this also brings up some questions such as: If $K$’s can be chosen differently for minibatch vs local SGD, are we really making fair comparisons? Can we always choose $K$ whatever we want? The next revision of the paper should better address these questions.

- **Motivation and novelty of Theorems 3.1 and 4.1**.
  - In their response to Reviewer YJY4, the authors noted that the goal of Theorem 3.1 is to show that the suboptimal time complexity of the classical Local SGD method can be easily fixed, and the point of Theorem 4.1 is to show that the local step sizes can be increased even further.
  - As for the novelty of the $1/\sqrt{n}$ scaling, the authors emphasized that their analysis decouples the two step sizes.

- **Limited empirical validation**. The authors updated the plots to show the time on the x-axis. They claimed that local SGD is one of the most widely evaluated and tested algorithms, and their experiments confirm this, and that the small-scale experiments are meant to compare the methods under controlled conditions. In my view, I do not fully agree with this response, because the primary purpose of numerical experiments in a theory paper is to **validate and corroborate** the theory. It would have been nicer to see the benefit of the proposed methods in larger scale experiments.

- **Minor writing issues**: The authors clarified these minor issues.

**Reviewer Scores:**

The initial reviews had scores 8/6/4/4/2. Here is what I expect about the reviewers’ final assessment, focusing on the negative reviewers:
- I believe major concerns raised by **Reviewer YJY4** (score 2) remain outstanding. The paper claims results on lower bounds based on existing results on upper bounds. After excluding the “wrong” results, the contribution of the paper is significantly weaker. Given this, I expect that the reviewer would not have raised their score. Regardless of other reviewers’ assessments, I believe this alone can be grounds for rejection.
- I concur with **Reviewer 14YH**’s (score 4) concern that the claimed correct $1/\sqrt{n}$ scaling is not novel. Although the authors emphasized their decoupled learning rates, it is hard to see why the decoupling is a technically significant contribution. I expect that the reviewer’s rating would have remained the same.
- The concerns by **Reviewer K8KK** (score 4) were handled relatively better. This reviewer may or may not have raised the score after reading the rebuttal.

The time complexity analysis has a good potential to show that local SGD is no better than non-accelerated minibatch SGD and hero SGD in general. However, given the flawed derivations of lower bounds, the advantage is now limited to the convex homogeneous setting, which weakens the paper’s contribution. I also am not sure about the significance of Theorems 3.1 and 4.1. Despite the decoupling, the $1/\sqrt{n}$ scaling is not new, and the claim of “larger step size” in Theorem 4.1 is a slight overclaim because the step sizes are chosen to satisfy $\sum_{j=0}^{K-1} \eta_{j}^2 \lesssim \eta_g^2 b \asymp \eta_g^2 nK$ anyway. If we don’t ignore the log factor, this means that $\eta_j$ needs to be smaller than $\sqrt{n} \eta_g$ for larger values of $j$. Overall, the contributions of the paper are of limited significance and novelty, and the paper does not meet the bar for acceptance at ICLR.

---

### Decision · Program_Chairs · 2026-01-26

Reject